# MEASURING VISION-LANGUAGE STEM SKILLS OF NEURAL MODELS

**Jianhao Shen**[1,2*]**, Ye Yuan**[1,2,3*]**, Srbuhi Mirzoyan**[1,2,3]**, Ming Zhang**[1,2,3†]**, Chenguang Wang**[4†]

[1]School of Computer Science, Peking University
[2]National Key Laboratory for Multimedia Information Processing, Peking University
[3]Peking University-Anker Embodied AI Lab
[4]Washington University in St. Louis
{jhshen,yuanye_pku,mzhang_cs}@pku.edu.cn, srbuhimirzoyan@stu.pku.edu.cn
chenguangwang@wustl.edu

## ABSTRACT

We introduce a new challenge to test the STEM skills of neural models. The problems in the real world often require solutions, combining knowledge from STEM (science, technology, engineering, and math). Unlike existing datasets, our dataset requires the understanding of multimodal vision-language information of STEM. Our dataset features one of the largest and most comprehensive datasets for the challenge. It includes 448 skills and $1,073,146$ questions spanning all STEM subjects. Compared to existing datasets that often focus on examining expert-level ability, our dataset includes fundamental skills and questions designed based on the K-12 curriculum. We also add state-of-the-art foundation models such as CLIP and GPT-3.5-Turbo to our benchmark. Results show that the recent model advances only help master a very limited number of lower grade-level skills (2.5% in the third grade) in our dataset. In fact, these models are still well below (averaging 54.7%) the performance of elementary students, not to mention near expert-level performance. To understand and increase the performance on our dataset, we teach the models on a training split of our dataset. Even though we observe improved performance, the model performance remains relatively low compared to average elementary students. To solve STEM problems, we will need novel algorithmic innovations from the community. [1].

## 1 INTRODUCTION

STEM, namely, science, technology, engineering, and math, is the basis of solving a wide set of real-world problems. This helps solve hard problems to better understand the world and universe, such as modeling gravitational waves and protein structures, proving mathematics theorem, designing new principles for quantum computing, and engineering the James Webb telescope. Mirroring real-world scenarios, understanding multimodal vision-language information is vital to a great variety of STEM skills. For example, we are asked to compute the magnetic force given a diagram in physics. Geometry problems often require mathematical reasoning based on diagrams.

The challenges of the real world often require solutions that combine knowledge from STEM. Existing vision-language benchmarks, however, often concentrate on evaluating one of the STEM subjects. For example, IconQA (Lu et al., 2021b) and Geometry3K (Lu et al., 2021a) focus on evaluating mathematics understanding, while ScienceQA (Lu et al., 2022) examines science related skills. Other multimodal datasets such as VQA (Antol et al., 2015) and CLEVR (Johnson et al., 2017) are not specifically designed for STEM. Another set of benchmarks often includes textual STEM skill sets, where images are converted to LaTeX or formal languages (Hendrycks et al., 2021a;b).

---

[*]Equal contribution.

[†]Corresponding authors.

[1]The dataset and leaderboard are available at https://huggingface.co/datasets/stemdataset/STEM

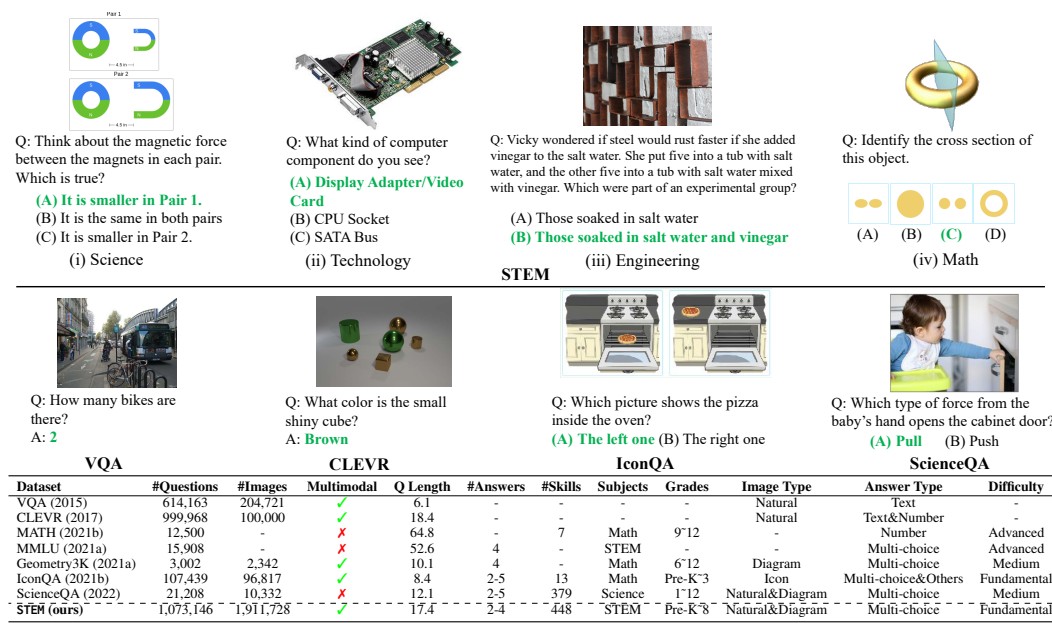

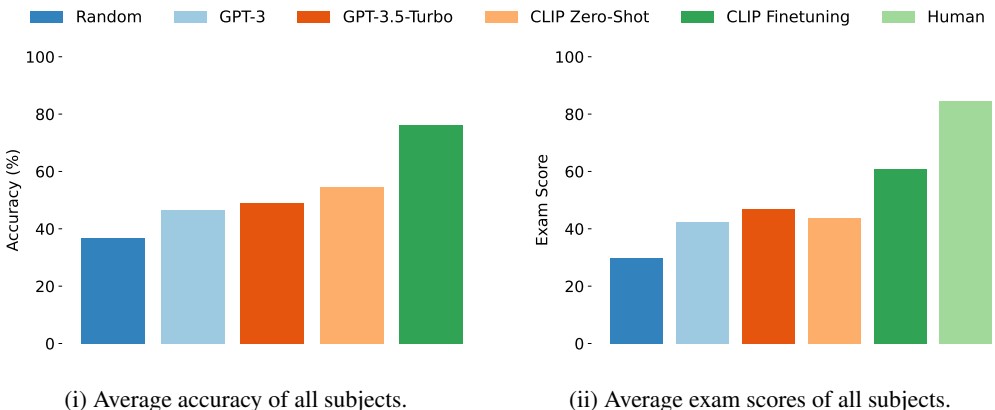

(a) Comparison between STEM and existing datasets. Upper: examples of STEM and other datasets. Lower: key statistics of STEM and other datasets. "#Questions", "#Images", "#Answers", "#Skills" denote the number of questions, images, answers, skills. "Multimodal" indicates whether every question of a dataset contains both text and image. "Q Length" means the average question length.

(b) Neural model performance on STEM dataset.

Figure 1: Summary of our dataset and results.

In this paper, we create a new challenge to test the STEM skills of neural models. We collect a large-scale multimodal dataset, called STEM, consisting of $448$ skills and $1,073,146$ questions spanning across all four STEM subjects. STEM provides the largest set of both skills and questions among existing datasets. Figure 1(a) shows the comparison of its key statistics with other datasets. The dataset consists of multi-choice questions, and Figure 1(a) shows an example for each subject. STEM is multimodal as we exclude a question if both the question and its answers are text. Each question consists of a question text with an optional image context. The corresponding answers to the question are either in text (Figure 1(a)(i)) or image (Figure 1(a)(iv)). The design of skills in STEM is important: we focus on fundamental skills based on the K-12 curriculum. This enables us to present a diverse and comprehensive STEM skill set. More importantly, this facilitates the understanding of neural models from different perspectives such as at skill level. We use IXL Learning (Learning, 2019) as our main data source to create STEM as it aligns best with our design principle.

The STEM dataset is challenging. Although our dataset focuses on the fundamentals of STEM, its multimodal nature makes it very difficult for modern neural models. Different from previous

multimodal benchmarks, we include foundation models such as the state-of-the-art vision-language model, CLIP (Radford et al., 2021), and the large language model, GPT-3.5-Turbo (Ouyang et al., 2022). While these models are able to advance the model performance compared to the near random-chance performance of previous neural models, they still drop the performance by averaging $54.7\%$ compared to that of average elementary students. For example, the models are only capable of understanding $2.5\%$ third-grade skills. Notably, our model results are evaluated quantitatively under the same real-world exam environment as humans. Instead of manual evaluation which is expensive, we simulate the conditions of IXL's online exams and use their scoring system to grade the model results. Compared to accuracy, this score (Bashkov et al., 2021) aims to measure humans' true understanding of skills by integrating the learning progress into the final score calculation. While the majority of existing benchmarks do not yet provide detailed meta information for analysis, the design of STEM supports deep performance analysis at different granularities, e.g., at a particular subject, skill, or grade level. For example, we show that basic math skills are still challenging for existing models. This is often due to the models failing to parse the images that are of great importance to mastering multimodal skills (e.g., geometry). To understand and increase the model performance on STEM, we teach models on a large-scale training split of STEM. However, the model performance still remains relatively low compared to general elementary students, not to mention near expert-level performance.

Our contributions are as follows. (i) We create a new dataset, called STEM, to benchmark the multimodal STEM skills of neural models. STEM is the largest dataset among existing datasets. Its design focuses on fundamental skills in the K-12 curriculum. This enables diverse and comprehensive tests across all STEM subjects. To facilitate future research, we also contribute a large-scale training set in STEM. STEM is challenging and useful to help advance models to solve more real-world problems. (ii) We benchmark a wide set of neural models including foundation models such as GPT-3.5-Turbo and CLIP on STEM. The meta information in STEM (e.g., skills and grades) supports a deeper understanding of model performance, and helps point out important shortcomings of existing models. (iii) We show current neural model performances are still far behind that of average elementary students in terms of STEM problem solving. We conclude important insights that suggest new algorithmic advancements from the community are necessary for understanding STEM skills.

## 2 THE STEM BENCHMARK

### 2.1 DATASET

We create a massive dataset, called STEM to test the STEM problem solving abilities. Unlike existing benchmarks, STEM features a large-scale multimodal dataset covering all STEM subjects spanning science, technology, engineering, and mathematics. We split the dataset into a train set, a validation set, and a test set for model development and evaluation. The overall dataset statistics are included in Table 1. More details of STEM dataset are described in the appendix.

**Attributes** Our dataset includes the following key attributes to support deep analysis of model performances. (i) **Subjects.** There are four subjects in STEM, namely science, technology, engineering, and math. We follow this high-level concept to create our dataset. (ii) **Skills.** We design skills according to the U.S. National Education and California Common Core Content Standards. This design also aligns with the skill categorization of our data resources (details are below) and closely follows recent studies (Hendrycks et al., 2021b; Lu et al., 2021b). (iii) **Grades.** We use the grade information of our dataset resources in STEM. STEM does not contain grade information for the technology subset as its raw data does not provide the grade-level information. (iv) **Questions.** Each question in STEM is a multi-choice question and is multimodal. We exclude a question if both the question and its answers are text. Each question belongs to a particular skill, hence a subject.

**Science** Science includes branches of domain knowledge focusing on testing reasoning abilities. Subject areas include biology, chemistry, physics and so on. Science tests specific domain knowledge, e.g., physics tests understanding of fundamental physics principles. It includes skills examining basics of science such as identifying properties of an object or calculating density. For example, to test the skill of comparing magnitudes of magnetic forces, an example question in Figure 1(a)(i) will be asked. We collect questions from IXL Science. Its skills and questions are designed based

Table 1: STEM dataset statistics.

| Subject | #Skills | #Questions | Average #A | #Train | #Valid | #Test |
|---|---|---|---|---|---|---|
| Science | 82 | 186,740 | 2.8 | 112,120 | 37,343 | 37,277 |
| Technology | 9 | 8,566 | 4.0 | 5,140 | 1,713 | 1,713 |
| Engineering | 6 | 18,981 | 2.5 | 12,055 | 3,440 | 3,486 |
| Math | 351 | 858,859 | 2.8 | 515,482 | 171,776 | 171,601 |
| Total | 448 | 1,073,146 | 2.8 | 644,797 | 214,272 | 214,077 |

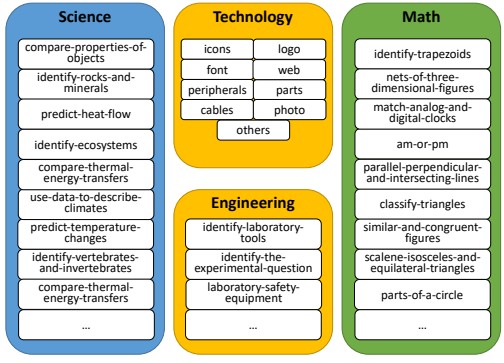

Figure 2: A summary of skills.

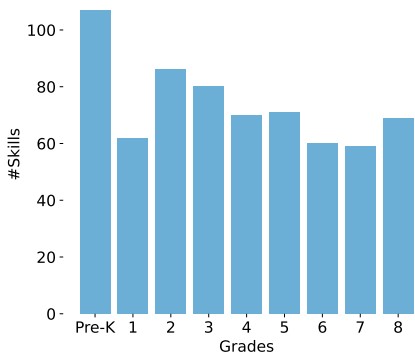

Figure 3: #Skills per grade.

on U.S. National Education and California Common Core Content Standards. It includes questions from second grade to eighth grade. We also processed the data such as deduplicating questions and randomly shuffling the order of answers to each question. We exclude a question if both the question and its answers are text.

**Technology**  Technology includes principles that test the knowledge of empirical methods. This subject mainly includes computer science. An example is included in Figure 1(a)(ii). It includes fundamental skills such as identifying parts of a computer or the basics of programming languages. We collect the questions from Triviaplaza Computer, which includes questions for tech interviews. To the best of our knowledge, STEM provides the first technology problem set for the multimodal test.

**Engineering**  This engineering subset includes a skill set that covers fundamental engineering practices ranging from solving problems using magnets to exploring the design of spaceships. Figure 1(a)(iii) illustrates an example. The dataset is constructed based on the engineering portion of IXL. The skills and questions are ranging from third grade to eighth grade. To our knowledge, this subset is considered an early exploration on testing multimodal practical knowledge in engineering.

**Mathematics**  Mathematics often requires reasoning and abstract knowledge. For example, solving math tests algebra generalization abilities. For example, the addition of numbers obeys the same rules everywhere. This subset includes fundamental math skills such as addition, algebra, comparison, counting, geometry and spatial reasoning. An example is shown in Figure 1(a)(iv). The questions are from IXL Math spanning from pre-K to eighth grade. To encode mathematical expressions, we use LaTeX to avoid unusual symbols or cumbersome formal languages.

**Comparison with Existing Datasets**  STEM is the first large-scale mulitmodal STEM dataset. As shown in Figure 1(a), STEM provides the largest number of questions and skills among existing STEM related datasets. Compared to the previous largest multimodal STEM datasets, STEM is about 10 times larger in terms of the number of questions. STEM offers the most thorough fundamental skill and question set ranging from pre-K to eighth grade. Compared to datasets of a particular subject, STEM covers all STEM subjects and is at least competitive in terms of the number of questions and skills. For example, STEM's math subset has 27 times more skills compared to the recent math benchmark (Lu et al., 2021b).

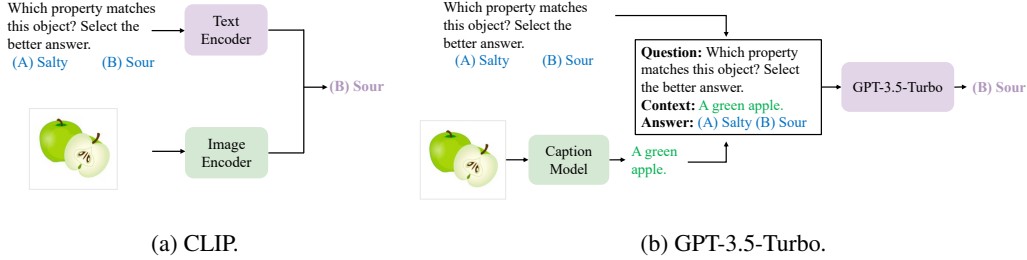

(a) CLIP.        (b) GPT-3.5-Turbo.

Figure 4: Zero-shot model setups.

## 2.2 ANALYSIS

To provide more insights into our dataset, we conduct the below analysis with a focus on the unique perspectives of STEM including skills and grades. Other dataset details such as question analysis are shown in the appendix.

**Skills** The design of STEM emphasizes diverse skills spanning all STEM subjects. Figure 2 presents a brief summary of the skills (a complete skill set is included in the appendix). STEM contains the largest skill set among existing datasets (Figure 1(a)). Each skill contains 2, 395 questions on average. A large number of new skills are introduced to STEM that are not yet covered by existing datasets, e.g., skills in technology and engineering. Besides, understanding multimodal information is crucial to these skills. For example, solving the geometry problem in Figure 1(a)(iv) is challenging since both the image and text contribute to the problem solving. Through this design, STEM helps to recognize important shortcomings of machine learning models by referring to difficult skills for these models.

**Grades** STEM is designed with a comprehensive K-12 curriculum to examine fundamentals of STEM. This leads to another unique feature of testing on STEM: we are able to obtain the grade-level performance of models. The majority of existing datasets aim to compare models with human experts e.g., solving competition-level questions (Hendrycks et al., 2021b; Zheng et al., 2022). However, thanks to the grade-level information provided by STEM we find that models are only competitive with first graders in understanding certain STEM skills. Figure 3 shows the total number of skills per grade of all subjects.

## 2.3 MODELS

We benchmark both state-of-the-art and foundation models on STEM including: multimodal (vision-language) models such as CLIP and language models such as GPT-3.5-Turbo.

**Vision-Language Models**

(i) **Zero-Shot.** We use CLIP (Radford et al., 2021), ViLBERT (Lu et al., 2019), 12-in-1 (Lu et al., 2020), UNITER (Chen et al., 2020b), and Virtex (Desai & Johnson, 2021) for the zero-shot evaluation of multimodal models. Multimodal models generally include two modules: an image encoder and a text encoder. CLIP is considered one of the state-of-the-art multimodal models. For zero-shot CLIP, we follow its original setup in Radford et al. (2021). Figure 4(a) illustrates an example. Other models follow the same zero-shot setup.

(ii) **Finetuning.** To test the learning ability of the models, we also finetune CLIP. We follow the linear probe setup presented in Radford et al. (2021). For each subject, we train the model on its entire training set as shown in Table 1 and select the best model on the validation set. At test time, the evaluation is the same as the zero-shot setup.

**Language Models**

(i) **Zero-Shot.** We use GloVe (Pennington et al., 2014), UnifiedQA (Khashabi et al., 2020), GPT-3 (Chen et al., 2020a) and GPT-3.5-Turbo (Ouyang et al., 2022) zero-shot for the language model evaluation. We formalize the task as a question answering task. We use the OpenAI API "text-davinci-002" and "gpt-3.5-turbo" corresponding to the best-performing GPT-3 and GPT-3.5-Turbo

Table 2: Results on STEM dataset. All evaluation scores are higher the better.

| Model | | Science | Technology | Engineering | Math | Average |
|---|---|---|---|---|---|---|
| Random Guesses | | 38.6 | 25.0 | 44.9 | 39.1 | 36.9 |
| **Language Models** | | | | | | |
| GloVe (Pennington et al., 2014) | | 38.0 | 25.2 | 48.1 | 39.0 | 37.6 |
| UnifiedQA$_{Small}$ (Khashabi et al., 2020) | | 39.6 | 27.2 | 58.0 | 39.6 | 41.1 |
| UnifiedQA$_{Base}$ (Khashabi et al., 2020) | | 42.6 | 28.8 | 55.4 | 40.0 | 41.7 |
| GPT-3 (Brown et al., 2020) | | 47.1 | 22.1 | 73.5 | 44.0 | 46.7 |
| GPT-3.5-Turbo | | 50.1 | 26.3 | 74.6 | 45.0 | 49.0 |
| **Vision-Language Models** | | | | | | |
| Virtex (Desai & Johnson, 2021) | | 37.5 | 24.0 | 48.1 | 38.9 | 37.1 |
| 12-in-1 (Lu et al., 2020) | | 39.4 | 27.5 | 44.2 | 41.9 | 38.3 |
| ViLBERT (Lu et al., 2019) | | 39.0 | 32.1 | 44.2 | 42.7 | 39.5 |
| UNITER (Chen et al., 2020b) | | 50.8 | 34.6 | 55.1 | 43.2 | 45.9 |
| | RN50 | 47.8 | 64.4 | 55.8 | 43.6 | 52.9 |
| | RN101 | 50.3 | 65.3 | 46.7 | 43.7 | 51.5 |
| | RN50x4 | 48.8 | 69.2 | 49.4 | 44.1 | 52.9 |
| | RN50x16 | 49.8 | 66.1 | 51.4 | 44.3 | 52.9 |
| CLIP | RN50x64 | 50.9 | 70.0 | 55.5 | 43.2 | 54.9 |
| (Radford et al., 2021) | ViT-B/32 | 48.3 | 63.7 | 59.5 | 42.8 | 53.6 |
| | ViT-B/16 | 48.6 | 65.9 | 47.2 | 43.6 | 51.3 |
| | ViT-L/14 | 49.8 | 68.6 | 54.3 | 43.1 | 54.0 |
| | ViT-L/14@336px | 50.3 | 68.7 | 55.1 | 43.6 | 54.4 |
| | +Finetuning | 87.0 | 71.9 | 67.7 | 78.4 | 76.3 |

respectively. We convert images to visual context text based on a captioning model following Lu et al. (2022). Figure 4(b) shows an example. All language models follow the same setup.

## 2.4 METRICS AND HUMAN PERFORMANCE

We report accuracy on the test set of each subject. We use accuracy as the evaluation metric since all questions in our dataset are multiple-choice questions. We also compute macro average accuracy across the test sets of all subjects. Unlike the micro evaluation setting, this score relieves data or class imbalance issues. In addition, we focus on two kinds of evaluations for human performance comparison purposes. (i) Exam score. In particular, for science, engineering, and math, we use the IXL SmartScore (Learning, 2019). Different from accuracy, SmartScore considers the progress of learning and is designed to measure how well humans understand a STEM skill (Bashkov et al., 2021). It starts at 0, increases as students answer questions correctly, and decreases if questions are answered incorrectly. We simulate the conditions of its real online exams. The final score is graded by IXL's SmartScore system. According to IXL (IXL, b;a), a score higher than 90.0 is considered excellent for a mastered skill. Therefore, we use this score as a reference to human performance. For technology, we use the average human accuracy available at Triviaplaza. The average accuracy is 68.6. (ii) Accuracy. We sampled 80 questions from our test sets (20 questions for each subject) and collected the responses from seven university students. They attained an average accuracy of 83.0 on all subjects. All evaluation scores are higher the better.

## 3 EXPERIMENTS

In this section, we show the performance of a wide set of neural models as well as humans on STEM. The results show that state-of-the-art foundation models like CLIP and GPT-3.5-Turbo still underperform general elementary students. The details of the experimental setup, additional results and analysis are described in the appendix.

### 3.1 MAIN RESULTS

**Zero-Shot** The results are shown in Table 2. We first test language models to see whether models that only understand text are proficient at the multimodal skills in STEM. GloVe has near random-chance accuracy. This means that STEM cannot be solved by simply matching the text semantic similarity between questions and answers. UnifiedQA does slightly better than GloVe with an improvement of averaging 4.1% points. GPT-3.5-Turbo performs the best among these language models, reaching 49.0% accuracy on average. Both foundation models (GPT-3.5-Turbo and GPT-3) perform well in engineering. This is mainly because engineering practices are mainly described in the

text (see Figure 1(a)(iii)). Recent advancements in large language models help dramatically improve text understanding capabilities. However, large language models still struggle in other subjects. This implies that the understanding of both vision and language information is essential to STEM skills.

Next, we examine vision-language models. We find that the performance of Virtex, 12-in-1, and ViLBERT is nearing the performance of random guesses. These models capture very limited knowledge of STEM subjects. On the other hand, UNITER and CLIP show significant improvements over the random-chance accuracy. Specifically, CLIP-RN50x64 achieves the best result on STEM. It achieves 18.0% points improvements over random guesses. Notably, CLIP-RN50x64 outperforms GPT-3.5-Turbo by 5.9% points. This shows that CLIP has a basic understanding of multimodal STEM skills. Its vision understanding ability certainly contributes to this performance. Among all subjects, we see only marginal improvements in math. This applies to all foundation models. In addition, the result implies that math is the most challenging subject for current neural models. Novel algorithm advancements that can enable strong reasoning ability are necessary to solve math problems.

**Finetuning** The results are shown in Table 2. It is encouraging as finetuning CLIP ViT-L/14@336px is able to significantly boost the performance on science and math by averaging 30% points over its zero-shot setting. The performance improvements on other subjects are 7.9% points, which is much smaller. While having a large amount of training data helps to some extent, the finetuning performance is still far behind that of an average elementary student (the human-level performance is presented in Sec. 3.3). This indicates that more fundamental advancements are required to solve STEM questions in the STEM dataset. For simplicity, we use CLIP to represent CLIP ViT-L/14@336px in the rest of this section.

## 3.2 RESULTS ANALYSIS

**Skills** As STEM provides massive skills, analyzing models' performance at the skill level helps understand models better. We show the performance of foundation models

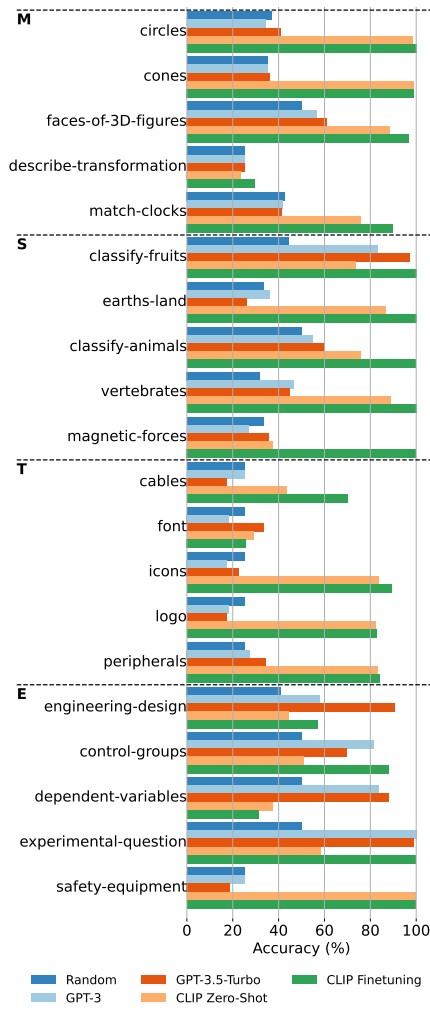

Figure 5: Results categorized by sampled skills of each subject. M: math. S: science. T: technology. E: engineering. Full results are in the appendix.

(GPT-3, GPT-3.5-Turbo, and CLIP) on an uncurated set of skills of each subject in Figure 5. We find that these foundation models are able to perform well zero-shot on skills focusing on identifying common objects (e.g., classifying fruits). However, zero-shot and finetuned foundation models all fail in challenging skills that require abstract knowledge and complex reasoning (e.g., describing transformation).

**Grades** Intuitively, questions for higher graders are more difficult than those for lower graders. We illustrate the grade-level model performance to investigate if the same trend holds for neural models as well. We show the exam scores of models along each grade in Figure 6. Surprisingly, there is no obvious performance drop as the increase in grade levels. This implies the learning curve for neural models may be different from that of humans. A reason is that neural models are trained on data including all grade-level questions simultaneously while humans

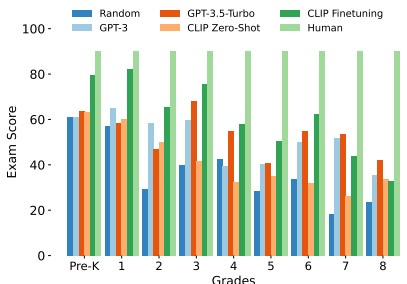

Figure 6: Average grade-level exam scores.

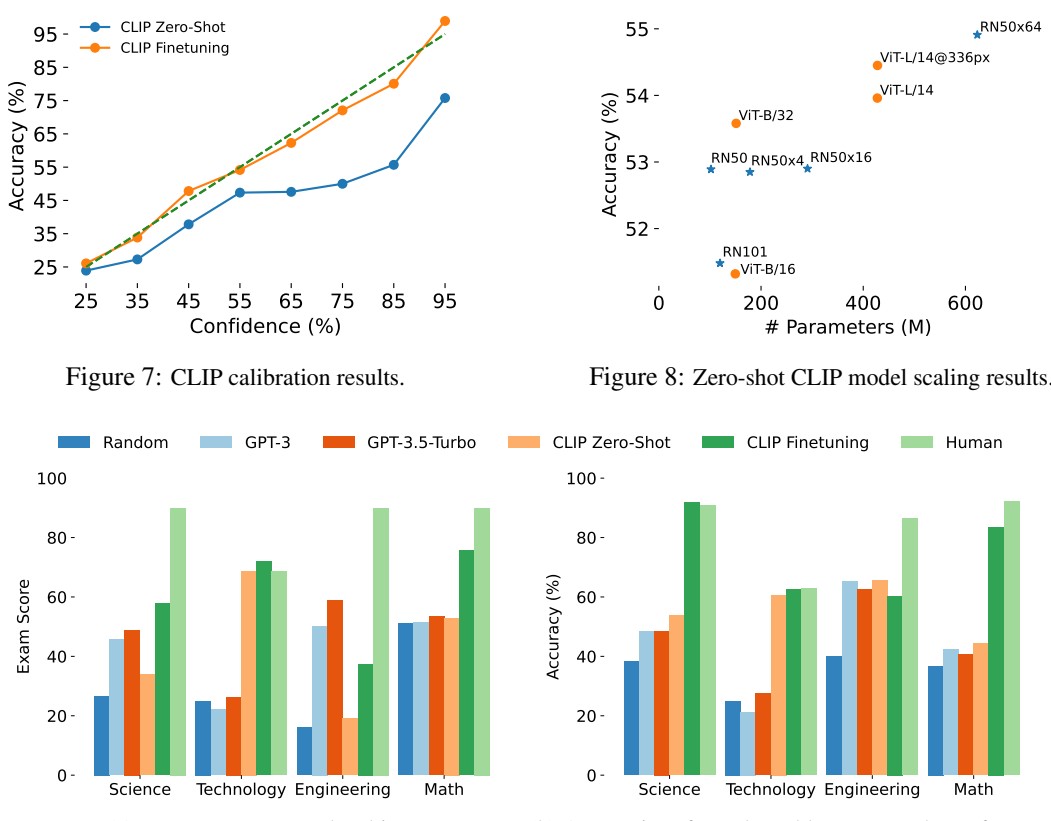

Figure 7: CLIP calibration results.

Figure 8: Zero-shot CLIP model scaling results.

(a) Exam scores on each subject.

(b) Accuracies of a real-world test on a subset of STEM.

Figure 9: Comparison between models and humans.

gradually learn from lower to higher grade-level questions. Also, the average exam scores of elementary grades (grades 1-6) equals 40.8, which is 54.7% lower than human reference (i.e., 90).

**Calibration** A trustworthy model should be calibrated. This means that its confidence should approximately match the actual probability of the prediction being correct (Guo et al., 2017a). However modern neural networks are often not well calibrated (Nguyen et al., 2015; Guo et al., 2017b). We show the relationship between the confidence of CLIP and the corresponding accuracy in Figure 7. We use the softmax probability as the confidence. We observe that the zero-shot CLIP model is not well calibrated. In fact, it is overconfident about its predictions and is only loosely related to its actual accuracy. After finetuning, CLIP is more calibrated. The results suggest that further improving calibration on STEM is another promising direction.

**Scaling Laws** Figure 8 shows the average accuracy of zero-shot CLIP with different model sizes. As expected, the performance improves as models grow larger. But the performance also saturates. This implies that other than increasing model scales, new advancements in model design or training schema are required to improve the performance on STEM.

### 3.3 COMPARISON WITH HUMAN

In this section, we explore whether the best-performing foundation models namely CLIP, GPT-3, and GPT-3.5-Turbo are nearing human-level performance.

Figure 9(a) shows the exam scores (Sec. 2.4) of models and humans on each subject. A score of 90 means a student is proficient in the subject. The zero-shot performances of all tested neural models are well below that bar. In technology, CLIP finetuning achieves human-level performance. This is mainly because most technology skills are about specific empirical knowledge, which is learnable for neural models after finetuning. Overall, there is still a large performance gap between general neural models and average elementary students even in understanding the fundamental skills in STEM. In

addition, the offline real-world test-takers (Sec. 2.4) produce similar outputs with the above online setup on a subset of questions in the STEM. The results are shown in Figure 9(b).

## 3.4 CASE STUDY

We show examples of GPT-3.5-Turbo predictions in Figure 10. We show an example of correct and incorrect predictions respectively. For the correct ones, the corresponding skills are mainly about the basics, such as names of objects (e.g., shapes or animals). The incorrect predictions are mainly due to the complex nature of skills. These skills are often about abstract concepts such as symmetry and the direction of force. They are also more relevant to logical reasoning, such as finding patterns or inferring the function of animal adaption.

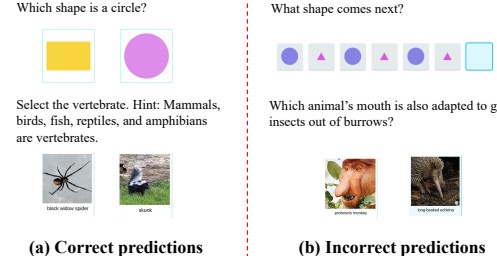

(a) Correct predictions      (b) Incorrect predictions

Figure 10: Examples of GPT-3.5-Turbo predictions.

## 4 RELATED WORK

There are various types of vision-language tasks, such as reference resolution (Kazemzadeh et al., 2014), image captioning or tagging (Thomee et al., 2016; Sharma et al., 2018), image-text retrieval (Lin et al., 2014; Plummer et al., 2015), visual question answering (Antol et al., 2015; Goyal et al., 2017; Zhang et al., 2016; Zhu et al., 2016), and visual reasoning (Suhr et al., 2017; Johnson et al., 2017). Our STEM differs from the previous datasets in that it covers diverse fundamentals of STEM and requires both multimodal understanding and domain knowledge in STEM. This makes STEM a natural testbed to evaluate the real-world problem solving abilities of machine learning models.

Existing STEM related benchmarks do not cover all STEM skills for multimodal understanding. There are benchmarks targeting math (Saxton et al., 2019; Hendrycks et al., 2021b; Zheng et al., 2022; Lu et al., 2021a;b; Xiong et al., 2023b). PIQA (Bisk et al., 2020) is a benchmark for physical commonsense understanding. ScienceQA (Lu et al., 2022) is a multimodal dataset for general science. MMLU (Hendrycks et al., 2021a) contains 57 tasks including STEM but is only restricted to single text modality. Our STEM is the first to include all STEM subjects for vision-language understanding.

Pretrained foundation models help achieve state-of-the-art performance in both NLP and computer vision tasks. Pretrained language models (Radford et al., 2018; 2019; Devlin et al., 2019), especially the recent large language models (Chen et al., 2020a; Wang et al., 2020; 2022a; Ouyang et al., 2022; Crispino et al., 2023; OpenAI, 2023; Chowdhery et al., 2022) have significantly advanced the performance in general natural language understanding tasks. Based on these models, various techniques (Shen et al., 2022a;b; Imani et al., 2023; Jiang et al., 2023; Wang et al., 2023; Xiong et al., 2023a; Pan et al., 2024b;a) have been developed to address specific challenges in a domain such as math. We focus on testing the basic STEM ability of state-of-the-art models in a zero-shot setting and identifying room for improvement by referring to our finetuning results. CLIP (Radford et al., 2021) is one of the state-of-the-art pretrained vision-language models (Lu et al., 2019; Krishna et al., 2017; Chen et al., 2020b; Desai & Johnson, 2021; Lu et al., 2020). Other similar models include GLIP (Li et al., 2022b), GLIDE (Nichol et al., 2022), OFA (Wang et al., 2022b), and BLIP (Li et al., 2022a; 2023). We use CLIP in our test while the majority of existing benchmarks have not explored it yet.

## 5 CONCLUSION

We introduce STEM, a new challenge to examine the STEM skills of neural models. STEM is the largest multimodal benchmark for this challenge. It consists of a large number of multi-choice questions and skills spanning all STEM subjects. STEM focuses on fundamentals of STEM based on the K-12 curriculum. We also include state-of-the-art foundation models such as GPT-3.5-Turbo and CLIP for evaluations. The benchmark results suggest that current neural model performances are still far behind that of elementary students. STEM poses unique challenges for the research community to develop fundamental algorithmic advancements. We hope our benchmark will foster future research in multimodal understanding.

ETHICS STATEMENT

We hereby acknowledge that all of the co-authors of this work are aware of the provided *ICLR Code of Ethics* and honor the code of conduct. We collected data from several sources, and we cited the data creators. The copyright belongs to the original data owners. The STEM dataset is under the CC BY-NC-SA 4.0 license (Creative Commons Attribution-NonCommercial-ShareAlike 4.0 International) and is used for non-commercial research purposes. The collected data does not contain any personally identifiable information or offensive content. Our dataset is mainly built upon instances from real-world exam data. Therefore it was less likely to contain sensitive data. We evaluate foundation models, for which the risks and potential harms are discussed (Brown et al., 2020; Radford et al., 2021).

ACKNOWLEDGEMENTS

This paper is partially supported by the National Key Research and Development Program of China with Grant No. 2023YFC3341203 as well as the National Natural Science Foundation of China with Grant No.62276002.

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

# A   MORE DETAILS ON STEM

In this section, we provide more details on STEM, including dataset analysis, models, evaluation settings, and dataset collection.

## A.1   ANALYSIS

**Questions and Answers**   STEM contains multi-choice questions (Appendix D provides a question example for each skill). The question contains a textual description with an optional image context. Answer options are in text or in an image. We further analyze the questions from the following aspects. (i) The number of answers. STEM has averaging 2.8 answer options for each question. The distribution is presented in Figure 11. In practice, the more answer options one question has, the more difficult it is. (ii) Question type. We categorize questions based on the first three words of the question text as shown in Figure 12. STEM mostly includes factoid questions that start with words such as "which" and "what". We also show the word cloud of our STEM in Figure 13. We can see the most common words like "shape" and "number". This indicates the questions require joint reasoning of the text and images. (iii) Question distribution. Figure 14 depicts the distribution of question lengths. We can see all subjects generally follow a long-tail distribution, while math distribution is most steep and science distribution is flatter. Heuristically, longer questions are more difficult to solve. Figure 15 shows the number of questions in each grade. While pre-K has more questions, the number of questions in other grades is approximately evenly distributed.

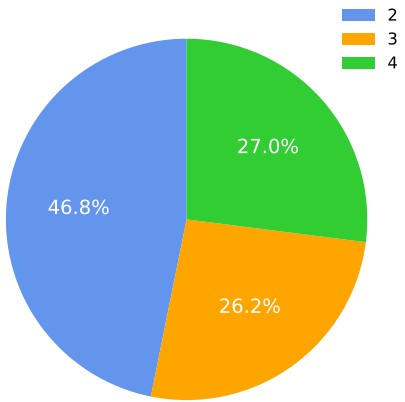

Figure 11: #Answers distribution.

Figure 12: Question type distribution.

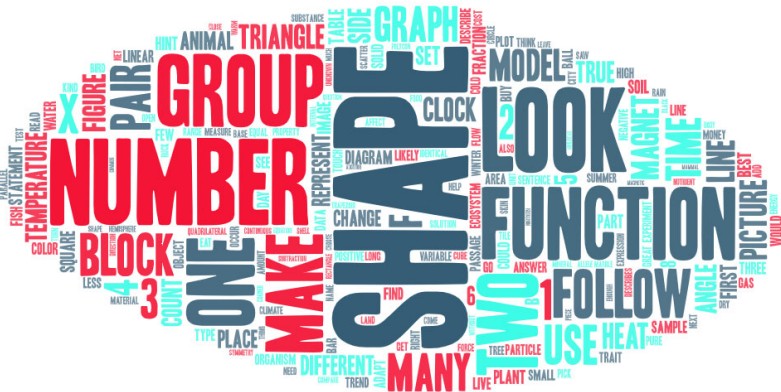

Figure 13: Word cloud of question texts in STEM.

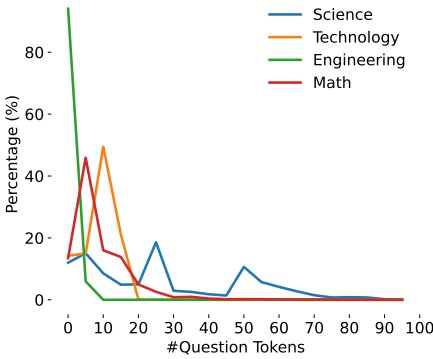

Figure 14: Question length distribution.

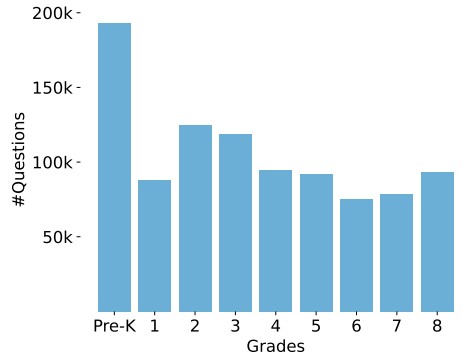

Figure 15: #Questions per grade.

Table 3: Skill comparison between STEM and existing datasets (IconQA and ScienceQA).

(a) Number of skills.

| Subject | IconQA | ScienceQA | STEM |
|---|---|---|---|
| Science | 0 | 167 | 82 |
| Technology | 0 | 0 | 9 |
| Engineering | 0 | 0 | 6 |
| Math | 13 | 0 | 351 |
| Total | 13 | 167 | 448 |

(b) Skill comparison between STEM and IconQA.

| IconQA | | STEM |
|---|---|---|
| Counting | | Count to 10, Count shapes in rows, Count sides and corners . . . |
| Geometry | | Classify triangles, Identify symmetry, Identify shapes . . . |
| Time | | Match times, Identify A.M./P.M., Read a calendar . . . |
| . . . | | . . . |
| Not cover | *Science* | Compare concentrations of solutions . . . |
| | *Technology* | Identify peripherals . . . |
| | *Engineering* | Identify laboratory tools . . . |
| | *Math* | Linear and exponential functions . . . |

**Skill Comparison**   We compare the skills of STEM with other related datasets in Table 3. STEM contains the largest skill set among existing datasets, with a great number of new skills introduced to STEM that are not yet covered by existing datasets, e.g., skills in technology and engineering.

## A.2   MODELS

In this section, we introduce the foundation models we benchmark in detail.

**Vision-Language Models**

**CLIP (Radford et al., 2021).** CLIP is pretrained on a sufficiently large dataset of 400 million text-image pairs across the Internet. It uses a Transformer as the text encoder, and has several variants of image encoder, including ResNet (RN) backbones and Vision Transformers (ViT) (Dosovitskiy et al., 2020). CLIP aligns the text and image representation by training on in-batch contrastive loss, and is able to zero-shot transfer to downstream vision language tasks. To align with CLIP pretraining, we formulate question answering as matching text and images. We use the cosine similarity between the text and image embeddings as the matching function, the same as the original zero-shot image-text retrieval settings in CLIP (Radford et al., 2021).

**ViLBERT and 12-in-1 (Lu et al., 2019; 2020).** ViLBERT adopts two parallel streams to process image regions and text segments separately, with co-attentional transformer layers connecting them. There is also a multi-task version called 12-in-1 (Lu et al., 2020) that trains 12 different tasks with individual task-specific heads sharing 1 "trunk" ViLBERT model. Its multi-modal alignment prediction serves as the matching score.

**UNITER (Chen et al., 2020b).** UNITER consists of an Image Embedder with Faster R-CNN (Anderson et al., 2018), a Text Embedder with Transformer (Vaswani et al., 2017), as well as a multi-layer Transformer to get cross-modality representation. During inference on STEM, the matching

score function is the same as CLIP, i.e., the cosine similarity between the text and image embeddings (Chen et al., 2020b).

**Virtex (Desai & Johnson, 2021).** Virtex first extracts visual features with ResNet-50 (He et al., 2016) backbone. The visual features are then fed into a text head, which consists of two unidirectional Transformers, to predict captions. We extract the image feature with the image encoder, then feed text into the textual head and use the sum of bidirectional generation logits as the matching score.

**Language Models**

**GPT-3 (Chen et al., 2020a) and GPT-3.5-Turbo (Ouyang et al., 2022).** These foundation language models are generation models pretrained on a large corpus of text. We use the OpenAI API "text-davinci-002" and "gpt-3.5-turbo" corresponding to the best-performing GPT-3 and GPT-3.5-Turbo respectively. We formalize the evaluation task as a question-answering task. The input to GPT-3 and GPT-3.5-Turbo is the concatenation of the question text, the context text, and multiple answer options. The output is to predict a final answer from answer options. For images in questions, we follow Lu et al. (2022) to convert them to visual context text based on a captioning model consisting of ViT (Dosovitskiy et al., 2020) and GPT-2 (Radford et al., 2019).

**UnifiedQA (Khashabi et al., 2020).** UnifiedQA is a pretrained question-answering model. We use both its base and small versions. Its evaluation setup is the same as that of GPT-3 and GPT-3.5-Turbo.

**GloVe (Pennington et al., 2014).** GloVe is a pretrained word embedding model. We use the similarity between the average embedding of the concatenation of the question and context and the average embedding of each answer option. The answer option with the largest similarity score is the answer output. We use average pooling based on the 300-dimensional word embeddings. The images are also converted to text using the same method as GPT-3 and GPT-3.5-Turbo.

## A.3 Evaluation Settings

We benchmark state-of-the-art foundation models on STEM under different settings, including zero-shot, few-shot, finetuning, and multi-task.

(i) **Zero-Shot.** We use CLIP (Radford et al., 2021), ViLBERT (Lu et al., 2019), 12-in-1 (Lu et al., 2020), UNITER (Chen et al., 2020b), and Virtex (Desai & Johnson, 2021) for the zero-shot evaluation of foundation multimodal models. CLIP is the state-of-the-art multimodal model. For zero-shot CLIP, we follow its original setup in Radford et al. (2021). The input to the text encoder is the concatenation of the question text and an answer option. The input to the image encoder is the image context. The output is the cosine similarity scores between the text embeddings and image embedding. Then the answer option with the largest similarity score serves as an answer. For questions with image answer options, the input to the image encoder will also add the image answer options.

(ii) **Few-Shot.** We also use CLIP to benchmark the multimodal few-shot results. For $k$-shot setup, we randomly select $k$ questions for each skill from the training set as a meta training set. For each STEM subject, we train the model on the meta training set and select the best model on the validation set. At test time, the evaluation is the same as the zero-shot setup.

(iii) **Finetuning.** We also finetune CLIP on the entire training set for each subject. The remaining setup is the same as the few-shot setting.

(iv) **Multi-Task.** Under this setting, we train CLIP on the mixture of training sets of four subjects to produce a single model for all subjects.

## A.4 Dataset Collection

We collect science, engineering and math problems from *IXL*[2], and technology problems from *ProProfs Quizzes*[3] and *Triviaplaza*[4]. We first collect multi-choice problems that have at least one image in either question context or answers. We collect at most 2,000 problems for each skill and remove duplicated problems. There are many formulas embedded in math problems that are not

---

[2]https://www.ixl.com/

[3]https://www.proprofs.com/quiz-school

[4]https://www.triviaplaza.com/

Table 4: Results of CLIP with different training schemes.

| | Method | Science | Technology | Engineering | Math | Average |
|---|---|---|---|---|---|---|
| CLIP | Zero-Shot | 50.3 | 68.7 | 55.1 | 43.6 | 54.4 |
| | Few-Shot | 75.2 | 70.9 | 61.9 | 63.2 | 67.8 |
| | Finetuning | 87.0 | 71.9 | 67.7 | 78.4 | 76.3 |
| | Multi-Task | 86.3 | 60.4 | 73.4 | 77.7 | 74.5 |

represented in the text. We use the Mathpix[5] OCR API to convert these math formulas into the latex format.

## B  MORE DETAILS ON EXPERIMENTS

### B.1  EXPERIMENTAL SETUP

For the zero-shot setting, we evaluate all models on the test set. For the few-shot, finetuning, and multi-task setting, we train CLIP-ViT-L/14@336px on the corresponding train set, tune hyperparameters on the valid set, and finally evaluate on the test set. We use AdamW for optimization and tune hyperparameters as follows: batch size is chosen from {16, 32, 64, 128}, and set to 16 for few-shot learning, 128 for finetuning and multi-task learning after hyperparameter tuning. The learning rate is chosen between [5e-6, 5e-5] and set to 1e-5 for all training. We set the warm-up ratio to 0.1 and set weight decay as 0.2. We set the maximum of training samples to 100k for finetuning, 200k for multitask training, and 10 epochs for few-shot training, all with early stopping on the valid set. We use NVIDIA GeForce RTX 3090 GPUs for training.

### B.2  DETAILED EXPERIMENTAL ANALYSIS

**Few-Shot**  In the few-shot setting, we sample different number of samples in each grade to see how the learning performance varies. Specifically, we sample 16 samples per skill and train CLIP on the sampled data. The results are shown in Table 4. We observe that CLIP gains much improvement in all subjects after few-shot learning. This implies that CLIP has already stored STEM-related knowledge and a few samples are able to trigger such knowledge. We also show performance varies when the number of samples of each skill changes (Figure 16). The overall performance improves with more samples, but 1-shot and 2-shot in technology are worse than zero-shot. Since there are only 9 skills in technology, 1-shot and 2-shot learning in technology might lead to overfitting.

**Multi-Task**  We show the results in Table 4. Multi-task learning improves in engineering but performs worse in other subjects compared with individual finetuned models. The reason for the great drop in technology is mainly because its data is much less than other subjects. Multi-task training actually improves performance in engineering. This implies that data from one subject may be beneficial for another when the knowledge is transferable. For example, science shares many common topics with engineering like chemical experiments.

**Number of Answers**  We also analyze how model performance changes with the number of answers. The results are shown in Figure 17. We find that for GPT-3, GPT-3.5-Turbo, CLIP zero-shot, and few-shot, the accuracy drops as the number of answers increases, but the accuracy of CLIP finetuning and multi-task does not drop. This implies that models after full training are actually solving the problem rather than guessing, so the number of choices does not affect the performance much.

**Question Lengths**  Figure 18 shows how the question length affects model accuracy. For GPT-3, GPT-3.5-Turbo and CLIP zero-shot, the accuracy decreases slightly as the question becomes longer. For tuned models, the same trend holds for questions less than 70 tokens, but the accuracy starts to increase for longer questions. We think this may be caused by some bias in longer questions and the

---

[5]https://mathpix.com/

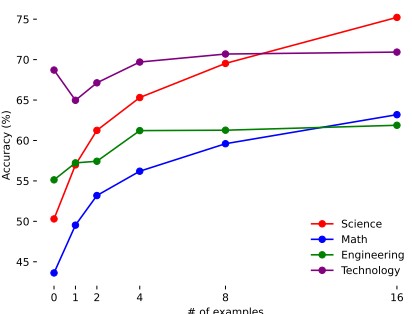

Figure 16: Result of few-shot CLIP.

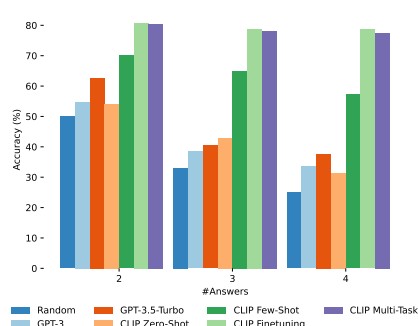

Figure 17: Results on questions with different numbers of answers.

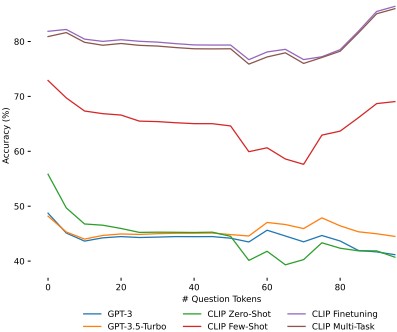

Figure 18: Results on questions with different lengths.

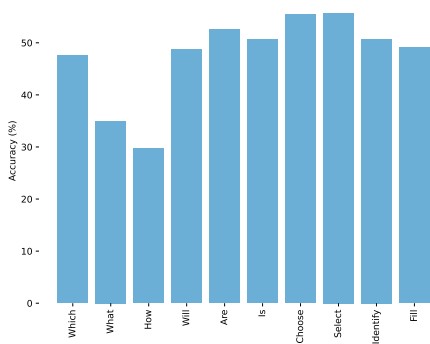

Figure 19: Zero-shot CLIP performance on different question types.

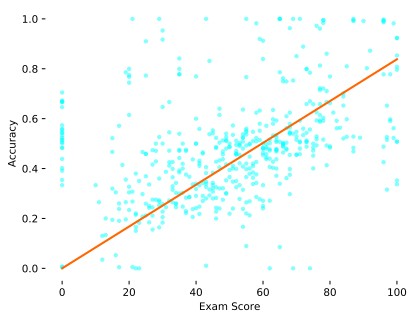

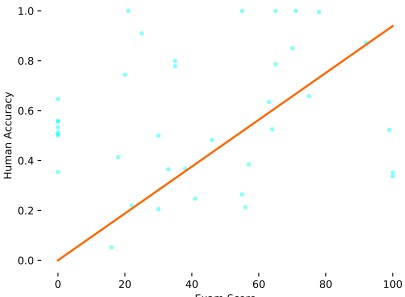

Figure 20: The correlation graphs of exam scores with model accuracy (left) and human accuracy (right).

tuned models learn such bias and achieve higher accuracy. Since there are only a small proportion of questions that are longer than 70 tokens, such bias will not affect the whole dataset much.

**Question Type** We mark the types of problems as the first word in the question or request of each problem. In Figure 19 we show the accuracy of the top 10 frequent types. Questions starting with "What" and "How" have relatively low accuracy, as these questions are more difficult to answer.

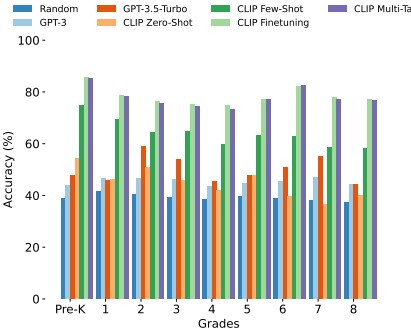

Figure 21: Average accuracies on each grade.

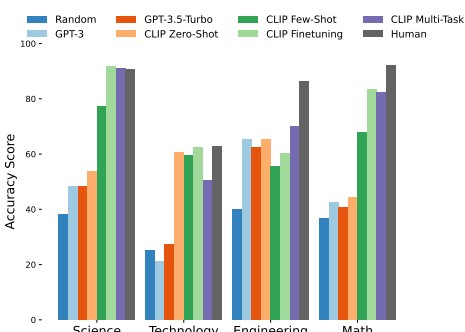

Figure 22: Accuracy on sampled STEM for human performance.

Table 5: Error analysis of CLIP on math and science subsets of STEM.

| Subject | Reason | Ratio (%) |
|---|---|---|
| Math | Commonsense | 36 |
| | Numerical calculation | 24 |
| | Counting | 16 |
| | Read table/graph | 12 |
| | Transformation | 12 |
| Science | Comparison | 40 |
| | Commonsense | 32 |
| | Direction | 20 |
| | Read table/graph | 8 |

**Grades**   We show the model accuracy on each grade in Figure 21. There is no obvious performance drop as the increase in grade levels, which is similar to the trend of exam scores. This implies the learning curve for neural models may be different from that of humans.

**Correlation Between Exam Scores and Accuracy**   We evaluate exam scores' correlation with model accuracy and human accuracy(Figure 20). They in general positively correlated to each other. Even though exam score is different from accuracy, it overall captures accuracy as an important factor.

### B.3   Error Analysis

To better understand the errors made by CLIP zero-shot, we sample 25 error cases of CLIP zero-shot on math and science. We manually check the reasons for these errors. Table 5 shows the analysis results. For math, 36% errors are caused by a lack of mathematical commonsense, such as area formulas and symmetry. Other errors include failure of calculation (24%), counting objects (16%), reading tables or graphs (12%, e.g., graphs of functions), and transformation (12%, e.g., rotation of a 3D object). For science, comparison causes the most errors with a ratio of 40%. Most of these questions only require a straightforward comparison like the distance between two pairs of magnets. However, CLIP fails on such basic problems. This indicates that it is not good at comparing objects and properties yet. Lacking science commonsense also leads to a good number of errors (32%), followed by identifying directions (20%, e.g., the directions of push and pull, towards and away) and reading tables or graphs (8%).

Moreover, we show the top-5 skills with the most errors of fine-tuned models on math and science subsets in Table 6 and Table 7 respectively.

| Skill | Error Rate | Example |
|---|---|---|
| greatest-and-least-word-problems-up-to-100 | 76.8% | Description: The school district compared how many swings each elementary school has. Which school has the fewest swings?
Picture: [table: Swings — School / Number of swings: Shoreline Elementary 20, Hillside Elementary 17, Valley Elementary 10, Lincoln Elementary 14]
Choices: [Shoreline Elementary, Hillside Elementary, Valley Elementary, Lincoln Elementary, ]
Answer index: 2
Prediction: 0 |
| greatest-and-least-word-problems-up-to-1000 | 76.0% | Description: Paul kept a log of how many minutes he spent practicing ice skating over the past 4 days. On which day did Paul practice the least?
Picture: [table: Minutes spent practicing — Day / Minutes: Tuesday 249, Wednesday 234, Thursday 243, Friday 223]
Choices: [Tuesday, Wednesday, Thursday, Friday, ]
Answer index: 3
Prediction: 2 |
| reading-schedules | 75.0% | Description: Look at the following schedule: Which meeting ends at 12:00 P.M.?
Picture: [table: City government schedule — Meeting / Begin / End: fire department meeting 10:50 A.M. 11:40 A.M., city council meeting 11:20 A.M. 12:00 P.M., construction permit meeting 11:50 A.M. 1:00 P.M., police meeting 1:35 P.M. 2:30 P.M., planning department meeting 3:05 P.M. 4:30 P.M., transportation meeting 4:10 P.M. 4:40 P.M., budget meeting 4:55 P.M. 6:15 P.M., parking meter meeting 6:20 P.M. 7:35 P.M.]
Choices: [the city council meeting, the construction permit meeting, the parking meter meeting, the police meeting, ]
Answer index: 0
Prediction: 2 |
| angles-of-90-180-270-and-360-degrees | 73.8% | Description: What fraction of a turn is this angle?
Picture:
Choices: [3/4, 1 full turn, 1/2, 1/4, ]
Answer index: 2
Prediction: 3 |
| points-lines-line-segments-rays-and-angles | 73.8% | Description: What is this?
Picture:
Choices: [a line segment, a ray, a line, a point, ]
Answer index: 1
Prediction: 0 |

Table 6: Error analysis of top-5 skills with most errors on math.

## B.4 COMPARISON WITH HUMAN

**Exam Score**   We test exam scores on all skills in engineering and technology, and randomly choose 40 skills from math, and 30 skills from science due to technical and time constraints. We compare neural models with humans using the exam score, and the results are shown in Table 8. The detailed scores and skills are listed in Table 10.

**Accuracy**   We randomly sample 20 problems for each subject and ask 7 Ph.D. students to answer these questions, and calculate the average accuracy for each subject. To evaluate neural models on these questions, we use the corresponding skill accuracy for each sampled problem as the models' score on this problem and average all accuracy together as the final score. We do not evaluate models on these sampled data directly since the small number of samples will lead to a large variance, and skill accuracy can avoid such variance. The comparison results are shown in Table 8 and Figure 22. All sampled problems are listed in Table 12 to 17.

## B.5 ZERO-SHOT PROMPT SENSITIVITY

We study the effect of prompts on CLIP zero-shot. We design 5 types of prompts and demonstrate them with an example problem. The example question is "Which property matches this object?" and the answer is "Rough". Examples of different prompt types and the corresponding accuracies are shown in Table 9. We observe that "Q+A results in the best performance on average, but the difference is only marginal, meaning that CLIP zero-shot is not very sensitive to the format of prompts.

## B.6 DETAILED PERFORMANCE ON SKILLS

We show the accuracy of neural models on all 448 skills in Figure 23 to 28. We can see that the zero-shot performance is generally better than random guesses on most skills and achieves near 100% on some skills (e.g., "circles" and "cones"). After finetuning, accuracy improves on most skills and becomes near 100% on many skills.

## B.7 VQA RESULTS

| Skill | Error Rate | Example |
|---|---|---|
| use-punnett-squares-to-calculate-ratios-of-offspring-types | 69.10% | Description: This passage describes the antenna type trait in fruit flies. Most fruit flies have a pair of antennae on their head. But, some flies appear to have an extra pair of legs on their head instead! These flies have a mutation, or change, in a gene that affects body development. This mutation makes the cells in the fly's head form mutated antennae that are like legs. In a group of fruit flies, some individuals have mutated antennae and others have normal antennae. In this group, the gene for the antenna type trait has two alleles. The allele for normal antennae (a) is recessive to the allele for mutated antennae (A). This Punnett square shows a cross between two fruit flies. What is the expected ratio of offspring with normal antennae to offspring with mutated antennae? Choose the most likely ratio.  Picture: Choices: [0:4, 3:1, 2:2, 1:3, 4:0, ] Answer index: 0 Prediction: 3 |
| use-punnett-squares-to-calculate-probabilities-of-offspring-types | 60.10% | Description: In a group of tomato plants, some individuals have smooth fruit and others have fuzzy fruit. In this group, the gene for the fruit texture trait has two alleles. The allele for smooth fruit (F) is dominant over the allele for fuzzy fruit (f). This Punnett square shows a cross between two tomato plants. What is the probability that a tomato plant produced by this cross will be homozygous recessive for the fruit texture gene?  Picture: Choices: [0/4, 1/4, 2/4, 3/4, 4/4, ] Answer index: 0 Prediction: 3 |
| predict-temperature-changes | 55.00% | Description: Two identical blocks are heated to different temperatures. The blocks are placed so that they touch, and heat begins to flow between the blocks. The pair of blocks is insulated, so no energy escapes. Later, the temperature of each block is measured again. Which pair of temperatures is possible?  Picture: Choices: Answer index: 1 Prediction: 0 |
| identify-magnets-that-attract-or-repel | 21.10% | Description: Two magnets are placed as shown. Hint: Magnets that attract pull together. Magnets that repel push apart.  Picture: Choices: [attract, repel, ] Answer index: 1 Prediction: 0 |
| predict-heat-flow | 16.20% | Description: Two solid blocks are at different temperatures. The blocks are touching. Which picture shows how heat will move? Picture: None  Choices: Answer index: 0 Prediction: 1 |

Table 7: Error analysis of top-5 skills with most errors on science.

Table 8: Comparison between models and humans.

| Method | | Exam Score | | | | Accuracy | | | |
|---|---|---|---|---|---|---|---|---|---|
| | | Science | Engineering | Math | Technology | Science | Technology | Engineering | Math |
| Human | | 90.0 | 90.0 | 90.0 | 68.6 | 90.7 | 62.9 | 86.4 | 92.1 |
| Random | | 26.7 | 16.1 | 51.1 | 25.0 | 38.3 | 25.0 | 40.0 | 36.8 |
| GPT-3 | | 45.7 | 50.2 | 51.4 | 22.1 | 48.4 | 21.3 | 65.2 | 42.4 |
| GPT-3.5-Turbo | | 48.9 | 58.7 | 53.5 | 26.3 | 48.5 | 27.4 | 62.5 | 40.6 |
| CLIP | Zero-Shot | 33.9 | 19.0 | 52.9 | 68.7 | 53.8 | 60.7 | 65.5 | 44.3 |
| | Few-Shot | 39.1 | 43.9 | 67.6 | 70.9 | 77.3 | 59.7 | 55.5 | 67.8 |
| | Finetuning | 57.8 | 37.4 | 75.7 | 71.9 | 91.9 | 62.6 | 60.3 | 83.5 |
| | Multi-Task | 61.9 | 50.3 | 72.0 | 60.4 | 90.9 | 50.6 | 70.2 | 82.5 |

Table 9: Examples for different prompts and their zero-shot accuracy.

| Prompt Format | Example | Science | Technology | Engineering | Math | Average |
|---|---|---|---|---|---|---|
| Q+A | Which property matches this object? Rough. | 50.3 | 68.7 | 55.1 | 43.6 | 54.4 |
| A+Q | Rough. Which property matches this object? | 50.0 | 66.0 | 49.6 | 43.2 | 52.2 |
| Q "Choose the best answer:" A | Which property matches this object? **Choose the best answer:** Rough. | 50.1 | 70.7 | 49.7 | 44.2 | 53.7 |
| "Answer the question:" Q + A | **Answer the question:** Which property matches this object? Rough. | 49.4 | 67.6 | 51.0 | 43.6 | 52.9 |
| A "best answers the question" Q | Rough **best answers the question:** Which property matches this object? | 49.7 | 69.5 | 50.8 | 43.8 | 53.4 |

We evaluate the zero-shot CLIP model and models finetuned on each subject on the VQA (Antol et al., 2015) dataset. Results are shown in Table 11. The average increase of the finetuned models over the zero-shot setting is 1.2%.

| Model | Accuracy |
|---|---|
| Zero-Shot CLIP | 24.7% |
| Finetuning with Science | 27.3% |
| Finetuning with Technology | 26.5% |
| Finetuning with Engineering | 24.8% |
| Finetuning with Math | 24.9% |

Table 11: Results on the VQA (Antol et al., 2015) dataset.

## C  ADDITIONAL RELATED WORK

In addition to vision-language foundation models included in the main text, we expand the discussion to some recent models, including BLIP-2 (Li et al., 2023), EVA-ClIP (Sun et al., 2023), and KOSMOS-2 (Peng et al., 2023). BLIP-2 provides a versatile and efficient strategy for pre-training. This strategy enhances the vision-language pre-training process by utilizing frozen pre-trained image encoders and frozen large language models, while EVA-CLIP proposes a series of methods to increase the training efficiency of the CLIP model. KOSMOS-2 enables new capabilities for perceiving object descriptions. This work focuses on the creation of a dataset to evaluate the multimodal STEM understanding and we chose the foundation models like CLIP for a pilot study on our dataset. There are more benchmarks targeting formal math reasoning (Zheng et al., 2022; Liu et al., 2023; Xiong et al., 2023b), however, they are all restricted to single text modality and they can not evaluate fundamental skills.

## D  SUMMARY OF SKILLS

We list all skills in STEM in Table 18 to 20 and show some examples in Table 21 to 27.

| Subject | Grade/Skill | Random | Zero-shot | Finetune |
|---------|-------------|--------|-----------|----------|
| Science | grade-2/classify-matter-as-solid-liquid-or-gas | 28 | 40 | 100 |
| | grade-2/identify-animals-with-and-without-backbones | 0 | 70 | 70 |
| | grade-2/identify-mammals-birds-fish-reptiles-and-amphibians | 0 | 0 | 18 |
| | grade-2/identify-materials-in-objects | 21 | 40 | 100 |
| | grade-2/identify-properties-of-an-object | 35 | 65 | 65 |
| | grade-3/compare-strengths-of-magnetic-forces | 0 | 18 | 63 |
| | grade-3/describe-ecosystems | 65 | 50 | 100 |
| | grade-3/find-evidence-of-changes-to-earths-surface | 17 | 38 | 100 |
| | grade-3/identify-ecosystems | 35 | 100 | 100 |
| | grade-3/identify-minerals-using-properties | 35 | 11 | 35 |
| | grade-4/compare-properties-of-objects | 10 | 17 | 20 |
| | grade-4/describe-ecosystems | 74 | 100 | 100 |
| | grade-4/identify-minerals-using-properties | 35 | 16 | 35 |
| | grade-4/use-evidence-to-classify-mammals-birds-fish-reptiles-and-amphibians | 26 | 35 | 35 |
| | grade-5/animal-adaptations-beaks-mouths-and-necks | 17 | 27 | 35 |
| | grade-5/classify-elementary-substances-and-compounds-using-models | 75 | 75 | 75 |
| | grade-5/compare-ancient-and-modern-organisms-use-observations-to-support-a-hypothesis | 32 | 32 | 50 |
| | grade-5/identify-directions-of-forces | 0 | 26 | 35 |
| | grade-5/identify-the-photosynthetic-organism | 0 | 0 | 100 |
| | grade-5/predict-temperature-changes | 0 | 22 | 0 |
| | grade-5/use-evidence-to-classify-animals | 35 | 35 | 35 |
| | grade-5/use-evidence-to-classify-mammals-birds-fish-reptiles-and-amphibians | 18 | 35 | 35 |
| | grade-5/weather-and-climate-around-the-world | 60 | 36 | 60 |
| | grade-6/compare-concentrations-of-solutions | 15 | 11 | 100 |
| | grade-6/describe-the-effects-of-gene-mutations-on-organisms | 52 | 13 | 69 |
| | grade-6/diffusion-across-membranes | 50 | 25 | 50 |
| | grade-7/describe-the-effects-of-gene-mutations-on-organisms | 42 | 13 | 69 |
| | grade-8/classify-symbiotic-relationships | 25 | 36 | 45 |
| | grade-8/diffusion-across-membranes | 0 | 18 | 35 |
| | grade-8/moss-and-fern-life-cycles | 0 | 12 | 0 |
| Engineer | grade-6/evaluate-tests-of-engineering-design-solutions | 0 | 0 | 100 |
| | grade-6/identify-control-and-experimental-groups | 0 | 0 | 0 |
| | grade-6/identify-independent-and-dependent-variables | 0 | 0 | 100 |
| | grade-6/identify-the-experimental-question | 30 | 30 | 30 |
| | grade-7/evaluate-tests-of-engineering-design-solutions | 0 | 0 | 0 |
| | grade-7/identify-control-and-experimental-groups | 0 | 0 | 40 |
| | grade-7/identify-independent-and-dependent-variables | 0 | 0 | 30 |
| | grade-7/identify-the-experimental-question | 40 | 0 | 40 |
| | grade-8/identify-control-and-experimental-groups | 0 | 0 | 0 |
| | grade-8/identify-the-experimental-question | 60 | 0 | 40 |
| | grade-5/identify-laboratory-tools | 21 | 42 | 31 |
| | grade-6/identify-laboratory-tools | 21 | 21 | 21 |
| | grade-6/laboratory-safety-equipment | 24 | 65 | 52 |
| | grade-7/identify-laboratory-tools | 10 | 28 | 21 |
| | grade-7/laboratory-safety-equipment | 9 | 58 | 52 |
| | grade-8/identify-laboratory-tools | 49 | 21 | 21 |
| | grade-8/laboratory-safety-equipment | 9 | 58 | 58 |
| Math | algebra-2/factor-quadratics-using-algebra-tiles | 40 | 51 | 55 |
| | algebra-2/outliers-in-scatter-plots | 55 | 47 | 97 |
| | calculus/determine-continuity-using-graphs | 36 | 63 | 80 |
| | calculus/find-limits-at-vertical-asymptotes-using-graphs | 60 | 65 | 85 |
| | grade-1/subtraction-sentences-up-to-10-which-model-matches | 50 | 30 | 99 |
| | grade-2/identify-halves-thirds-and-fourths | 65 | 75 | 97 |
| | grade-2/identify-lines-of-symmetry | 70 | 64 | 99 |
| | grade-2/interpret-bar-graphs-ii | 14 | 23 | 12 |
| | grade-2/ordinal-numbers-up-to-10th | 32 | 61 | 28 |
| | grade-3/compare-fractions-in-recipes | 55 | 50 | 68 |
| | grade-3/identify-parallelograms | 51 | 64 | 70 |
| | grade-3/is-it-a-polygon | 71 | 60 | 98 |
| | grade-3/parallel-sides-in-quadrilaterals | 29 | 66 | 45 |
| | grade-4/nets-of-three-dimensional-figures | 68 | 40 | 99 |
| | grade-5/nets-of-three-dimensional-figures | 53 | 40 | 99 |
| | grade-6/changes-in-mean-median-mode-and-range | 38 | 14 | 15 |
| | grade-6/classify-triangles | 47 | 38 | 45 |
| | grade-6/identify-polyhedra | 75 | 75 | 75 |
| | grade-6/mean-median-mode-and-range-find-the-missing-number | 55 | 41 | 99 |
| | grade-6/model-and-solve-equations-using-algebra-tiles | 36 | 36 | 57 |
| | grade-6/rational-numbers-find-the-sign | 31 | 78 | 99 |
| | grade-6/rotational-symmetry | 62 | 56 | 78 |
| | grade-6/similar-and-congruent-figures | 34 | 33 | 46 |
| | grade-6/which-figure-is-being-described | 36 | 27 | 86 |
| | grade-7/rational-numbers-find-the-sign | 47 | 58 | 99 |
| | grade-8/rotational-symmetry-amount-of-rotation | 47 | 32 | 63 |
| | kindergarten/count-on-ten-frames-up-to-10 | 15 | 2 | 49 |
| | kindergarten/fewer-and-more-up-to-20 | 80 | 62 | 97 |
| | kindergarten/subtraction-sentences-up-to-5-which-model-matches | 41 | 30 | 96 |
| | pre-k/addition-sentences-up-to-10-which-model-matches | 60 | 55 | 96 |
| | pre-k/count-on-ten-frames-up-to-3 | 84 | 50 | 51 |
| | pre-k/fewer-and-more-compare-by-matching | 63 | 52 | 90 |
| | pre-k/one-less-with-pictures-up-to-10 | 61 | 37 | 66 |
| | pre-k/one-more-with-pictures-up-to-5 | 48 | 36 | 75 |
| | pre-k/shapes-of-everyday-objects | 67 | 96 | 96 |
| | pre-k/spheres | 67 | 96 | 96 |
| | pre-k/triangles | 57 | 75 | 75 |
| | pre-k/what-comes-next | 75 | 56 | 70 |
| | pre-k/ordinal-numbers-up-to-tenth | 27 | 84 | 82 |
| | kindergarten/are-there-enough | 40 | 99 | 96 |

Table 10: Exam scores for each skill.

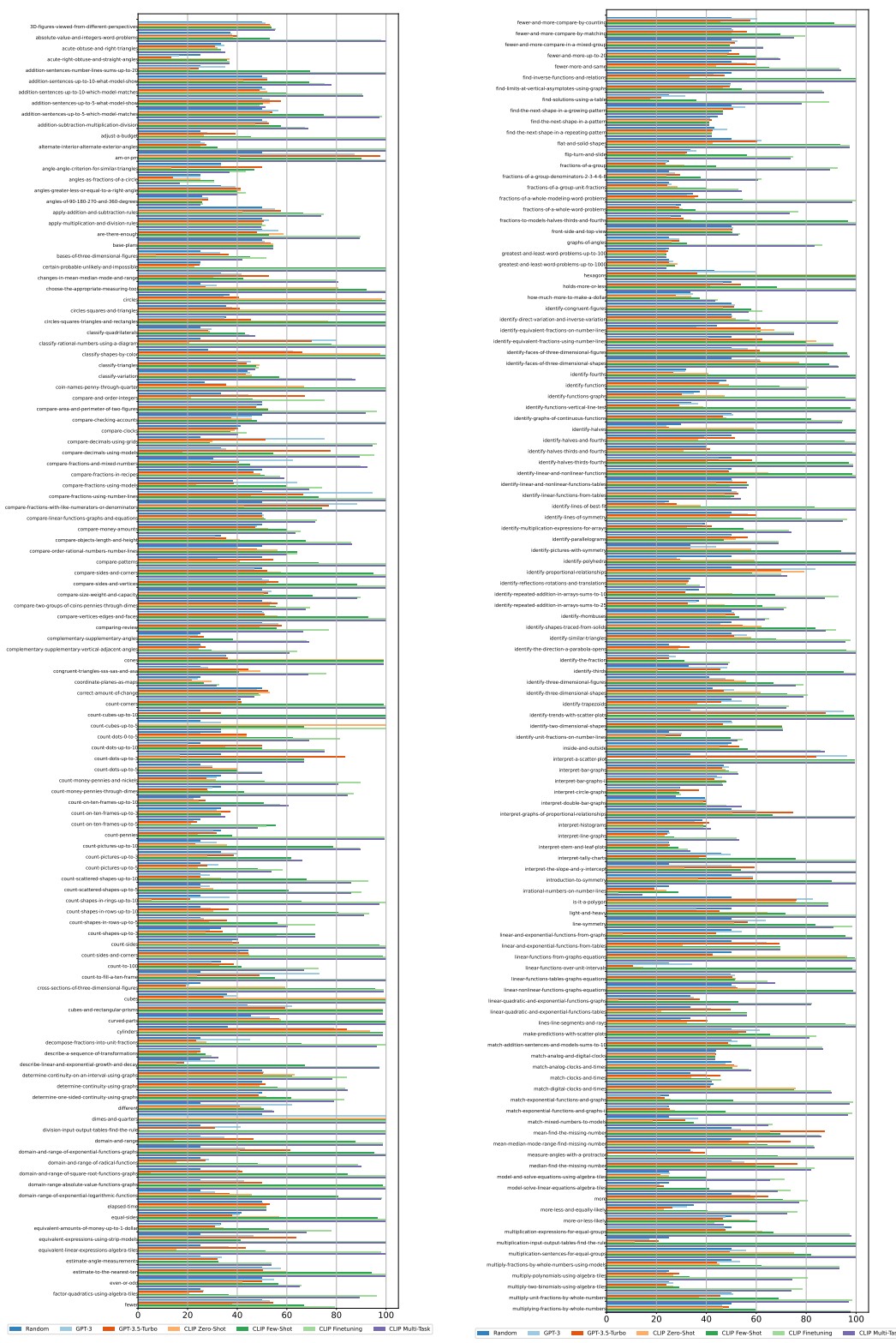

Figure 23: Accuracy per skill on math (part 1).

Figure 24: Accuracy per skill on math (part 2).

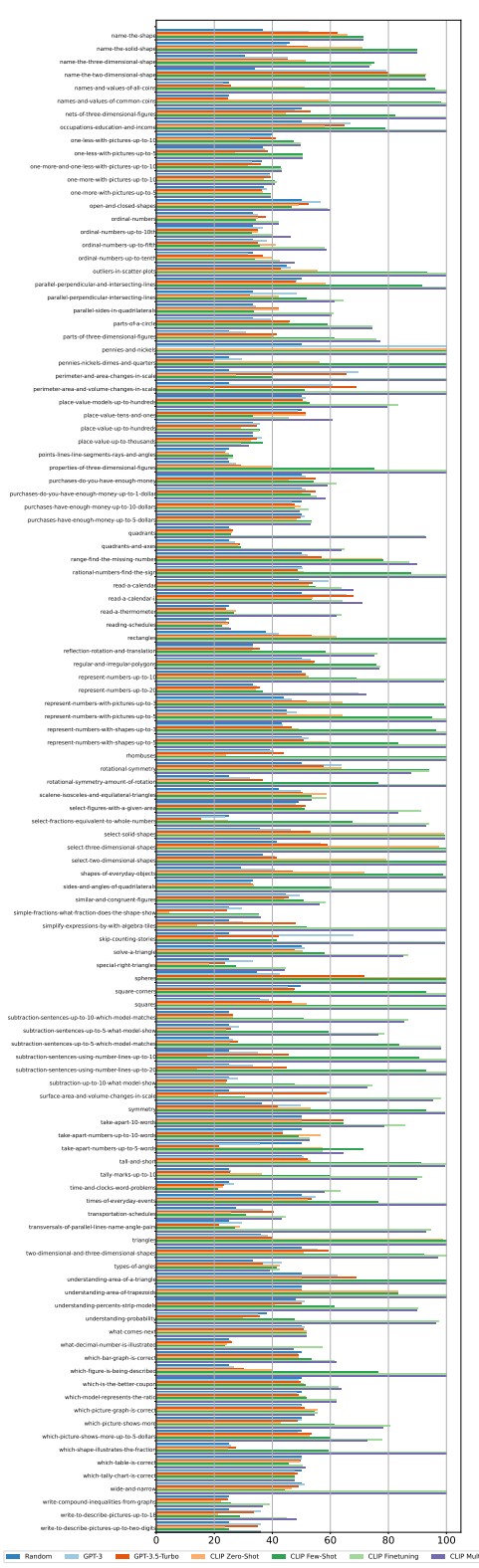

Figure 25: Accuracy per skill on math (part 3).

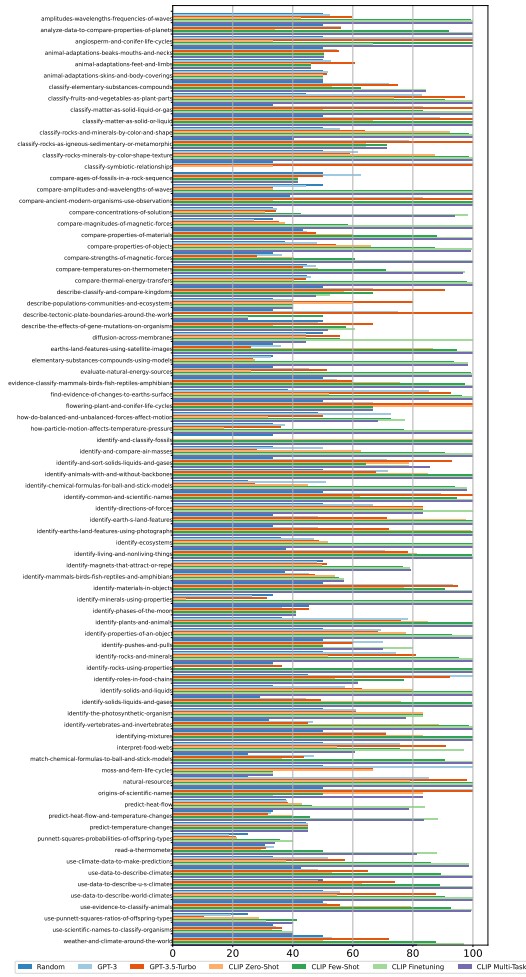

Figure 26: Accuracy per skill on science.

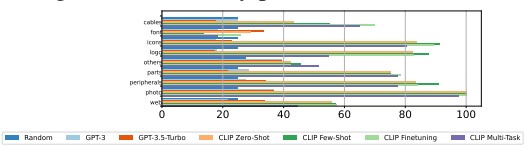

Figure 27: Accuracy per skill on technology.

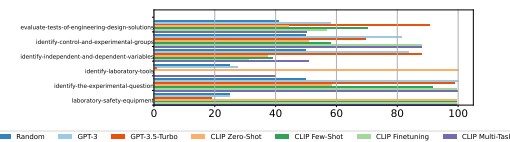

Figure 28: Accuracy per skill on engineering.

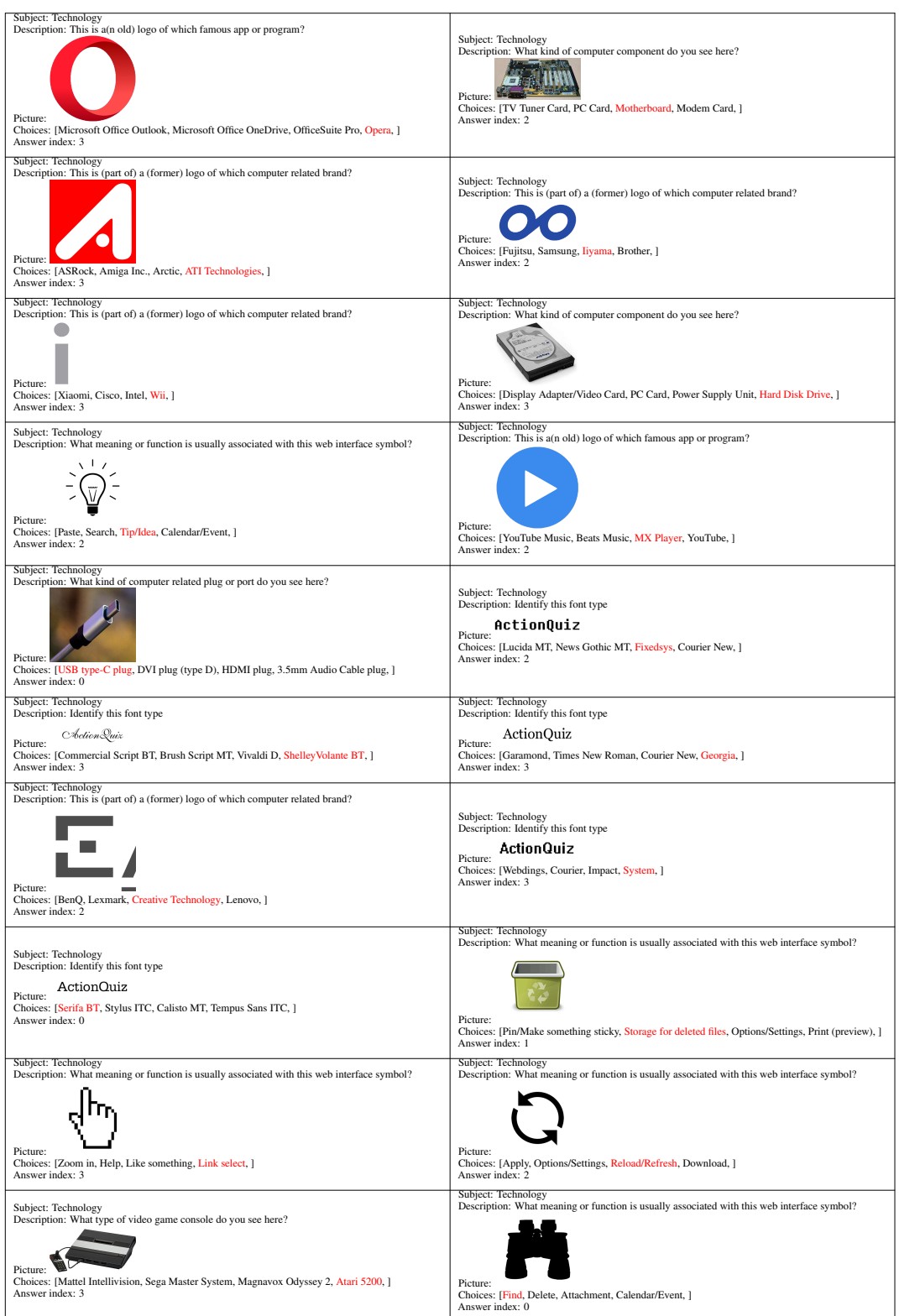

Table 12: Human evaluation problem set (part 1).

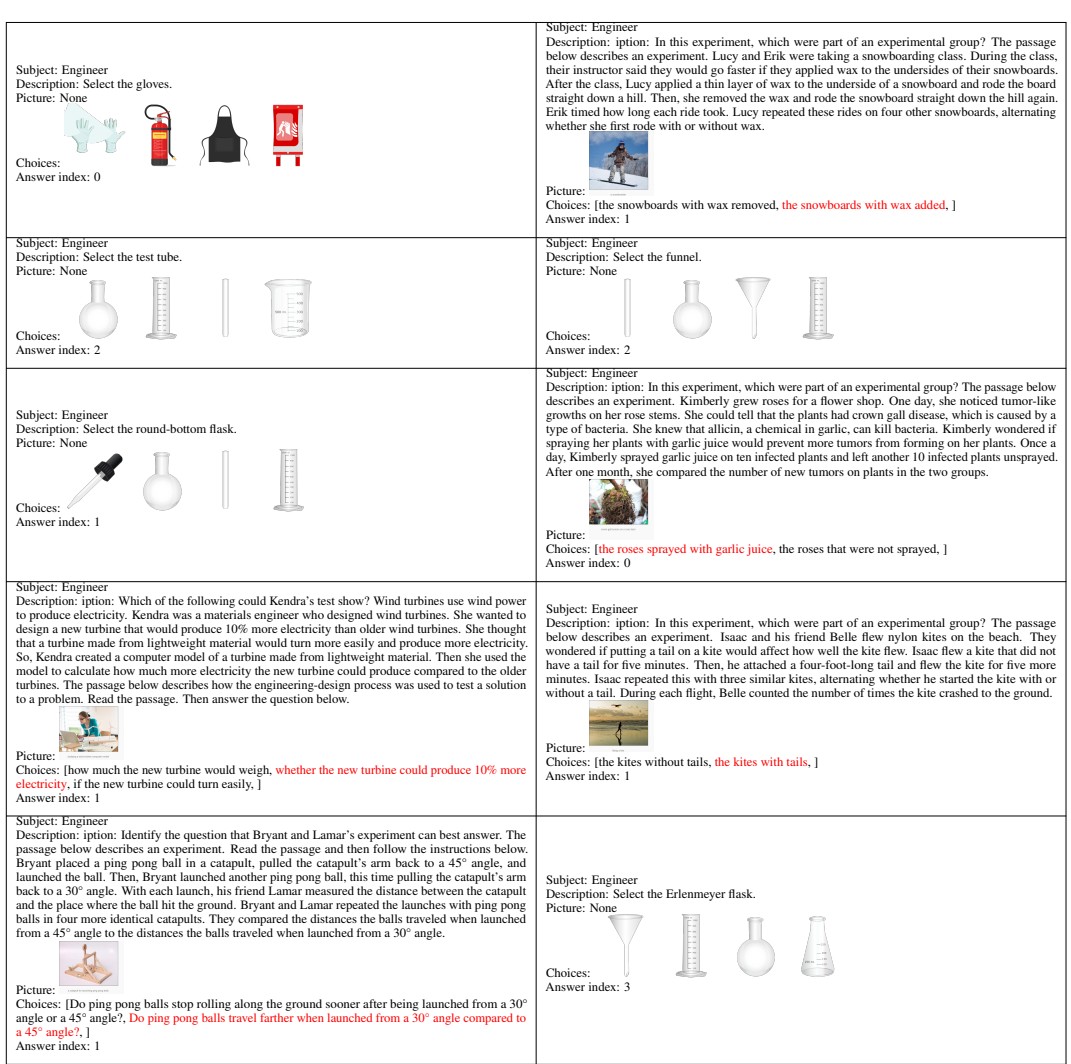

| | |
|---|---|
| Subject: Engineer
Description: Select the gloves.
Picture: None

Choices:
Answer index: 0 | Subject: Engineer
Description: iption: In this experiment, which were part of an experimental group? The passage below describes an experiment. Lucy and Erik were taking a snowboarding class. During the class, their instructor said they would go faster if they applied wax to the undersides of their snowboards. After the class, Lucy applied a thin layer of wax to the underside of a snowboard and rode the board straight down a hill. Then, she removed the wax and rode the snowboard straight down the hill again. Erik timed how long each ride took. Lucy repeated these rides on four other snowboards, alternating whether she first rode with or without wax.

Picture:
Choices: [the snowboards with wax removed, the snowboards with wax added, ]
Answer index: 1 |
| Subject: Engineer
Description: Select the test tube.
Picture: None

Choices:
Answer index: 2 | Subject: Engineer
Description: Select the funnel.
Picture: None

Choices:
Answer index: 2 |
| Subject: Engineer
Description: Select the round-bottom flask.
Picture: None

Choices:
Answer index: 1 | Subject: Engineer
Description: iption: In this experiment, which were part of an experimental group? The passage below describes an experiment. Kimberly grew roses for a flower shop. One day, she noticed tumor-like growths on her rose stems. She could tell that the plants had crown gall disease, which is caused by a type of bacteria. She knew that allicin, a chemical in garlic, can kill bacteria. Kimberly wondered if spraying her plants with garlic juice would prevent more tumors from forming on her plants. Once a day, Kimberly sprayed garlic juice on ten infected plants and left another 10 infected plants unsprayed. After one month, she compared the number of new tumors on plants in the two groups.

Picture:
Choices: [the roses sprayed with garlic juice, the roses that were not sprayed, ]
Answer index: 0 |
| Subject: Engineer
Description: iption: Which of the following could Kendra's test show? Wind turbines use wind power to produce electricity. Kendra was a materials engineer who designed wind turbines. She wanted to design a new turbine that would produce 10% more electricity than older wind turbines. She thought that a turbine made from lightweight material would turn more easily and produce more electricity. So, Kendra created a computer model of a turbine made from lightweight material. Then she used the model to calculate how much more electricity the new turbine could produce compared to the older turbines. The passage below describes how the engineering-design process was used to test a solution to a problem. Read the passage. Then answer the question below.

Picture:
Choices: [how much the new turbine would weigh, whether the new turbine could produce 10% more electricity, if the new turbine could turn easily, ]
Answer index: 1 | Subject: Engineer
Description: iption: In this experiment, which were part of an experimental group? The passage below describes an experiment. Isaac and his friend Belle flew nylon kites on the beach. They wondered if putting a tail on a kite would affect how well the kite flew. Isaac flew a kite that did not have a tail for five minutes. Then, he attached a four-foot-long tail and flew the kite for five more minutes. Isaac repeated this with three similar kites, alternating whether he started the kite with or without a tail. During each flight, Belle counted the number of times the kite crashed to the ground.

Picture:
Choices: [the kites without tails, the kites with tails, ]
Answer index: 1 |
| Subject: Engineer
Description: iption: Identify the question that Bryant and Lamar's experiment can best answer. The passage below describes an experiment. Read the passage and then follow the instructions below. Bryant placed a ping pong ball in a catapult, pulled the catapult's arm back to a 45° angle, and launched the ball. Then, Bryant launched another ping pong ball, this time pulling the catapult's arm back to a 30° angle. With each launch, his friend Lamar measured the distance between the catapult and the place where the ball hit the ground. Bryant and Lamar repeated the launches with ping pong balls in four more identical catapults. They compared the distances the balls traveled when launched from a 45° angle to the distances the balls traveled when launched from a 30° angle.

Picture:
Choices: [Do ping pong balls stop rolling along the ground sooner after being launched from a 30° angle or a 45° angle?, Do ping pong balls travel farther when launched from a 30° angle compared to a 45° angle?, ]
Answer index: 1 | Subject: Engineer
Description: Select the Erlenmeyer flask.
Picture: None

Choices:
Answer index: 3 |

Table 13: Human evaluation problem set (part 2).

| | |
|---|---|
| Subject: Engineer
Description: iption: Which of the following could Ivan's test show? Ivan was a landscape architect who was hired to design a new city park. The city council wanted the park to have space for outdoor concerts and to have at least 20% of the park shaded by trees. Ivan thought the concert area should be at least 150 meters from the road so traffic noise didn't interrupt the music. He developed three possible designs for the park with the concert area in a different location in each design. Then, he tested each design by measuring the distance between the road and the concert area. The passage below describes how the engineering-design process was used to test a solution to a problem. Read the passage. Then answer the question below.
Picture: 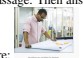
Choices: [if at least 20% of the park would be shaded by trees in each design, which design would have the greatest distance between the concert area and the road, which design would have the least traffic noise in the concert area, ]
Answer index: 1 | Subject: Engineer
Description: Select the beaker.
Picture: None
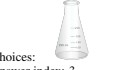 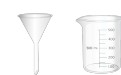
Choices:
Answer index: 3 |
| Subject: Engineer
Description: iption: Identify the question that Zeke's experiment can best answer. The passage below describes an experiment. Read the passage and then follow the instructions below. Zeke divided 40 unripe bananas evenly among eight paper bags and sealed the bags. He poked 20 small holes in four of the bags and left the other four without holes. He kept the bags at room temperature for three days. Then, Zeke opened the bags and counted the number of brown spots on each banana. He compared the average number of brown spots on bananas from bags with holes to the average number of brown spots on bananas from bags without holes.
Picture: 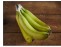
Choices: [Do bananas develop more brown spots if they are kept in bags with holes compared to bags without holes?, Do bananas develop more brown spots when they are kept at room temperature compared to in a cold refrigerator?, ]
Answer index: 0 | Subject: Engineer
Description: iption: Hint: An independent variable is a variable whose effect you are investigating. A dependent variable is a variable that you measure. Which of the following was an independent variable in this experiment? The passage below describes an experiment. Read the passage and think about the variables that are described. Tyler designed an electric circuit to test how well different types of metal conduct electricity. The circuit included a battery, a light bulb, wires, and clips that could be attached to a sheet of metal. If the metal conducted electricity poorly, the light bulb would appear dim. If the metal conducted electricity well, the light bulb would appear bright. Tyler collected nine equally sized sheets of metal: three sheets of copper, three sheets of iron, and three sheets of aluminum. He used the clips to attach each metal sheet, one sheet at a time, to the circuit. For each sheet, Tyler used a light meter to measure how much light the bulb produced.
Picture: 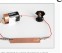
Choices: [the amount of light produced by the light bulb, the type of metal sheet used in the circuit, ]
Answer index: 1 |
| Subject: Engineer
Description: iption: Identify the question that Devon's experiment can best answer. The passage below describes an experiment. Read the passage and then follow the instructions below. Devon poured four ounces of water into each of six glasses. Devon dissolved one tablespoon of salt in each of three glasses, and did not add salt to the other three. Then, Devon placed an egg in one glass and observed if the egg floated. He removed the egg and dried it. She repeated the process with the other five glasses, recording each time if the egg floated. Devon repeated this test with two more eggs and counted the number of times the eggs floated in fresh water compared to salty water.
Picture: 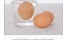
Choices: [Does the amount of water in a glass affect whether eggs sink or float in the water?, Are eggs more likely to float in fresh water or salty water?, ]
Answer index: 1 | Subject: Engineer
Description: iption: Which of the following could Luke's test show? Luke had a cookie recipe that made soft, thick cookies. But he preferred crunchy cookies. Luke read that using different types of sugar affects how firm the cookies are. His recipe used both white and brown sugar, so he decided to see if the cookies would be crunchy if he didn't use any brown sugar. Luke baked a batch of cookies using his recipe, but he left out the brown sugar and doubled the amount of white sugar. He baked the cookies for the same amount of time as in his original recipe. After the cookies finished baking and cooling, he tried one to find out how firm it was. The passage below describes how the engineering-design process was used to test a solution to a problem. Read the passage. Then answer the question below.
Picture: 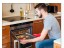
Choices: [if cookies made with only white sugar were soft, if baking cookies for longer made them more crunchy, if cookies made with double the amount of brown sugar were crunchy, ]
Answer index: 0 |
| Subject: Engineer
Description: iption: Identify the question that Myra's experiment can best answer. The passage below describes an experiment. Read the passage and then follow the instructions below. Myra glued lids onto 16 cardboard shoe boxes of equal size. She painted eight of the boxes black and eight of the boxes white. Myra made a small hole in the side of each box and then stuck a thermometer partially into each hole so she could measure the temperatures inside the boxes. She placed the boxes in direct sunlight in her backyard. Two hours later, she measured the temperature inside each box. Myra compared the average temperature inside the black boxes to the average temperature inside the white boxes.
Picture: 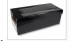
Choices: [Do the temperatures inside boxes depend on the sizes of the boxes?, Do the insides of white boxes get hotter than the insides of black boxes when the boxes are left in the sun?, ]
Answer index: 1 | Subject: Engineer
Description: iption: Which of the following could Zoe and Evelyn's test show? Zoe and Evelyn were making batches of concrete for a construction project. To make the concrete, they mixed together dry cement powder, gravel, and water. Then, they checked if each batch was firm enough using a test called a slump test. They poured some of the fresh concrete into an upside-down metal cone. They left the concrete in the metal cone for 30 seconds. Then, they lifted the cone to see if the concrete stayed in a cone shape or if it collapsed. If the concrete in a batch collapsed, they would know the batch should not be used. The passage below describes how the engineering-design process was used to test a solution to a problem. Read the passage. Then answer the question below.
Picture: 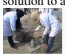
Choices: [if the concrete from each batch took the same amount of time to dry, if a new batch of concrete was firm enough to use, ]
Answer index: 1 |
| Subject: Engineer
Description: iption: Identify the question that Belle's experiment can best answer. The passage below describes an experiment. Read the passage and then follow the instructions below. Belle planted 25 tomato seeds one-half inch below the soil surface in each of six pots. Belle added an equal amount of fertilizer to three of the six pots. She placed the pots in a plant growth chamber where all the seeds experienced the same temperature, amount of light, and humidity level. After two weeks, Belle counted the number of seedlings that grew in each pot. She compared the number of seedlings in the pots with fertilizer to the number of seedlings in the pots without fertilizer.
Picture: 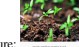
Choices: [Do more tomato seedlings grow when they are planted in soil with fertilizer compared to soil without fertilizer?, Does the humidity level where tomato seeds are planted affect the number of tomato seedlings that grow?, ]
Answer index: 0 | Subject: Engineer
Description: iption: In this experiment, which were part of a control group? The passage below describes an experiment. After a severe winter storm, Sandeep's driveway was covered with ice. He read that salt makes ice melt at a lower temperature. Before covering his entire driveway with salt, he wanted to know if adding salt could actually help melt ice in the freezing outdoor temperatures. Sandeep weighed twenty ice cubes. He sprinkled salt on half of the ice cubes and left the other half unsalted. He placed all the ice cubes outside. One hour later, Sandeep quickly dried each ice cube and reweighed it to see how much it had melted.
Picture: 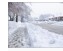
Choices: [the salted ice cubes, the unsalted ice cubes, ]
Answer index: 1 |

Table 14: Human evaluation problem set (part 3).

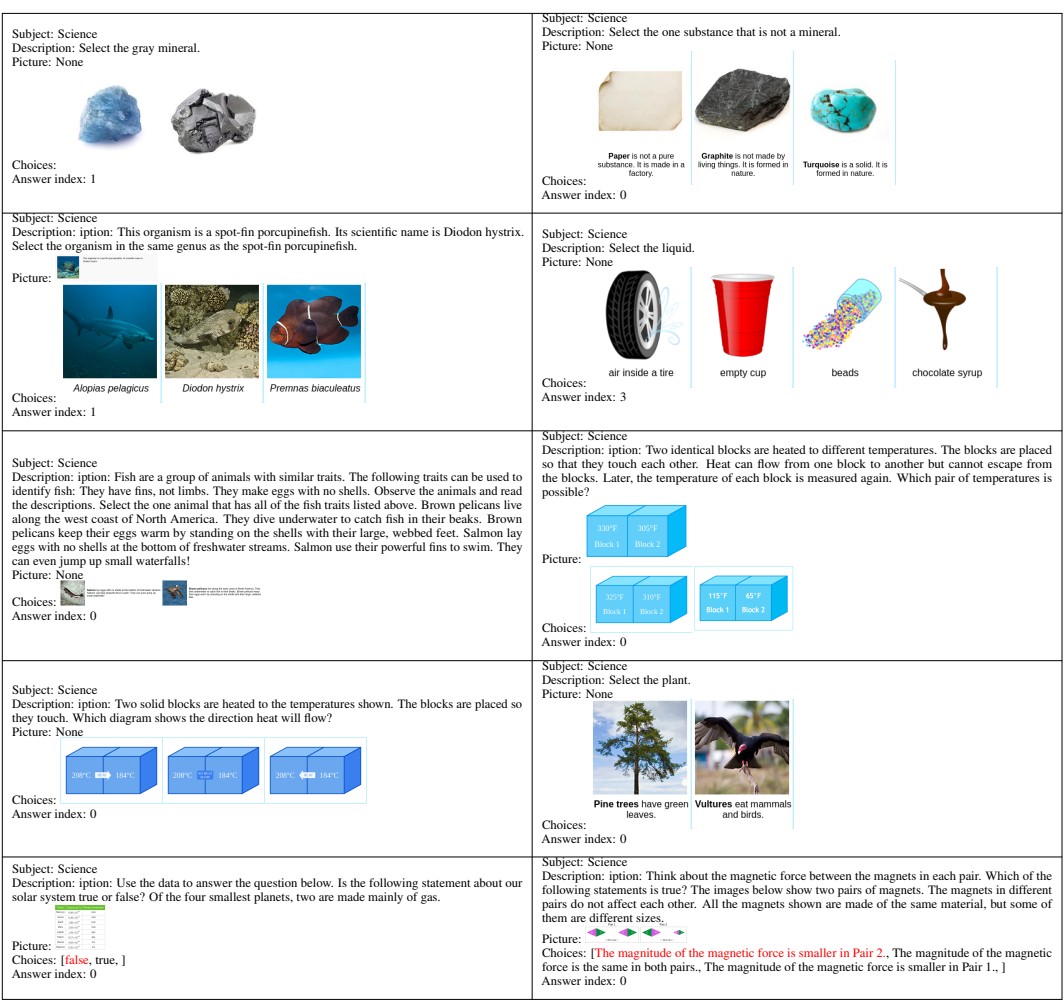

Table 15: Human evaluation problem set (part 4).

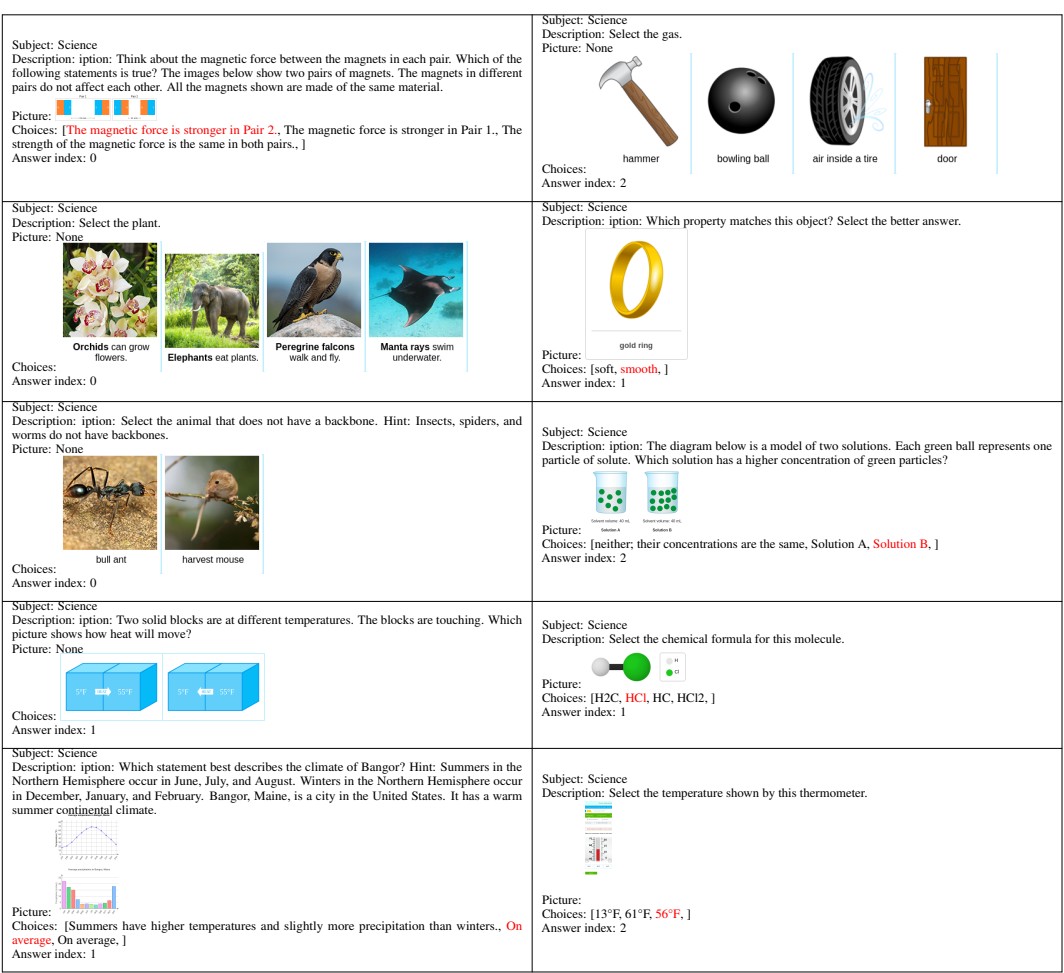

Table 16: Human evaluation problem set (part 5).

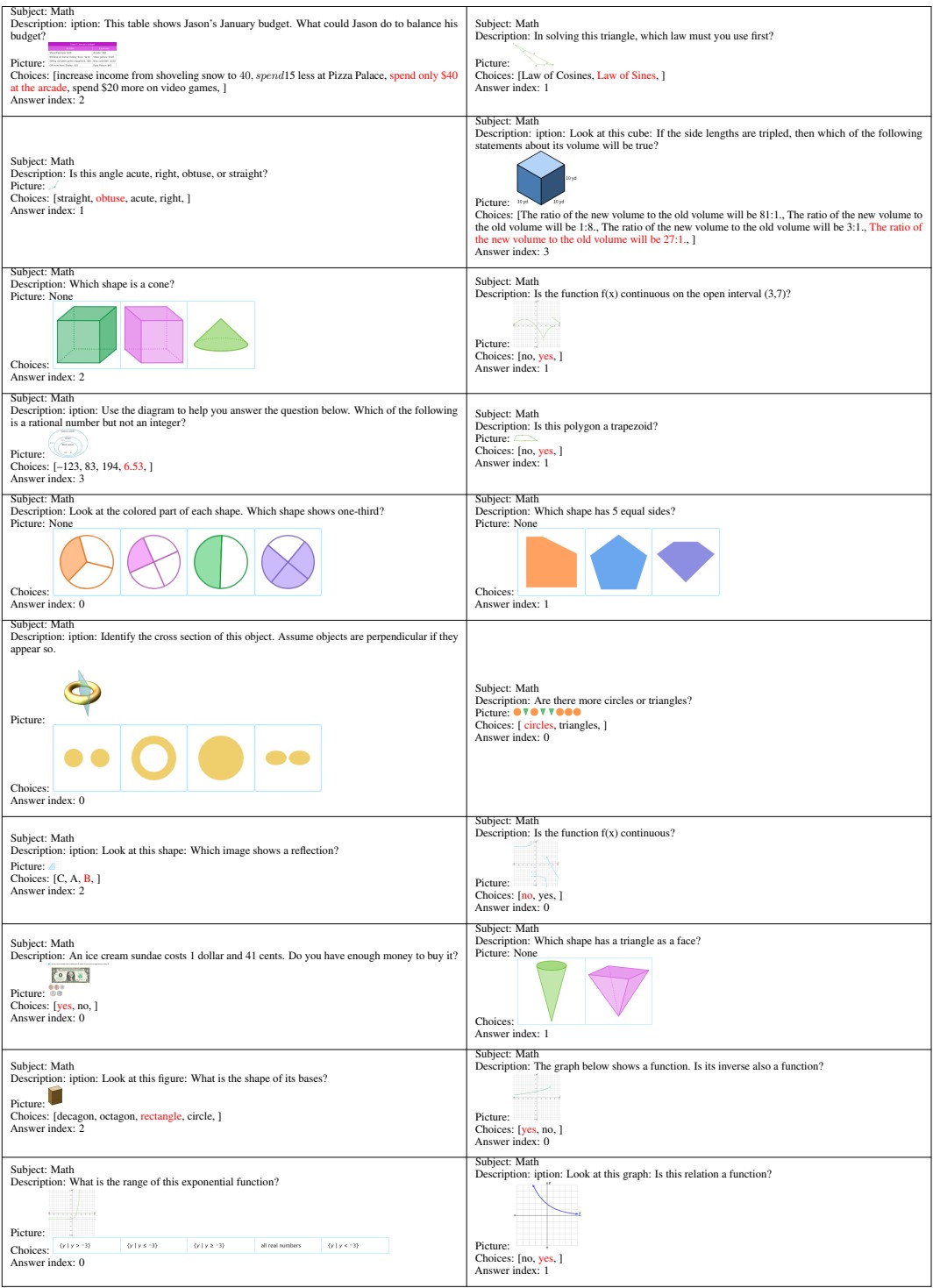

Table 17: Human evaluation problem set (part 6).

| Subject | Grade | Skills |
|---|---|---|
| Science | grade-2 | classify-fruits-and-vegetables-as-plant-parts, classify-matter-as-solid-liquid-or-gas, classify-matter-as-solid-or-liquid, classify-rocks-and-minerals-by-color-and-shape, compare-properties-of-materials, compare-properties-of-objects, compare-temperatures-on-thermometers, find-evidence-of-changes-to-earths-surface, identify-animals-with-and-without-backbones, identify-earth-s-land-features, identify-living-and-nonliving-things, identify-magnets-that-attract-or-repel, identify-mammals-birds-fish-reptiles-and-amphibians, identify-materials-in-objects, identify-plants-and-animals, identify-properties-of-an-object, identify-pushes-and-pulls, identify-solids-and-liquids, identify-solids-liquids-and-gases, identifying-mixtures, natural-resources, predict-heat-flow, read-a-thermometer |
| Science | grade-3 | animal-adaptations-beaks-mouths-and-necks, animal-adaptations-feet-and-limbs, animal-adaptations-skins-and-body-coverings, classify-matter-as-solid-liquid-or-gas, classify-rocks-and-minerals-by-color-shape-and-texture, classify-rocks-as-igneous-sedimentary-or-metamorphic, compare-ancient-and-modern-organisms-use-observations-to-support-a-hypothesis, compare-properties-of-materials, compare-properties-of-objects, compare-strengths-of-magnetic-forces, compare-temperatures-on-thermometers, find-evidence-of-changes-to-earths-surface, how-do-balanced-and-unbalanced-forces-affect-motion, identify-earth-s-land-features, identify-ecosystems, identify-living-and-nonliving-things, identify-magnets-that-attract-or-repel, identify-mammals-birds-fish-reptiles-and-amphibians, identify-materials-in-objects, identify-minerals-using-properties, identify-plants-and-animals, identify-properties-of-an-object, identify-pushes-and-pulls, identify-rocks-using-properties, identify-roles-in-food-chains, identify-solids-liquids-and-gases, identify-vertebrates-and-invertebrates, interpret-food-webs, natural-resources, predict-heat-flow, predict-temperature-changes, read-a-thermometer, use-climate-data-to-make-predictions, use-data-to-describe-u-s-climates, use-data-to-describe-world-climates, weather-and-climate-around-the-world |
| Science | grade-4 | animal-adaptations-beaks-mouths-and-necks, animal-adaptations-feet-and-limbs, animal-adaptations-skins-and-body-coverings, classify-fruits-and-vegetables-as-plant-parts, classify-rocks-as-igneous-sedimentary-or-metamorphic, compare-amplitudes-and-wavelengths-of-waves, compare-ancient-and-modern-organisms-use-observations-to-support-a-hypothesis, compare-properties-of-materials, compare-properties-of-objects, compare-strengths-of-magnetic-forces, compare-temperatures-on-thermometers, describe-classify-and-compare-kingdoms, evaluate-natural-energy-sources, how-do-balanced-and-unbalanced-forces-affect-motion, identify-and-sort-solids-liquids-and-gases, identify-common-and-scientific-names, identify-directions-of-forces, identify-earth-land-features-using-photographs, identify-earths-land-features-using-satellite-images, identify-ecosystems, identify-living-and-nonliving-things, identify-magnets-that-attract-or-repel, identify-mammals-birds-fish-reptiles-and-amphibians, identify-minerals-using-properties, identify-phases-of-the-moon, identify-rocks-using-properties, identify-roles-in-food-chains, interpret-food-webs, origins-of-scientific-names, predict-heat-flow, predict-temperature-changes, read-a-thermometer, use-climate-data-to-make-predictions, use-data-to-describe-climates, use-evidence-to-classify-animals, use-evidence-to-classify-mammals-birds-fish-reptiles-and-amphibians, use-scientific-names-to-classify-organisms, weather-and-climate-around-the-world |
| Science | grade-5 | animal-adaptations-beaks-mouths-and-necks, animal-adaptations-feet-and-limbs, animal-adaptations-skins-and-body-coverings, classify-elementary-substances-and-compounds-using-models, classify-fruits-and-vegetables-as-plant-parts, classify-rocks-as-igneous-sedimentary-or-metamorphic, compare-amplitudes-and-wavelengths-of-waves, compare-ancient-and-modern-organisms-use-observations-to-support-a-hypothesis, compare-magnitudes-of-magnetic-forces, compare-properties-of-objects, describe-classify-and-compare-kingdoms, evaluate-natural-energy-sources, flowering-plant-and-conifer-life-cycles, how-do-balanced-and-unbalanced-forces-affect-motion, identify-and-classify-fossils, identify-common-and-scientific-names, identify-directions-of-forces, identify-earths-land-features-using-photographs, identify-earths-land-features-using-satellite-images, identify-magnets-that-attract-or-repel, identify-rocks-and-minerals, identify-roles-in-food-chains, identify-the-photosynthetic-organism, identify-vertebrates-and-invertebrates, match-chemical-formulas-to-ball-and-stick-models, moss-and-fern-life-cycles, origins-of-scientific-names, predict-heat-flow, predict-temperature-changes, use-data-to-describe-climates, use-evidence-to-classify-animals, use-evidence-to-classify-mammals-birds-fish-reptiles-and-amphibians, use-scientific-names-to-classify-organisms, weather-and-climate-around-the-world |
| Science | grade-6 | analyze-data-to-compare-properties-of-planets, classify-elementary-substances-and-compounds-using-models, classify-rocks-as-igneous-sedimentary-or-metamorphic, classify-symbiotic-relationships, compare-ages-of-fossils-in-a-rock-sequence, compare-amplitudes-wavelengths-and-frequencies-of-waves, compare-concentrations-of-solutions, compare-magnitudes-of-magnetic-forces, compare-thermal-energy-transfers, describe-populations-communities-and-ecosystems, describe-tectonic-plate-boundaries-around-the-world, describe-the-effects-of-gene-mutations-on-organisms, flowering-plant-and-conifer-life-cycles, identify-and-compare-air-masses, identify-common-and-scientific-names, identify-earths-land-features-using-photographs, identify-earths-land-features-using-satellite-images, identify-ecosystems, identify-elementary-substances-and-compounds-using-models, identify-how-particle-motion-affects-temperature-and-pressure, identify-phases-of-the-moon, identify-rocks-and-minerals, identify-the-photosynthetic-organism, match-chemical-formulas-to-ball-and-stick-models, moss-and-fern-life-cycles, origins-of-scientific-names, predict-heat-flow-and-temperature-changes, use-data-to-describe-climates, use-scientific-names-to-classify-organisms, weather-and-climate-around-the-world |
| Science | grade-7 | analyze-data-to-compare-properties-of-planets, angiosperm-and-conifer-life-cycles, classify-elementary-substances-and-compounds-using-models, classify-rocks-as-igneous-sedimentary-or-metamorphic, classify-symbiotic-relationships, compare-ages-of-fossils-in-a-rock-sequence, compare-amplitudes-wavelengths-and-frequencies-of-waves, compare-concentrations-of-solutions, compare-magnitudes-of-magnetic-forces, compare-thermal-energy-transfers, describe-populations-communities-and-ecosystems, describe-tectonic-plate-boundaries-around-the-world, describe-the-effects-of-gene-mutations-on-organisms, diffusion-across-membranes, identify-and-compare-air-masses, identify-chemical-formulas-for-ball-and-stick-models, identify-common-and-scientific-names, identify-ecosystems, identify-how-particle-motion-affects-temperature-and-pressure, identify-phases-of-the-moon, identify-rocks-and-minerals, identify-the-photosynthetic-organism, moss-and-fern-life-cycles, origins-of-scientific-names, predict-heat-flow-and-temperature-changes, use-data-to-describe-climates, use-scientific-names-to-classify-organisms |
| Science | grade-8 | analyze-data-to-compare-properties-of-planets, angiosperm-and-conifer-life-cycles, classify-elementary-substances-and-compounds-using-models, classify-symbiotic-relationships, compare-ages-of-fossils-in-a-rock-sequence, compare-amplitudes-wavelengths-and-frequencies-of-waves, compare-concentrations-of-solutions, compare-magnitudes-of-magnetic-forces, compare-thermal-energy-transfers, describe-populations-communities-and-ecosystems, describe-tectonic-plate-boundaries-around-the-world, describe-the-effects-of-gene-mutations-on-organisms, diffusion-across-membranes, identify-and-compare-air-masses, identify-chemical-formulas-for-ball-and-stick-models, identify-common-and-scientific-names, identify-ecosystems, identify-how-particle-motion-affects-temperature-and-pressure, identify-phases-of-the-moon, identify-rocks-and-minerals, identify-the-photosynthetic-organism, moss-and-fern-life-cycles, origins-of-scientific-names, predict-heat-flow-and-temperature-changes, use-data-to-describe-climates, use-punnett-squares-to-calculate-probabilities-of-offspring-types, use-punnett-squares-to-calculate-ratios-of-offspring-types, use-scientific-names-to-classify-organisms |
| Technology | - | cables, font, icons, logo, parts, peripherals, photo, web, others |
| Engineering | grade-5 | identify-laboratory-tools |
| Engineering | grade-6 | evaluate-tests-of-engineering-design-solutions, identify-control-and-experimental-groups, identify-independent-and-dependent-variables, identify-laboratory-tools, identify-the-experimental-question, laboratory-safety-equipment |
| Engineering | grade-7 | evaluate-tests-of-engineering-design-solutions, identify-control-and-experimental-groups, identify-independent-and-dependent-variables, identify-laboratory-tools, identify-the-experimental-question, laboratory-safety-equipment |
| Engineering | grade-8 | identify-control-and-experimental-groups, identify-laboratory-tools, identify-the-experimental-question, laboratory-safety-equipment |

Table 18: Full skill summary (part 1), including science, technology and engineering skills.

| Subject | Grade | Skills |
|---|---|---|
| | algebra-1 | compare-linear-functions-graphs-and-equations, compare-linear-functions-tables-graphs-and-equations, describe-linear-and-exponential-growth-and-decay, domain-and-range-of-absolute-value-functions-graphs, domain-and-range-of-exponential-functions-graphs, domain-and-range-of-square-root-functions-graphs, factor-quadratics-using-algebra-tiles, identify-direct-variation-and-inverse-variation, identify-functions, identify-functions-vertical-line-test, identify-linear-and-exponential-functions-from-graphs, identify-linear-and-exponential-functions-from-tables, identify-linear-functions-from-graphs-and-equations, identify-linear-functions-from-tables, identify-linear-quadratic-and-exponential-functions-from-graphs, identify-linear-quadratic-and-exponential-functions-from-tables, identify-proportional-relationships, interpret-a-scatter-plot, interpret-the-slope-and-y-intercept-of-a-linear-function, linear-functions-over-unit-intervals, match-exponential-functions-and-graphs-ii, model-and-solve-linear-equations-using-algebra-tiles, multiply-two-binomials-using-algebra-tiles, perimeter-and-area-changes-in-scale, perimeter-area-and-volume-changes-in-scale, special-right-triangles, surface-area-and-volume-changes-in-scale, write-compound-inequalities-from-graphs |
| | algebra-2 | classify-variation, describe-linear-and-exponential-growth-and-decay, domain-and-range-of-absolute-value-functions-graphs, domain-and-range-of-exponential-and-logarithmic-functions, domain-and-range-of-radical-functions, factor-quadratics-using-algebra-tiles, find-inverse-functions-and-relations, find-solutions-using-a-table, graphs-of-angles, identify-the-direction-a-parabola-opens, linear-functions-over-unit-intervals, match-exponential-functions-and-graphs, outliers-in-scatter-plots, solve-a-triangle |
| | calculus | describe-linear-and-exponential-growth-and-decay, determine-continuity-on-an-interval-using-graphs, determine-continuity-using-graphs, determine-one-sided-continuity-using-graphs, domain-and-range, domain-and-range-of-exponential-and-logarithmic-functions, find-inverse-functions-and-relations, find-limits-at-vertical-asymptotes-using-graphs, identify-functions, identify-graphs-of-continuous-functions |

Table 19: Full skill summary (part 2), including math skills for algebra-{1,2} and calculus.

| Subject | Grade | Skills |
|---|---|---|
| Math | grade-1 | addition-sentences-up-to-10-what-does-the-model-show, addition-sentences-up-to-10-which-model-matches, addition-sentences-using-number-lines-sums-up-to-20, am-or-pm, certain-probable-unlikely-and-impossible, compare-clocks, compare-money-amounts, compare-objects-length-and-height, compare-sides-and-corners, compare-size-weight-and-capacity, compare-vertices-edges-and-faces, comparing-review, count-sides-and-corners, count-to-fill-a-ten-frame, cubes-and-rectangular-prisms, equal-sides, estimate-to-the-nearest-ten, even-or-odd, find-the-next-shape-in-a-growing-pattern, find-the-next-shape-in-a-pattern, flip-turn-and-slide, holds-more-or-less, identify-faces-of-three-dimensional-shapes, identify-fourths, identify-halves, identify-halves-and-fourths, identify-halves-thirds-and-fourths, identify-shapes-traced-from-solids, identify-thirds, interpret-bar-graphs-ii, light-and-heavy, match-analog-and-digital-clocks, match-analog-clocks-and-times, match-digital-clocks-and-times, more-less-and-equally-likely, name-the-three-dimensional-shape, name-the-two-dimensional-shape, names-and-values-of-all-coins, names-and-values-of-common-coins, open-and-closed-shapes, ordinal-numbers, purchases-do-you-have-enough-money, read-a-calendar, read-a-calendar-ii, rhombuses, select-three-dimensional-shapes, select-two-dimensional-shapes, shapes-of-everyday-objects, simple-fractions-what-fraction-does-the-shape-show, square-corners, subtraction-sentences-up-to-10-which-model-matches, subtraction-sentences-using-number-lines-up-to-10, subtraction-sentences-using-number-lines-up-to-20, symmetry, time-and-clocks-word-problems, times-of-everyday-events, two-dimensional-and-three-dimensional-shapes, which-bar-graph-is-correct, which-picture-graph-is-correct, which-table-is-correct, which-tally-chart-is-correct, wide-and-narrow |
| | grade-2 | am-or-pm, certain-probable-unlikely-and-impossible, choose-the-appropriate-measuring-tool, compare-clocks, compare-sides-and-vertices, compare-vertices-edges-and-faces, correct-amount-of-change, cubes, equal-sides, equivalent-amounts-of-money-up-to-1-dollar, estimate-to-the-nearest-ten, even-or-odd, find-the-next-shape-in-a-growing-pattern, find-the-next-shape-in-a-repeating-pattern, flip-turn-and-slide, fractions-of-a-group, fractions-of-a-whole-modeling-word-problems, greatest-and-least-word-problems-up-to-100, greatest-and-least-word-problems-up-to-1000, how-much-more-to-make-a-dollar, identify-faces-of-three-dimensional-shapes, identify-fourths, identify-halves, identify-halves-thirds-and-fourths, identify-lines-of-symmetry, identify-repeated-addition-in-arrays-sums-to-10, identify-repeated-addition-in-arrays-sums-to-25, identify-shapes-traced-from-solids, identify-the-fraction, identify-thirds, interpret-bar-graphs-ii, interpret-tally-charts, match-addition-sentences-and-models-sums-to-10, match-analog-and-digital-clocks, match-analog-clocks-and-times, match-digital-clocks-and-times, more-less-and-equally-likely, name-the-three-dimensional-shape, name-the-two-dimensional-shape, names-and-values-of-all-coins, names-and-values-of-common-coins, ordinal-numbers-up-to-10th, place-value-models-up-to-hundreds, place-value-tens-and-ones, place-value-up-to-hundreds, place-value-up-to-thousands, purchases-do-you-have-enough-money-up-to-1-dollar, purchases-do-you-have-enough-money-up-to-5-dollars, read-a-calendar, read-a-calendar-ii, read-a-thermometer, select-figures-with-a-given-area, select-three-dimensional-shapes, symmetry, which-shape-illustrates-the-fraction, which-table-is-correct, which-tally-chart-is-correct, write-subtraction-sentences-to-describe-pictures-up-to-18, write-subtraction-sentences-to-describe-pictures-up-to-two-digits |
| | grade-3 | acute-obtuse-and-right-triangles, am-or-pm, angles-greater-than-less-than-or-equal-to-a-right-angle, certain-probable-unlikely-and-impossible, choose-the-appropriate-measuring-tool, compare-area-and-perimeter-of-two-figures, compare-fractions-in-recipes, compare-fractions-using-models, compare-fractions-using-number-lines, coordinate-planes-as-maps, correct-amount-of-change, division-input-output-tables-find-the-rule, find-the-next-shape-in-a-pattern, fractions-of-a-group-denominators-2-3-4-6-8, fractions-of-a-group-unit-fractions, identify-multiplication-expressions-for-arrays, identify-multiplication-expressions-for-equal-groups, identify-parallelograms, identify-rhombuses, identify-three-dimensional-shapes, identify-trapezoids, identify-two-dimensional-shapes, identify-unit-fractions-on-number-lines, interpret-line-graphs, is-it-a-polygon, lines-line-segments-and-rays, match-analog-and-digital-clocks, match-clocks-and-times, match-fractions-to-models-halves-thirds-and-fourths, match-mixed-numbers-to-models, multiplication-input-output-tables-find-the-rule, open-and-closed-shapes, parallel-perpendicular-and-intersecting-lines, parallel-sides-in-quadrilaterals, purchases-do-you-have-enough-money-up-to-10-dollars, read-a-calendar, read-a-thermometer, reading-schedules, reflection-rotation-and-translation, scalene-isosceles-and-equilateral-triangles, select-figures-with-a-given-area, select-fractions-equivalent-to-whole-numbers-using-models, shapes-of-everyday-objects, symmetry, which-picture-shows-more |
| | grade-4 | acute-obtuse-and-right-triangles, acute-right-obtuse-and-straight-angles, angles-as-fractions-of-a-circle, angles-of-90-180-270-and-360-degrees, classify-triangles, compare-area-and-perimeter-of-two-figures, compare-decimals-using-models, compare-fractions-in-recipes, compare-fractions-using-models, compare-fractions-with-like-numerators-or-denominators-using-models, decompose-fractions-into-unit-fractions-using-models, elapsed-time, estimate-angle-measurements, find-the-next-shape-in-a-pattern, fractions-of-a-whole-word-problems, identify-equivalent-fractions-using-number-lines, identify-faces-of-three-dimensional-shapes, identify-lines-of-symmetry, identify-parallel-perpendicular-and-intersecting-lines, identify-parallelograms, identify-rhombuses, identify-three-dimensional-figures, identify-trapezoids, interpret-bar-graphs, interpret-stem-and-leaf-plots, is-it-a-polygon, measure-angles-with-a-protractor, multiplication-input-output-tables-find-the-rule, multiply-fractions-by-whole-numbers-using-models, multiply-unit-fractions-by-whole-numbers-using-models, nets-of-three-dimensional-figures, parallel-perpendicular-and-intersecting-lines, parallel-sides-in-quadrilaterals, points-lines-line-segments-rays-and-angles, properties-of-three-dimensional-figures, rotational-symmetry, scalene-isosceles-and-equilateral-triangles, sides-and-angles-of-quadrilaterals, transportation-schedules, what-decimal-number-is-illustrated |
| | grade-5 | acute-obtuse-and-right-triangles, adjust-a-budget, angles-of-90-180-270-and-360-degrees, classify-triangles, compare-decimals-using-grids, compare-fractions-and-mixed-numbers, compare-patterns, fractions-of-a-whole-word-problems, identify-parallelograms, identify-rhombuses, identify-three-dimensional-figures, identify-trapezoids, interpret-bar-graphs, is-it-a-polygon, line-symmetry, mean-find-the-missing-number, median-find-the-missing-number, multiplication-input-output-tables-find-the-rule, multiply-unit-fractions-by-whole-numbers-using-models, multiplying-fractions-by-whole-numbers-choose-the-model, nets-of-three-dimensional-figures, parallel-perpendicular-and-intersecting-lines, parallel-sides-in-quadrilaterals, points-lines-line-segments-rays-and-angles, range-find-the-missing-number, reflection-rotation-and-translation, regular-and-irregular-polygons, rotational-symmetry, rotational-symmetry-amount-of-rotation, scalene-isosceles-and-equilateral-triangles, three-dimensional-figures-viewed-from-different-perspectives, types-of-angles, understanding-probability |
| | grade-6 | absolute-value-and-integers-word-problems, changes-in-mean-median-mode-and-range, classify-rational-numbers-using-a-diagram, classify-triangles, compare-and-order-rational-numbers-using-number-lines, compare-area-and-perimeter-of-two-figures, compare-checking-accounts, front-side-and-top-view, identify-complementary-supplementary-vertical-adjacent-and-congruent-angles, identify-equivalent-expressions-using-strip-models, identify-polyhedra, identify-trapezoids, interpret-bar-graphs, interpret-double-bar-graphs, interpret-graphs-of-proportional-relationships, interpret-histograms, interpret-line-graphs, mean-median-mode-and-range-find-the-missing-number, model-and-solve-equations-using-algebra-tiles, nets-of-three-dimensional-figures, occupations-education-and-income, quadrants, rational-numbers-find-the-sign, reflection-rotation-and-translation, rotational-symmetry, rotational-symmetry-amount-of-rotation, similar-and-congruent-figures, understanding-area-of-a-triangle, understanding-area-of-trapezoids, understanding-percents-strip-models, which-figure-is-being-described, which-is-the-better-coupon, which-model-represents-the-ratio |
| | grade-7 | apply-addition-and-subtraction-rules, apply-multiplication-and-division-rules, bases-of-three-dimensional-figures, changes-in-mean-median-mode-and-range, classify-quadrilaterals, classify-rational-numbers-using-a-diagram, compare-and-order-integers, cross-sections-of-three-dimensional-figures, describe-a-sequence-of-transformations, front-side-and-top-view, identify-alternate-interior-and-alternate-exterior-angles, identify-complementary-supplementary-vertical-and-adjacent-angles, identify-equivalent-linear-expressions-using-algebra-tiles, identify-linear-and-nonlinear-functions, identify-reflections-rotations-and-translations, identify-trapezoids, identify-trends-with-scatter-plots, interpret-circle-graphs, interpret-graphs-of-proportional-relationships, line-symmetry, make-predictions-with-scatter-plots, mean-median-mode-and-range-find-the-missing-number, model-and-solve-equations-using-algebra-tiles, nets-of-three-dimensional-figures, parallel-perpendicular-and-intersecting-lines, parts-of-a-circle, perimeter-and-area-changes-in-scale, rational-numbers-find-the-sign, rotational-symmetry, rotational-symmetry-amount-of-rotation, similar-and-congruent-figures, simplify-expressions-by-combining-like-terms-with-algebra-tiles, transversals-of-parallel-lines-name-angle-pairs, which-is-the-better-coupon |
| | grade-8 | angle-angle-criterion-for-similar-triangles, apply-addition-and-subtraction-rules, apply-addition-subtraction-multiplication-and-division-rules, apply-multiplication-and-division-rules, base-plans, changes-in-mean-median-mode-and-range, classify-quadrilaterals, compare-and-order-integers, compare-linear-functions-graphs-and-equations, compare-linear-functions-tables-graphs-and-equations, congruent-triangles-sss-sas-and-asa, describe-a-sequence-of-transformations, front-side-and-top-view, identify-alternate-interior-and-alternate-exterior-angles, identify-complementary-supplementary-vertical-adjacent-and-congruent-angles, identify-congruent-figures, identify-functions-graphs, identify-linear-and-nonlinear-functions-graphs-and-equations, identify-linear-and-nonlinear-functions-tables, identify-lines-of-best-fit, identify-reflections-rotations-and-translations, identify-similar-triangles, identify-trapezoids, identify-trends-with-scatter-plots, interpret-graphs-of-proportional-relationships, interpret-the-slope-and-y-intercept-of-a-linear-function, irrational-numbers-on-number-lines, line-symmetry, make-predictions-with-scatter-plots, mean-median-mode-and-range-find-the-missing-number, model-and-solve-equations-using-algebra-tiles, multiply-polynomials-using-algebra-tiles, nets-of-three-dimensional-figures, parts-of-a-circle, parts-of-three-dimensional-figures, perimeter-and-area-changes-in-scale, quadrants-and-axes, rotational-symmetry, rotational-symmetry-amount-of-rotation, similar-and-congruent-figures, transversals-of-parallel-lines-name-angle-pairs |
| | kindergarten | addition-sentences-up-to-10-what-does-the-model-show, addition-sentences-up-to-10-which-model-matches, addition-sentences-up-to-5-what-does-the-model-show, addition-sentences-up-to-5-which-model-matches, am-or-pm, are-there-enough, circles, classify-shapes-by-color, coin-names-penny-through-quarter, compare-sides-and-corners, compare-size-weight-and-capacity, compare-two-groups-of-coins-pennies-through-dimes, cones, count-corners, count-cubes-up-to-10, count-cubes-up-to-5, count-dots-0-to-5, count-money-pennies-and-nickels, count-money-pennies-through-dimes, count-on-ten-frames-up-to-10, count-pictures-up-to-10, count-pictures-up-to-3, count-pictures-up-to-5, count-scattered-shapes-up-to-10, count-scattered-shapes-up-to-5, count-shapes-in-rings-up-to-10, count-shapes-in-rows-up-to-10, count-shapes-up-to-3, count-sides, count-cubes-up-to-100, count-to-fill-a-ten-frame, cubes, curved-parts, cylinders, different, equal-sides, fewer-and-more-compare-by-counting, fewer-and-more-compare-by-matching, fewer-and-more-compare-in-a-mixed-group, fewer-and-more-up-to-20, fewer-more-and-same, flat-and-solid-shapes, hexagons, hold-s-more-or-less, identify-halves-thirds-fourths, identify-pictures-with-symmetry, identify-shapes-traced-from-solids, inside-and-outside, introduction-to-symmetry, light-and-heavy, match-analog-and-digital-clocks, match-analog-clocks-and-times, match-digital-clocks-and-times, more-or-less-likely, name-the-three-dimensional-shape, name-the-two-dimensional-shape, one-less-with-pictures-up-to-5, one-more-and-one-less-with-pictures-up-to-10, one-more-with-pictures-up-to-10, one-more-with-pictures-up-to-5, ordinal-numbers-up-to-fifth, ordinal-numbers-up-to-tenth, rectangles, represent-numbers-up-to-10, represent-numbers-up-to-20, represent-numbers-with-shapes-up-to-3, represent-numbers-with-shapes-up-to-5, select-three-dimensional-shapes, select-two-dimensional-shapes, shapes-of-everyday-objects, spheres, square-corners, squares, subtraction-sentences-up-to-10-what-does-the-model-show, subtraction-sentences-up-to-10-which-model-matches, subtraction-sentences-up-to-5-what-does-the-model-show, subtraction-sentences-up-to-5-which-model-matches, take-apart-10-words, take-apart-numbers-up-to-10-words, take-apart-numbers-up-to-5-words, tall-and-short, times-of-everyday-events, triangles, wide-and-narrow |
| | pre-k | addition-sentences-up-to-10-what-does-the-model-show, addition-sentences-up-to-10-which-model-matches, addition-sentences-up-to-5-what-does-the-model-show, addition-sentences-up-to-5-which-model-matches, are-there-enough, circles, circles-squares-and-triangles, circles-squares-triangles-and-rectangles, classify-shapes-by-color, compare-size-weight-and-capacity, cones, count-corners, count-cubes-up-to-10, count-cubes-up-to-5, count-dots-up-to-10, count-dots-up-to-3, count-dots-up-to-5, count-on-ten-frames-up-to-10, count-on-ten-frames-up-to-3, count-on-ten-frames-up-to-5, count-pennies, count-pictures-up-to-10, count-pictures-up-to-3, count-pictures-up-to-5, count-scattered-shapes-up-to-10, count-scattered-shapes-up-to-5, count-shapes-in-rings-up-to-10, count-shapes-in-rows-up-to-10, count-shapes-in-rows-up-to-3, count-sides, count-sides-and-corners, cubes, cylinders, different, dimes-and-quarters, fewer-and-more-compare-by-matching, fewer-and-more-compare-in-a-mixed-group, fewer-more-and-same, flat-and-solid-shapes, holds-more-or-less, identify-shapes-traced-from-solids, inside-and-outside, light-and-heavy, more, name-the-shape, name-the-solid-shape, one-less-with-pictures-up-to-10, one-less-with-pictures-up-to-5, one-more-with-pictures-up-to-5, ordinal-numbers-up-to-fifth, ordinal-numbers-up-to-tenth, pennies-and-nickels, pennies-nickels-dimes-and-quarters, rectangles, represent-numbers-up-to-10, represent-numbers-up-to-20, represent-numbers-with-pictures-up-to-3, represent-numbers-with-pictures-up-to-5, represent-numbers-with-shapes-up-to-3, represent-numbers-with-shapes-up-to-5, select-solid-shapes, shapes-of-everyday-objects, spheres, squares, subtraction-sentences-up-to-10-which-model-matches, subtraction-sentences-up-to-5-which-model-matches, tall-and-short, tally-marks-up-to-10, triangles, what-comes-next, wide-and-narrow |
| | precalculus | determine-continuity-on-an-interval-using-graphs, determine-continuity-using-graphs, determine-one-sided-continuity-using-graphs, find-limits-at-vertical-asymptotes-using-graphs, identify-graphs-of-continuous-functions, outliers-in-scatter-plots, solve-a-triangle |

Table 20: Full skill summary (part 3), including math skills for grade 1-8 and pre-k, kindergarten and pre-calculus.

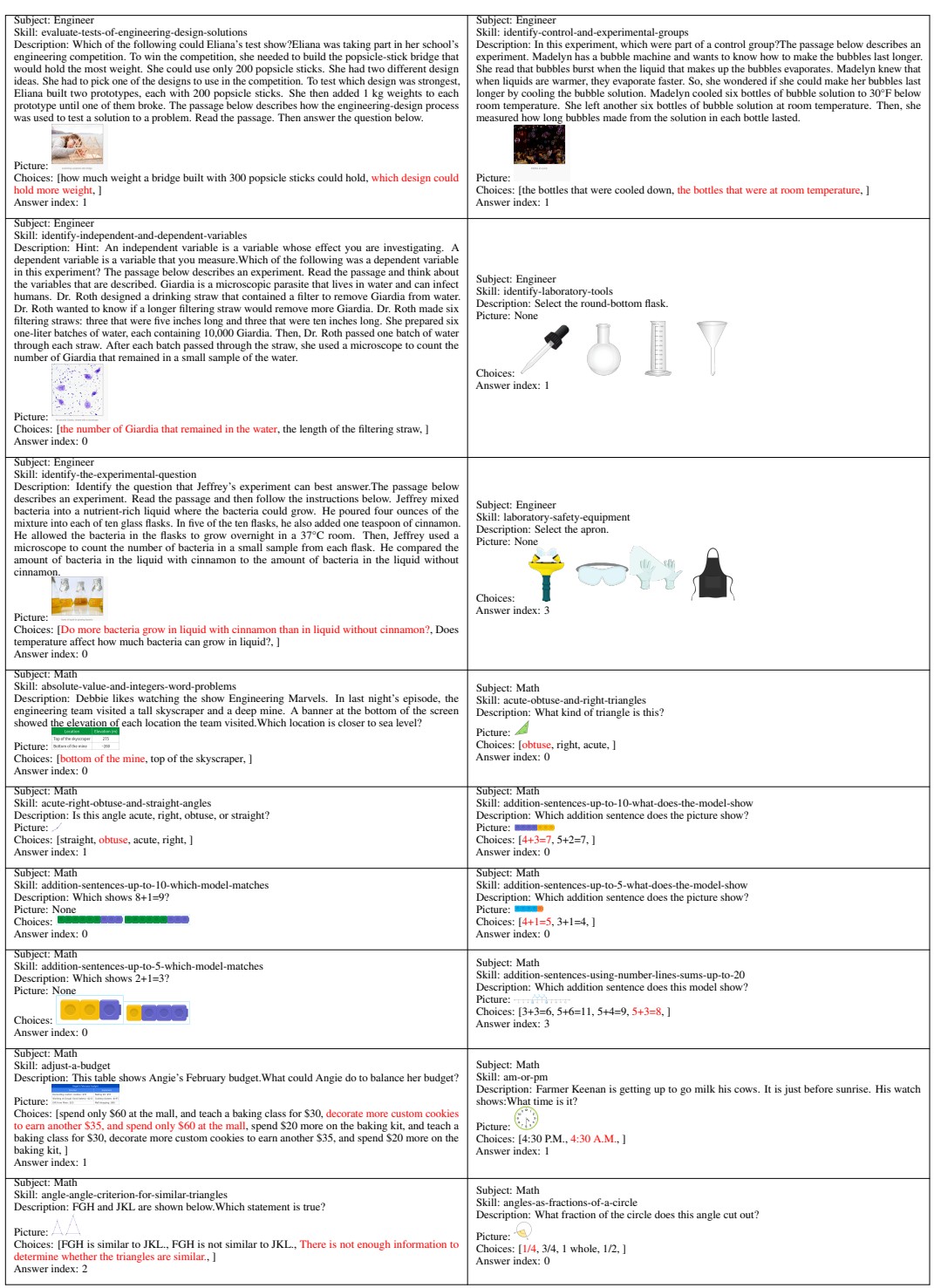

Table 21: Question examples for each skill (part 1).

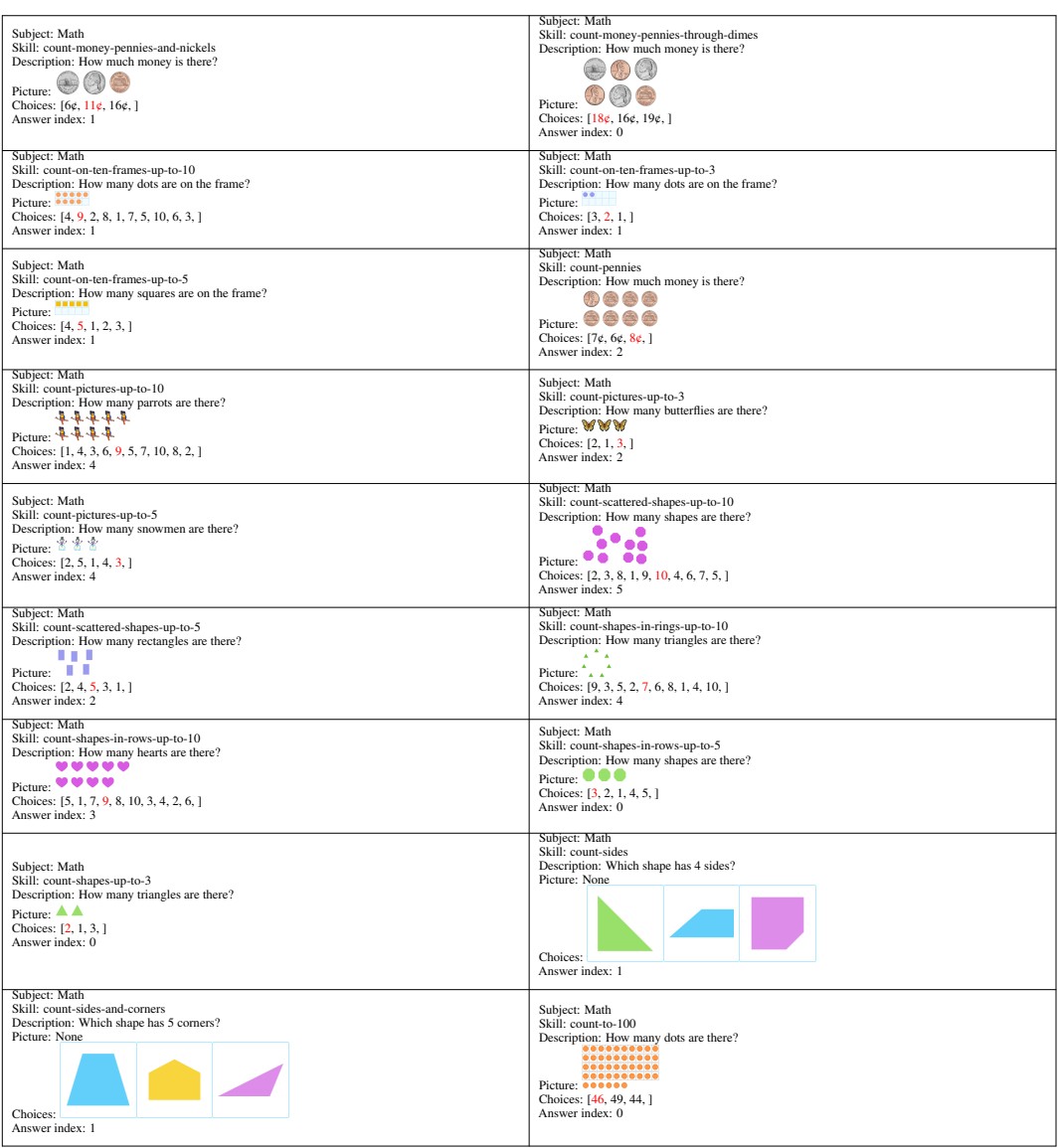

Table 22: Question examples for each skill (part 2).

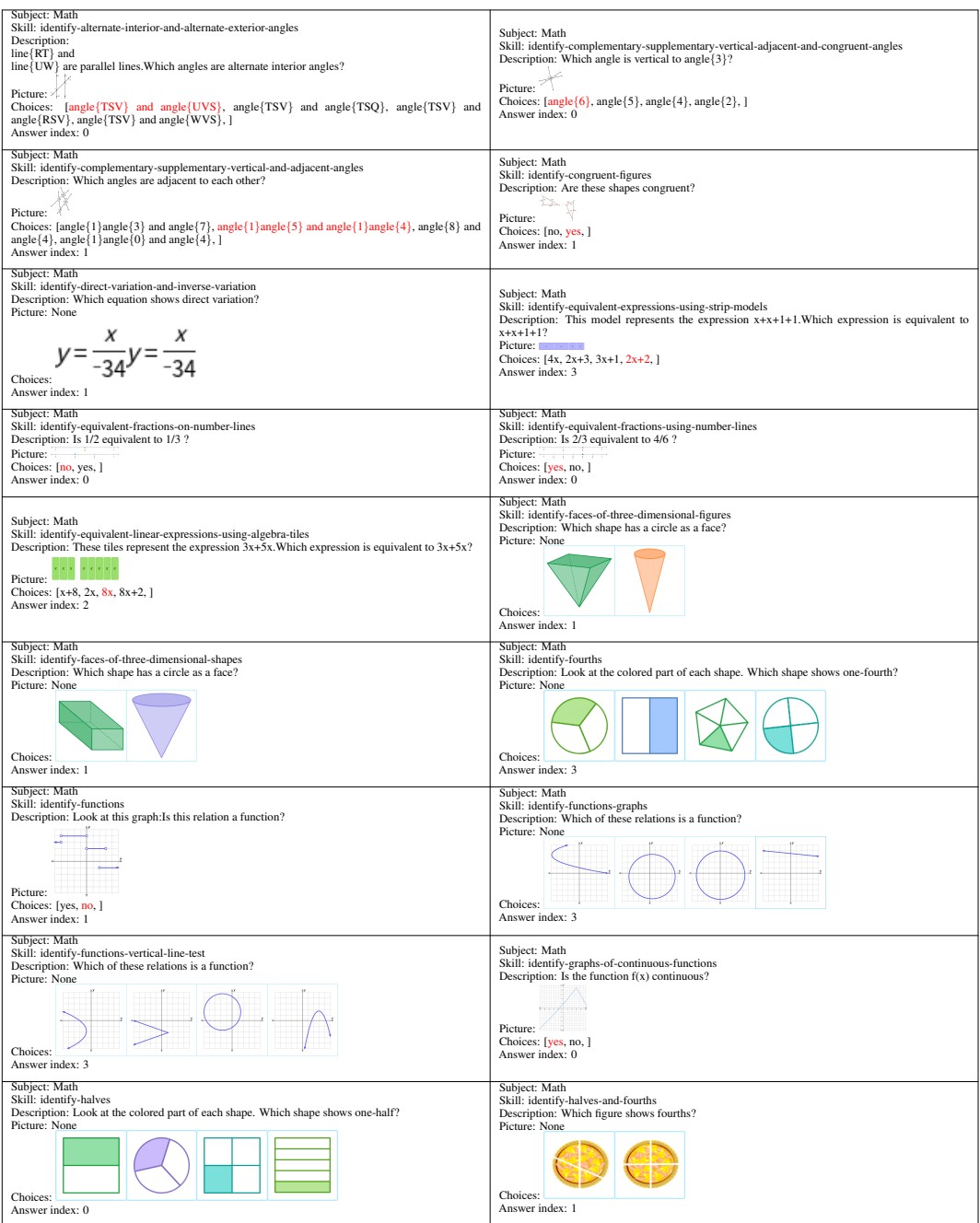

| | |
|---|---|
| Subject: Math
Skill: identify-alternate-interior-and-alternate-exterior-angles
Description:
line{RT} and
line{UW} are parallel lines.Which angles are alternate interior angles?
Picture:
Choices: [angle{TSV} and angle{UVS}, angle{TSV} and angle{TSQ}, angle{TSV} and angle{RSV}, angle{TSV} and angle{WVS}, ]
Answer index: 0 | Subject: Math
Skill: identify-complementary-supplementary-vertical-adjacent-and-congruent-angles
Description: Which angle is vertical to angle{3}?
Picture:
Choices: [angle{6}, angle{5}, angle{4}, angle{2}, ]
Answer index: 0 |
| Subject: Math
Skill: identify-complementary-supplementary-vertical-and-adjacent-angles
Description: Which angles are adjacent to each other?
Picture:
Choices: [angle{1}angle{3} and angle{7}, angle{1}angle{5} and angle{1}angle{4}, angle{8} and angle{4}, angle{1}angle{0} and angle{4}, ]
Answer index: 1 | Subject: Math
Skill: identify-congruent-figures
Description: Are these shapes congruent?
Picture:
Choices: [no, yes, ]
Answer index: 1 |
| Subject: Math
Skill: identify-direct-variation-and-inverse-variation
Description: Which equation shows direct variation?
Picture: None

$$y = \dfrac{x}{-34} \quad y = \dfrac{x}{-34}$$

Choices:
Answer index: 1 | Subject: Math
Skill: identify-equivalent-expressions-using-strip-models
Description: This model represents the expression x+x+1+1.Which expression is equivalent to x+x+1+1?
Picture:
Choices: [4x, 2x+3, 3x+1, 2x+2, ]
Answer index: 3 |
| Subject: Math
Skill: identify-equivalent-fractions-on-number-lines
Description: Is 1/2 equivalent to 1/3 ?
Picture:
Choices: [no, yes, ]
Answer index: 0 | Subject: Math
Skill: identify-equivalent-fractions-using-number-lines
Description: Is 2/3 equivalent to 4/6 ?
Picture:
Choices: [yes, no, ]
Answer index: 0 |
| Subject: Math
Skill: identify-equivalent-linear-expressions-using-algebra-tiles
Description: These tiles represent the expression 3x+5x.Which expression is equivalent to 3x+5x?
Picture:
Choices: [x+8, 2x, 8x, 8x+2, ]
Answer index: 2 | Subject: Math
Skill: identify-faces-of-three-dimensional-figures
Description: Which shape has a circle as a face?
Picture: None

Choices:
Answer index: 1 |
| Subject: Math
Skill: identify-faces-of-three-dimensional-shapes
Description: Which shape has a circle as a face?
Picture: None

Choices:
Answer index: 1 | Subject: Math
Skill: identify-fourths
Description: Look at the colored part of each shape. Which shape shows one-fourth?
Picture: None

Choices:
Answer index: 3 |
| Subject: Math
Skill: identify-functions
Description: Look at this graph:Is this relation a function?

Picture:
Choices: [yes, no, ]
Answer index: 1 | Subject: Math
Skill: identify-functions-graphs
Description: Which of these relations is a function?
Picture: None

Choices:
Answer index: 3 |
| Subject: Math
Skill: identify-functions-vertical-line-test
Description: Which of these relations is a function?
Picture: None

Choices:
Answer index: 3 | Subject: Math
Skill: identify-graphs-of-continuous-functions
Description: Is the function f(x) continuous?

Picture:
Choices: [yes, no, ]
Answer index: 0 |
| Subject: Math
Skill: identify-halves
Description: Look at the colored part of each shape. Which shape shows one-half?
Picture: None

Choices:
Answer index: 0 | Subject: Math
Skill: identify-halves-and-fourths
Description: Which figure shows fourths?
Picture: None

Choices:
Answer index: 1 |

Table 23: Question examples for each skill (part 3).

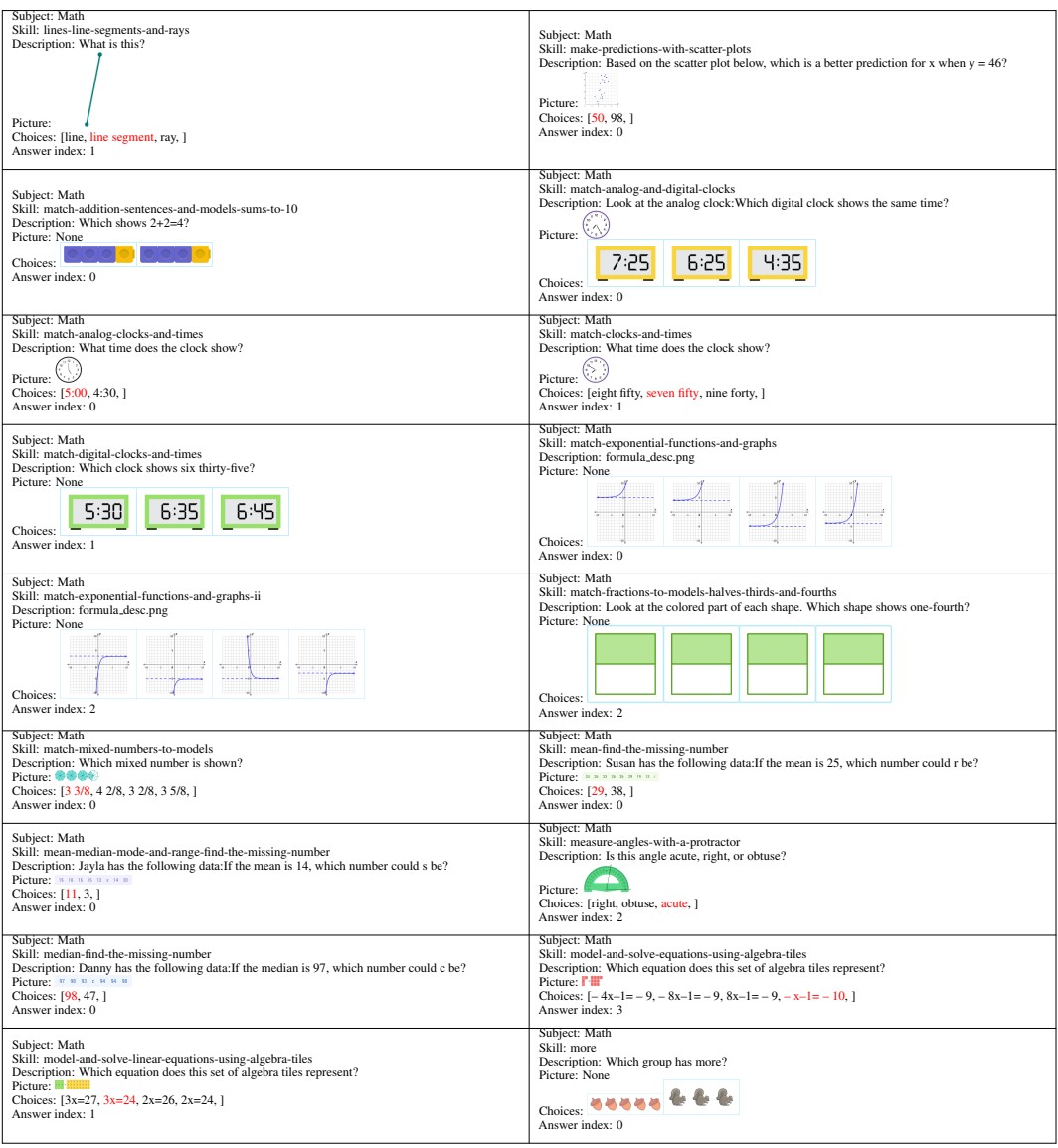

Table 24: Question examples for each skill (part 4).

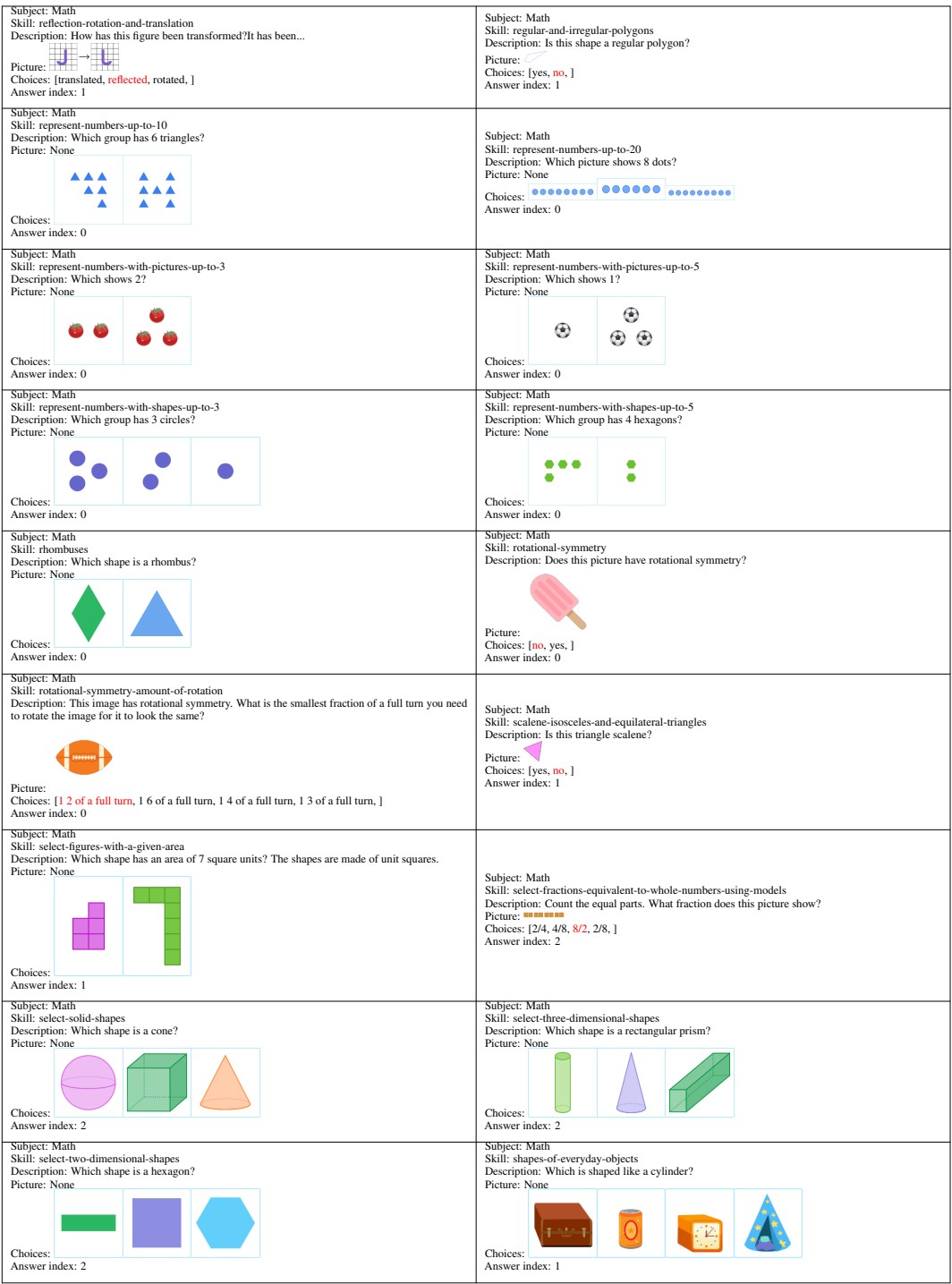

Table 25: Question examples for each skill (part 5).

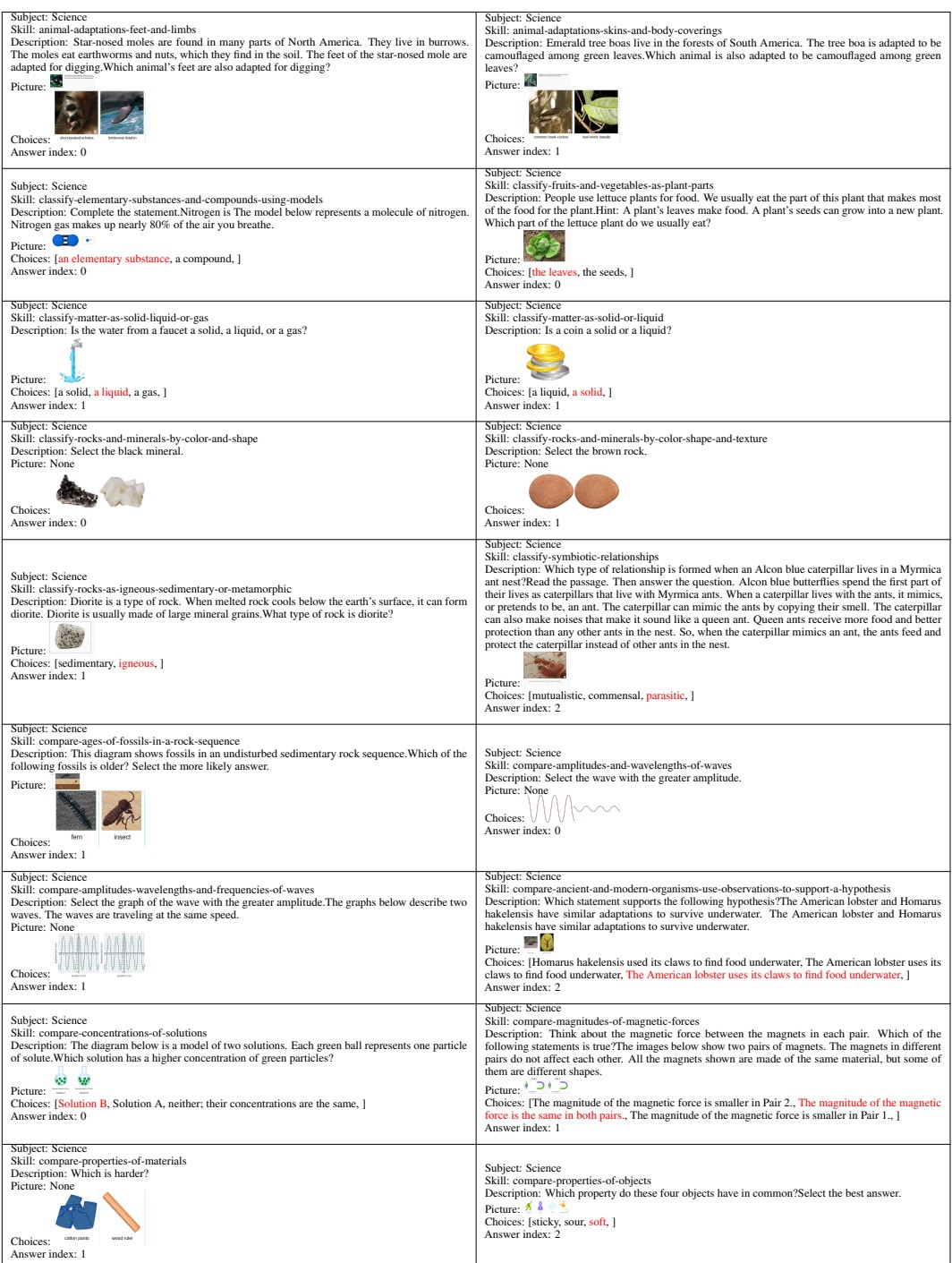

Table 26: Question examples for each skill (part 6).

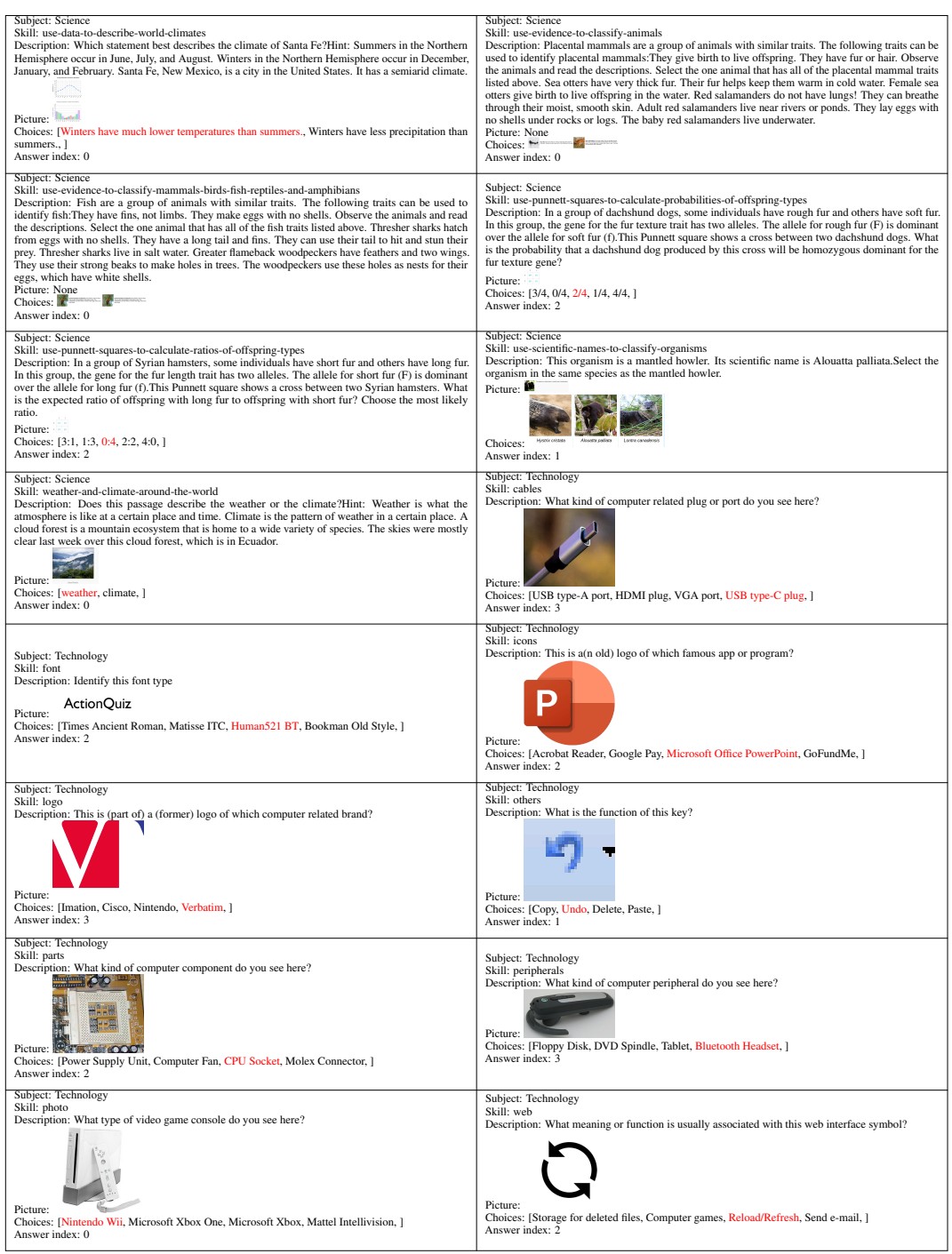

Table 27: Question examples for each skill (part 7).

