Description: iption: This table shows Jason's January budget. What could Jason do to balance his budget?

Picture:
Choices: [increase income from shoveling snow to $40, spend $15 less at Pizza Palace, spend only $40 at the arcade, spend $20 more on video games, ]
Answer index: 2

Subject: Math
Description: In solving this triangle, which law must you use first?

Picture:
Choices: [Law of Cosines, Law of Sines, ]
Answer index: 1

Subject: Math
Description: Is this angle acute, right, obtuse, or straight?
Picture:
Choices: [straight, obtuse, acute, right, ]
Answer index: 1

Subject: Math
Description: iption: Look at this cube: If the side lengths are tripled, then which of the following statements about its volume will be true?

Picture:
Choices: [The ratio of the new volume to the old volume will be 81:1., The ratio of the new volume to the old volume will be 1:8., The ratio of the new volume to the old volume will be 3:1., The ratio of the new volume to the old volume will be 27:1., ]
Answer index: 3

Subject: Math
Description: Which shape is a cone?
Picture: None

Choices:
Answer index: 2

Subject: Math
Description: Is the function f(x) continuous on the open interval (3,7)?

Picture:
Choices: [no, yes, ]
Answer index: 1

Subject: Math
Description: iption: Use the diagram to help you answer the question below. Which of the following is a rational number but not an integer?

Picture:
Choices: [–123, 83, 194, 6.53, ]
Answer index: 3

Subject: Math
Description: Is this polygon a trapezoid?
Picture:
Choices: [no, yes, ]
Answer index: 1

Subject: Math
Description: Look at the colored part of each shape. Which shape shows one-third?
Picture: None

Choices:
Answer index: 0

Subject: Math
Description: Which shape has 5 equal sides?
Picture: None

Choices:
Answer index: 1

Subject: Math
Description: iption: Identify the cross section of this object. Assume objects are perpendicular if they appear so.

Picture:

Choices:
Answer index: 0

Subject: Math
Description: Are there more circles or triangles?
Picture: ●▼●▼▼●●●
Choices: [ circles, triangles, ]
Answer index: 0

Subject: Math
Description: iption: Look at this shape: Which image shows a reflection?
Picture:
Choices: [C, A, B, ]
Answer index: 2

Subject: Math
Description: Is the function f(x) continuous?

Picture:
Choices: [no, yes, ]
Answer index: 0

Subject: Math
Description: An ice cream sundae costs 1 dollar and 41 cents. Do you have enough money to buy it?

Picture:
Choices: [yes, no, ]
Answer index: 0

Subject: Math
Description: Which shape has a triangle as a face?
Picture: None

Choices:
Answer index: 1

Subject: Math
Description: iption: Look at this figure: What is the shape of its bases?

Picture:
Choices: [decagon, octagon, rectangle, circle, ]
Answer index: 2

Subject: Math
Description: The graph below shows a function. Is its inverse also a function?

Picture:
Choices: [yes, no, ]
Answer index: 0

Subject: Math
Description: What is the range of this exponential function?

Picture:
Choices: [ {y | y > ⁻3}   {y | y ≤ ⁻3}   {y | y ≥ ⁻3}   all real numbers   {y | y < ⁻3} ]
Answer index: 0

Subject: Math

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

Subject: Engineer
Skill: evaluate-tests-of-engineering-design-solutions
Description: Which of the following could Eliana's test show?Eliana was taking part in her school's engineering competition. To win the competition, she needed to build the popsicle-stick bridge that would hold the most weight. She could use only 200 popsicle sticks. She had two different design ideas. She had to pick one of the designs to use in the competition. To test which design was strongest, Eliana built two prototypes, each with 200 popsicle sticks. She then added 1 kg weights to each prototype until one of them broke. The passage below describes how the engineering-design process was used to test a solution to a problem. Read the passage. Then answer the question below.
Picture:
Choices: [how much weight a bridge built with 300 popsicle sticks could hold, which design could hold more weight, ]
Answer index: 1

Subject: Engineer
Skill: identify-control-and-experimental-groups
Description: In this experiment, which were part of a control group?The passage below describes an experiment. Madelyn has a bubble machine and wants to know how to make the bubbles last longer. She read that bubbles burst when the liquid that makes up the bubbles evaporates. Madelyn knew that when liquids are warmer, they evaporate faster. So, she wondered if she could make her bubbles last longer by cooling the bubble solution. Madelyn cooled six bottles of bubble solution to 30°F below room temperature. She left another six bottles of bubble solution at room temperature. Then, she measured how long bubbles made from the solution in each bottle lasted.
Picture:
Choices: [the bottles that were cooled down, the bottles that were at room temperature, ]
Answer index: 1

Subject: Engineer
Skill: identify-independent-and-dependent-variables
Description: Hint: An independent variable is a variable whose effect you are investigating. A dependent variable is a variable that you measure.Which of the following was a dependent variable in this experiment? The passage below describes an experiment. Read the passage and think about the variables that are described. Giardia is a microscopic parasite that lives in water and can infect humans. Dr. Roth designed a drinking straw that contained a filter to remove Giardia from water. Dr. Roth wanted to know if a longer filtering straw would remove more Giardia. Dr. Roth made six filtering straws: three that were five inches long and three that were ten inches long. She prepared six one-liter batches of water, each containing 10,000 Giardia. Then, Dr. Roth passed one batch of water through each straw. After each batch passed through the straw, she used a microscope to count the number of Giardia that remained in a small sample of the water.
Picture:
Choices: [the number of Giardia that remained in the water, the length of the filtering straw, ]
Answer index: 0

Subject: Engineer
Skill: identify-laboratory-tools
Description: Select the round-bottom flask.
Picture: None
Choices:
Answer index: 1

Subject: Engineer
Skill: identify-the-experimental-question
Description: Identify the question that Jeffrey's experiment can best answer.The passage below describes an experiment. Read the passage and then follow the instructions below. Jeffrey mixed bacteria into a nutrient-rich liquid where the bacteria could grow. He poured four ounces of the mixture into each of ten glass flasks. In five of the ten flasks, he also added one teaspoon of cinnamon. He allowed the bacteria in the flasks to grow overnight in a 37°C room. Then, Jeffrey used a microscope to count the number of bacteria in a small sample from each flask. He compared the amount of bacteria in the liquid with cinnamon to the amount of bacteria in the liquid without cinnamon.
Picture:
Choices: [Do more bacteria grow in liquid with cinnamon than in liquid without cinnamon?, Does temperature affect how much bacteria can grow in liquid?, ]
Answer index: 0

Subject: Engineer
Skill: laboratory-safety-equipment
Description: Select the apron.
Picture: None
Choices:
Answer index: 3

Subject: Math
Skill: absolute-value-and-integers-word-problems
Description: Debbie likes watching the show Engineering Marvels. In last night's episode, the engineering team visited a tall skyscraper and a deep mine. A banner at the bottom of the screen showed the elevation of each location the team visited.Which location is closer to sea level?
Picture:
Choices: [bottom of the mine, top of the skyscraper, ]
Answer index: 0

Subject: Math
Skill: acute-obtuse-and-right-triangles
Description: What kind of triangle is this?
Picture:
Choices: [obtuse, right, acute, ]
Answer index: 0

Subject: Math
Skill: acute-right-obtuse-and-straight-angles
Description: Is this angle acute, right, obtuse, or straight?
Picture:
Choices: [straight, obtuse, acute, right, ]
Answer index: 1

Subject: Math
Skill: addition-sentences-up-to-10-what-does-the-model-show
Description: Which addition sentence does the picture show?
Picture:
Choices: [4+3=7, 5+2=7, ]
Answer index: 0

Subject: Math
Skill: addition-sentences-up-to-10-which-model-matches
Description: Which shows 8+1=9?
Picture: None
Choices:
Answer index: 0

Subject: Math
Skill: addition-sentences-up-to-5-what-does-the-model-show
Description: Which addition sentence does the picture show?
Picture:
Choices: [4+1=5, 3+1=4, ]
Answer index: 0

Subject: Math
Skill: addition-sentences-up-to-5-which-model-matches
Description: Which shows 2+1=3?
Picture: None
Choices:
Answer index: 0

Subject: Math
Skill: addition-sentences-using-number-lines-sums-up-to-20
Description: Which addition sentence does this model show?
Picture:
Choices: [3+3=6, 5+6=11, 5+4=9, 5+3=8, ]
Answer index: 3

Subject: Math
Skill: adjust-a-budget
Description: This table shows Angie's February budget.What could Angie do to balance her budget?
Picture:
Choices: [spend only $60 at the mall, and teach a baking class for $30, decorate more custom cookies to earn another $35, and spend only $60 at the mall, spend $20 more on the baking kit, and teach a baking class for $30, decorate more custom cookies to earn another $35, and spend $20 more on the baking kit, ]
Answer index: 1

Subject: Math
Skill: am-or-pm
Description: Farmer Keenan is getting up to go milk his cows. It is just before sunrise. His watch shows:What time is it?
Picture:
Choices: [4:30 P.M., 4:30 A.M., ]
Answer index: 1

Subject: Math
Skill: angle-angle-criterion-for-similar-triangles
Description: FGH and JKL are shown below.Which statement is true?
Picture:
Choices: [FGH is similar to JKL., FGH is not similar to JKL., There is not enough information to determine whether the triangles are similar., ]
Answer index: 2

Subject: Math
Skill: angles-as-fractions-of-a-circle
Description: What fraction of the circle does this angle cut out?
Picture:
Choices: [1/4, 3/4, 1 whole, 1/2, ]
Answer index: 0

Table 21: Question examples for each skill (part 1).

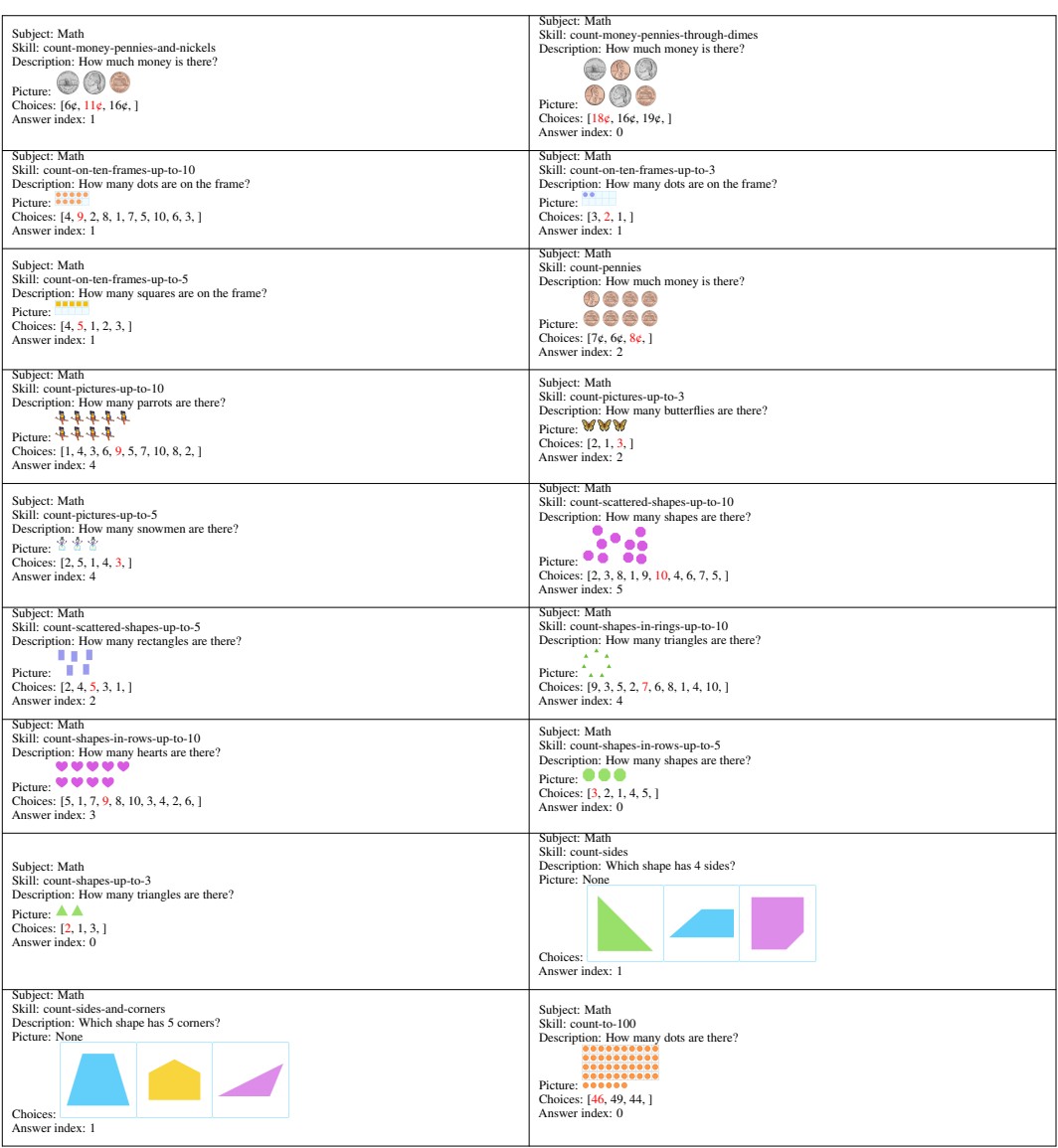

Table 22: Question examples for each skill (part 2).

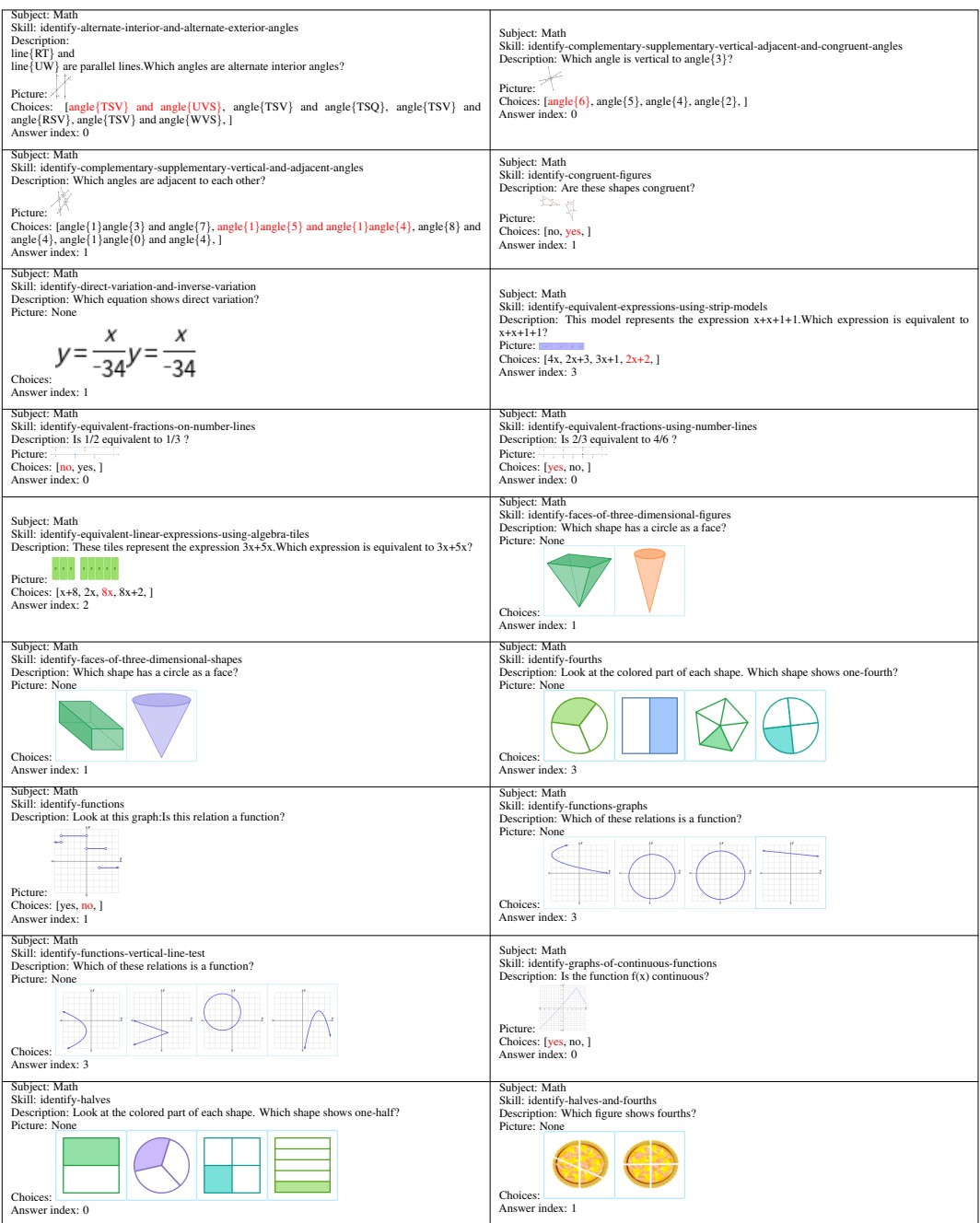

Table 23: Question examples for each skill (part 3).

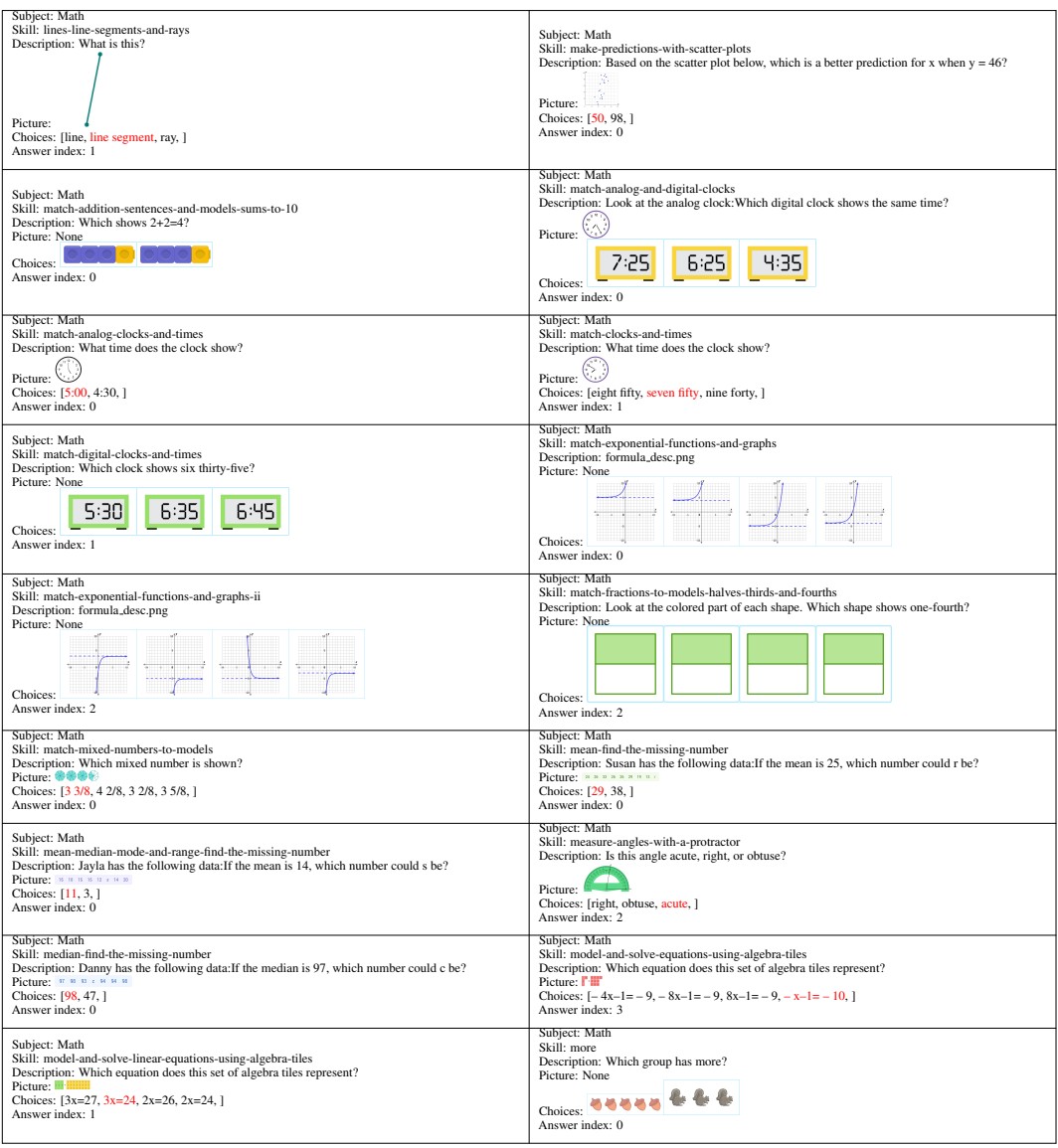

Table 24: Question examples for each skill (part 4).

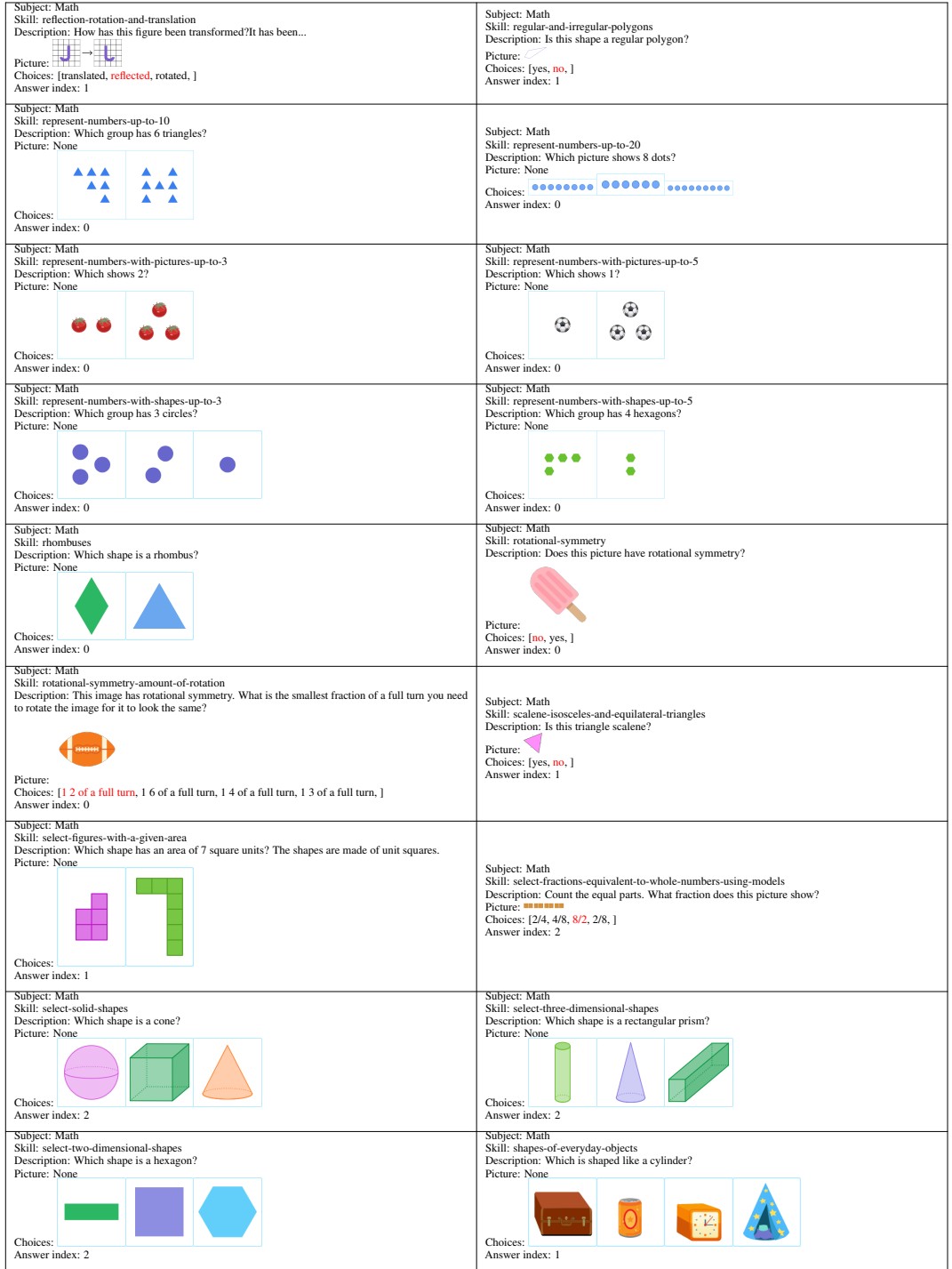

Table 25: Question examples for each skill (part 5).

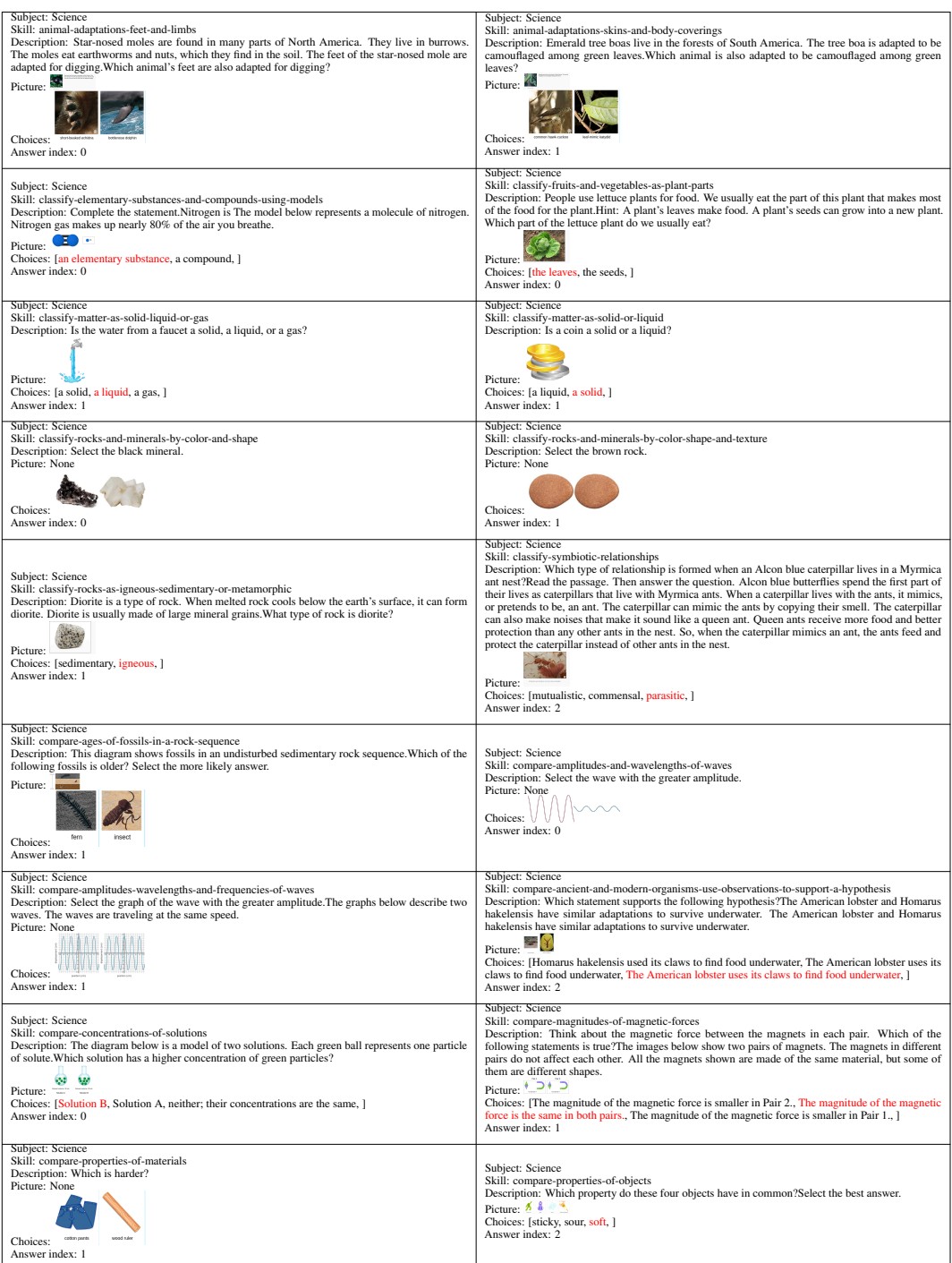

Table 26: Question examples for each skill (part 6).

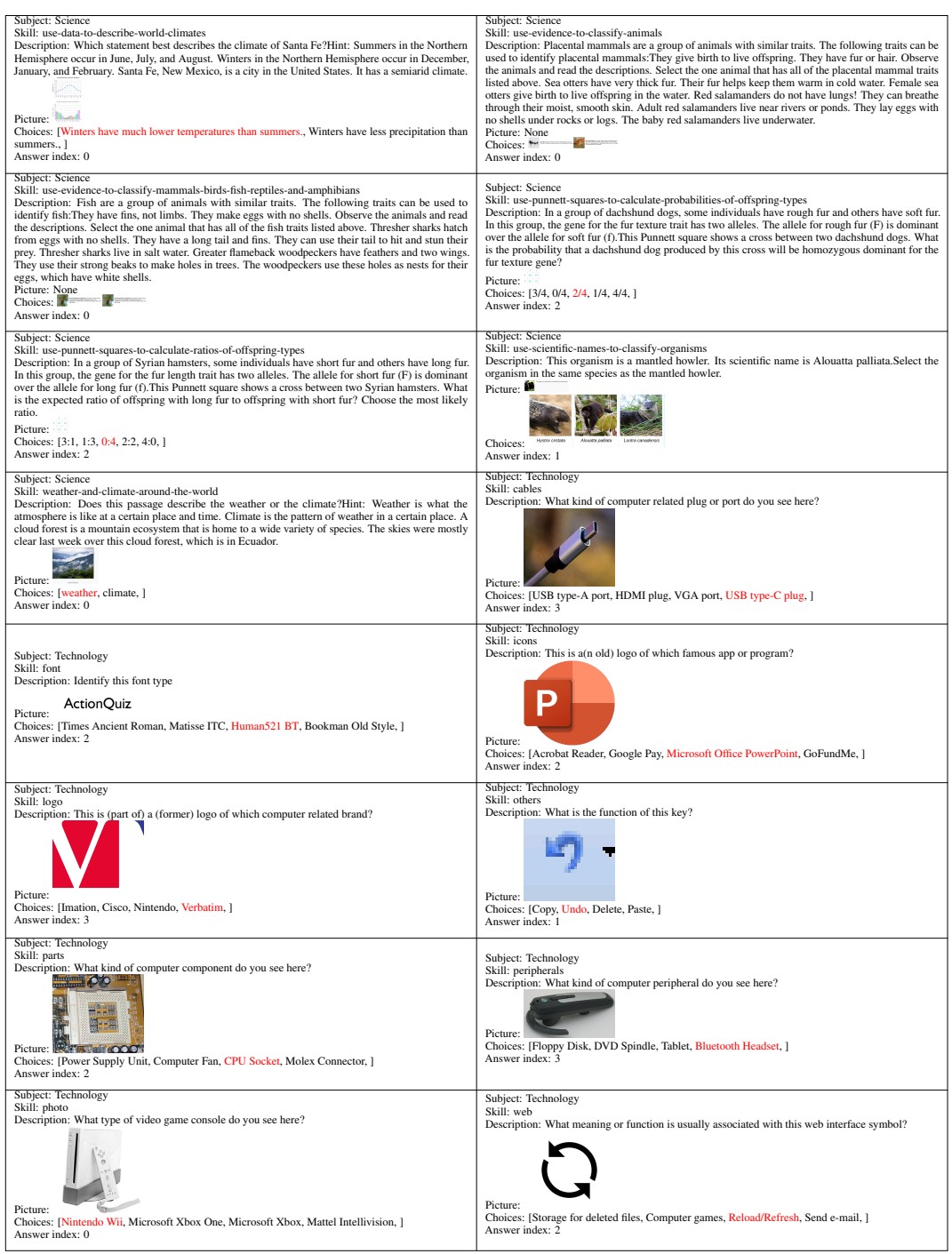

Table 27: Question examples for each skill (part 7).

**Subject: Science**
Skill: use-data-to-describe-world-climates
Description: Which statement best describes the climate of Santa Fe?Hint: Summers in the Northern Hemisphere occur in June, July, and August. Winters in the Northern Hemisphere occur in December, January, and February. Santa Fe, New Mexico, is a city in the United States. It has a semiarid climate.
Picture:
Choices: [Winters have much lower temperatures than summers., Winters have less precipitation than summers., ]
Answer index: 0

**Subject: Science**
Skill: use-evidence-to-classify-animals
Description: Placental mammals are a group of animals with similar traits. The following traits can be used to identify placental mammals:They give birth to live offspring. They have fur or hair. Observe the animals and read the descriptions. Select the one animal that has all of the placental mammal traits listed above. Sea otters have very thick fur. Their fur helps keep them warm in cold water. Female sea otters give birth to live offspring in the water. Red salamanders do not have lungs! They can breathe through their moist, smooth skin. Adult red salamanders live near rivers or ponds. They lay eggs with no shells under rocks or logs. The baby red salamanders live underwater.
Picture: None
Choices:
Answer index: 0

**Subject: Science**
Skill: use-evidence-to-classify-mammals-birds-fish-reptiles-and-amphibians
Description: Fish are a group of animals with similar traits. The following traits can be used to identify fish:They have fins, not limbs. They make eggs with no shells. Observe the animals and read the descriptions. Select the one animal that has all of the fish traits listed above. Thresher sharks hatch from eggs with no shells. They have a long tail and fins. They can use their tail to hit and stun their prey. Thresher sharks live in salt water. Greater flameback woodpeckers have feathers and two wings. They use their strong beaks to make holes in trees. The woodpeckers use these holes as nests for their eggs, which have white shells.
Picture: None
Choices:
Answer index: 0

**Subject: Science**
Skill: use-punnett-squares-to-calculate-probabilities-of-offspring-types
Description: In a group of dachshund dogs, some individuals have rough fur and others have soft fur. In this group, the gene for the fur texture trait has two alleles. The allele for rough fur (F) is dominant over the allele for soft fur (f).This Punnett square shows a cross between two dachshund dogs. What is the probability that a dachshund dog produced by this cross will be homozygous dominant for the fur texture gene?
Picture:
Choices: [3/4, 0/4, 2/4, 1/4, 4/4, ]
Answer index: 2

**Subject: Science**
Skill: use-punnett-squares-to-calculate-ratios-of-offspring-types
Description: In a group of Syrian hamsters, some individuals have short fur and others have long fur. In this group, the gene for the fur length trait has two alleles. The allele for short fur (F) is dominant over the allele for long fur (f).This Punnett square shows a cross between two Syrian hamsters. What is the expected ratio of offspring with long fur to offspring with short fur? Choose the most likely ratio.
Picture:
Choices: [3:1, 1:3, 0:4, 2:2, 4:0, ]
Answer index: 2

**Subject: Science**
Skill: use-scientific-names-to-classify-organisms
Description: This organism is a mantled howler. Its scientific name is Alouatta palliata.Select the organism in the same species as the mantled howler.
Picture:
Choices: Hystrix cristata | Alouatta palliata | Lontra canadensis
Answer index: 1

**Subject: Science**
Skill: weather-and-climate-around-the-world
Description: Does this passage describe the weather or the climate?Hint: Weather is what the atmosphere is like at a certain place and time. Climate is the pattern of weather in a certain place. A cloud forest is a mountain ecosystem that is home to a wide variety of species. The skies were mostly clear last week over this cloud forest, which is in Ecuador.
Picture:
Choices: [weather, climate, ]
Answer index: 0

**Subject: Technology**
Skill: cables
Description: What kind of computer related plug or port do you see here?
Picture:
Choices: [USB type-A port, HDMI plug, VGA port, USB type-C plug, ]
Answer index: 3

**Subject: Technology**
Skill: font
Description: Identify this font type
ActionQuiz
Picture:
Choices: [Times Ancient Roman, Matisse ITC, Human521 BT, Bookman Old Style, ]
Answer index: 2

**Subject: Technology**
Skill: icons
Description: This is a(n old) logo of which famous app or program?
Picture:
Choices: [Acrobat Reader, Google Pay, Microsoft Office PowerPoint, GoFundMe, ]
Answer index: 2

**Subject: Technology**
Skill: logo
Description: This is (part of) a (former) logo of which computer related brand?
Picture:
Choices: [Imation, Cisco, Nintendo, Verbatim, ]
Answer index: 3

**Subject: Technology**
Skill: others
Description: What is the function of this key?
Picture:
Choices: [Copy, Undo, Delete, Paste, ]
Answer index: 1

**Subject: Technology**
Skill: parts
Description: What kind of computer component do you see here?
Picture:
Choices: [Power Supply Unit, Computer Fan, CPU Socket, Molex Connector, ]
Answer index: 2

**Subject: Technology**
Skill: peripherals
Description: What kind of computer peripheral do you see here?
Picture:
Choices: [Floppy Disk, DVD Spindle, Tablet, Bluetooth Headset, ]
Answer index: 3

**Subject: Technology**
Skill: photo
Description: What type of video game console do you see here?
Picture:
Choices: [Nintendo Wii, Microsoft Xbox One, Microsoft Xbox, Mattel Intellivision, ]
Answer index: 0

**Subject: Technology**
Skill: web
Description: What meaning or function is usually associated with this web interface symbol?
Picture:
Choices: [Storage for deleted files, Computer games, Reload/Refresh, Send e-mail, ]
Answer index: 2