# OpenReview forum: "Measuring Vision-Language STEM Skills of Neural Models"
_ICLR.cc/2024/Conference — ICLR 2024 poster_

### Official Review · Reviewer_aT3Q · 2023-10-24

**Soundness:** 3 good
**Presentation:** 3 good
**Contribution:** 3 good
**Rating:** 6
**Confidence:** 4

**Summary:**

The paper proposes a new dataset to test the STEM skills of neural models. This dataset is one of the largest and most comprehensive datasets for the STEM challenge, especially on fundamental skills in the K-12 curriculum. They test state-of-the-art foundation models on this challenge, but found limited performance. Even when training these models on a training split, the performance is still not satisfactory, which indicates the difficulty of this challenge and needs novel algorithms.

**Strengths:**

${\bf Strengths:}$

[$\textbf{New Dataset for STEM}$] This work introduces one of the largest and most comprehensive datasets for the STEM challenge.

[$\textbf{Presentation Quality}$] The presentation is clear and easy to follow.

**Weaknesses:**

${\bf Weaknesses:}$

[$\textbf{Missing Comparison}$] This work only shows the results of foundation models on the proposed dataset. It would be nice to also see the performance of the same foundation models on other datasets, which helps understand the difficulty of different datasets.

[$\textbf{Missing Limitation and Potential Impact}$] It would be nice to show the limitations and potential impacts of the proposed dataset.

[$\textbf{Confusions in Figure 9}$] In Figure 9, for RN50, RN101 and ViT-B/16, the performance decreases when the model size becomes larger. Could the authors explain this a bit more?

[$\textbf{Unconvincing Human Study}$] In the human study, the inclusion of only seven university students may not be fully representative, potentially limiting the generalizability of the findings to the broader population and raising considerations about the robustness of the observed human performance.

**Questions:**

- Could the authors show many examples for better understanding the dataset?

- Could the authors provide some directions for improving the performance according to your expertise?

**Details Of Ethics Concerns:**

The dataset may contain some human faces. I am not sure whether this has the privacy issue.

---

> ### Author Response · Authors · 2023-11-19
>
> We are glad that you like the paper! The responses to your key questions are below.
>
> **Comparison with other datasets**: We agree that comparing STEM with other datasets will help us better understand the difficulty of different datasets. As the goal of our dataset is to measure the fundamental STEM understanding of neural models, we focus on the evaluation of different models on our dataset. We provide the comparison of statistics of different datasets in Figure 1 (a). This follows the typical benchmark paper design such as VQA [1], MMLU [2], IconQA [3]. However, this is a new angle for the community, and among our future investigations.
>
> **Limitation and impact**: We understand the importance of limitations and potential impact. We have included the discussion in the ethics statement. We plan to expand the discussions to focus on the potential use of models after deep analysis of our dataset. For example, it will follow common limitations of the foundation models such as misuse, bias, and hallucinations [4, 5, 6, 7, 8].
>
> **Figure 9 clarification**: Thank you for pointing this out. Although the general trend is that the model performance increases when models become larger, as the scaling law indicated [9],  the model’s performance does not only depend on the size, other factors could also contribute (e.g., the length of the training). It is possible for a model with fewer parameters to outperform another model, which might explain why ViT-B/16 is outperformed by RN50 and RN101.
>
> **Human study**: As described in Sec. 2.4 and Sec. 3.3, our exam scores are calculated based on millions of IXL elementary users [10]. This is one of the main advantages of using our dataset, as the human performances are simulated in a really large environment. The IXL SmartScore [10] serves as the main resource for our human performance comparison. The performance of the graduate students mainly serves as an expert performance, setting up an upper bound for human performance, while the IXL exam scores are from elementary students. We agree that the pool of test takes can be expanded to more people, but it seems a common approach in recent literature such as MATH [11]. We will consider adding more expert test takers in future work.
>
> **More dataset examples**: Due to space limitations, we have included dataset examples for all skills of our dataset in Appendix C and a case study in Section 3.4.
>
> **Directions for improvement**: Thanks for raising this question. We are also considering approaches to improve the performance on the dataset. One possible direction is to include more high-quality textbook data during training since STEM is knowledge-intensive. Another direction could be allowing models to access relevant knowledge resources during inference.
>
> **Ethics statement**: As indicated in our statement, the collected data does not contain sensitive data and the copyright restrictively follows the original data sources. We will add more discussion based on your suggestion.
>
> References:
>
> [1] Antol, Stanislaw, et al. "Vqa: Visual question answering." Proceedings of the IEEE international conference on computer vision. 2015.
>
> [2] Hendrycks, Dan, et al. "Measuring massive multitask language understanding." arXiv preprint arXiv:2009.03300 (2020).
>
> [3] Lu, Pan, et al. "Iconqa: A new benchmark for abstract diagram understanding and visual language reasoning." arXiv preprint arXiv:2110.13214 (2021).
>
> [4] Radford, Alec, et al. "Learning transferable visual models from natural language supervision." International conference on machine learning. PMLR, 2021.
>
> [5] Brown, Tom, et al. "Language models are few-shot learners." Advances in neural information processing systems 33 (2020): 1877-1901.
>
> [6] Ouyang, Long, et al. "Training language models to follow instructions with human feedback." Advances in Neural Information Processing Systems 35 (2022): 27730-27744.
>
> [7] Anil, Rohan, et al. "Palm 2 technical report." arXiv preprint arXiv:2305.10403 (2023).
>
> [8] Touvron, Hugo, et al. "Llama 2: Open foundation and fine-tuned chat models." arXiv preprint arXiv:2307.09288 (2023).
>
> [9] Kaplan, Jared, et al. "Scaling laws for neural language models." arXiv preprint arXiv:2001.08361 (2020).
>
> [10] Learning, I. X. L. "The impact of IXL Math and IXL ELA on student achievement in grades pre-K to 12 (pp. 1–27)." (2019).
>
> [11] Hendrycks, Dan, et al. "Measuring mathematical problem solving with the math dataset." arXiv preprint arXiv:2103.03874 (2021).

---

> > ### Comment · Reviewer_aT3Q · 2023-11-23
> > **Response to the Rebuttal**
> >
> > Thanks for the authors' response. It has solved most of my concerns. Considering the overall quality of paper, I will keep the "weak acceptance" score.

---

> ### Author Response · Authors · 2023-11-23
> **Response to Reviewer aT3Q (cont.)**
>
> Thanks for your positive feedback regarding our clarifications. It would be great if you could share your remaining concerns that have not been addressed yet. Despite the short time window left till the end of the discussion, we will try our best to address them. We appreciate your continued engagement and support.

---

### Official Review · Reviewer_FcAF · 2023-11-01

**Soundness:** 3 good
**Presentation:** 3 good
**Contribution:** 3 good
**Rating:** 6
**Confidence:** 4

**Summary:**

This paper propose a new dataset, named STEM, consisting of science, technology, engineering and math subjects. The proposed STEM includes 448 skills and 107M+ questions, which is a large-scale multimodal dataset. Upon the new dataset, this paper analyzes some existing foundation models, e.g., CLIP and ChatGP. Experiments show that these models are still below the human although the model is finetuned. Code and dataset are available.

**Strengths:**

1. The new dataset STEM is valuable, which consists a large-scale skills and questions.
2. The analyses of some foundation models on STEM are detailed and interesting.

**Weaknesses:**

1. Despite the analyses of foundation models, e.g., CLIP and ChatGPT, there are still some new and better models missing, such as EVA-CLIP, Kosmos-2, BLIP-2 and etc. What are the results of these models?
2. Does the quality of the caption model influence the performance of zero-shot performance on vision-language models. How about changing the caption model to a more accurate one, e.g., BLIP-2, GPT4 (I understand that the API may be not available now)?
3. What is the performance on other datasets after finetuning on STEM, e.g., VQA (2015)? Which is better when comparing with the model before finetuning? It would be important to verify that STEM is important and finetuning on STEM can maintain or improve the generalization of the model.

**Questions:**

Please refer to the Weakness Section.

---

> ### Author Response · Authors · 2023-11-19
>
> We appreciate your endorsement of the work’s contributions! Please find our response to your questions below.
>
> **Other models**: We agree that the evaluation of more models will be valuable. The main contribution of our paper is the creation of our dataset to evaluate the multimodal STEM understanding. We choose the foundation models like CLIP for a pilot study on this dataset. We anticipate performance improvements from the recent models and will add the discussion on the models you suggested in the new version.
>
> **Caption model**: Thank you for the helpful suggestion and we believe that caption models will influence the performance to some extent. As discussed in Appendix B.5, it seems the textual input (e.g., prompt) does not impact the performance significantly. But we will investigate this in future work.
>
> **Performance of the finetuned model**: We agree that the generalization after finetuning on STEM is important. Our main contribution is the creation of the STEM dataset. The dataset is designed for evaluating the vision-language STEM Skills of neural models. Therefore, the generalization might be out of the scope of the current paper. However, we plan to add the results on VQA (as you suggested) in the new version.

---

> > ### Comment · Reviewer_FcAF · 2023-11-23
> >
> > Thanks for the authors' response. I acknowledge the contribution of the dataset, while I still look forward to seeing more contributions when applying this valuable data to various newly released foundation models.
> > Therefore, I keep my initial rating of "marginally above the acceptance threshold".

---

> ### Author Response · Authors · 2023-11-22
> **Response to Reviewer FcAF (cont.)**
>
> Per your suggestion, we added VQA results of the zero-shot CLIP model and models finetuned on each subject in Appendix B.7. The average increase of the finetuned models over the zero-shot setting is 1.2%. This shows finetuning on our dataset can potentially help improve the generalization of models.

---

### Official Review · Reviewer_psfv · 2023-11-01

**Soundness:** 3 good
**Presentation:** 4 excellent
**Contribution:** 3 good
**Rating:** 6
**Confidence:** 4

**Summary:**

- The paper proposes a new dataset, STEM, to test the STEM skills of neural models. Focusing on fundamental skills in the K-12 curriculum, STEM features a high coverage of skills in all four subjects, a large number of questions, and its multi-modal property.

- The paper benchmarks a wide range of neural models, including baselines and SOTA models for both language models and vision-language models. Results show that neural models are still behind human performance.

- The paper breaks down its result analysis into different granularities, such as skills, subjects and grades, and identifies the shortcomings of models and challenges for future research.

**Strengths:**

1. The paper introduces a novel and challenging benchmark for multi-modal STEM skills. Compared to existing benchmarks, STEM covers all four subjects, contains more skills and questions, and strictly excludes single-modal questions. Therefore, it significantly contributes to the advancement of neural models' multi-modal skills.

2. The paper benchmarks a wide set of neural models, including the SOTA models such as ChatGPT and CLIP. The paper also analyzes the results at different granularities and in detail.

3. The paper presents illustrative figures and charts. For example, Figure 1(a) serves as a good demonstration of STEM questions that give the readers a good sense on what the dataset looks like; Figure 4 clearly shows the different mechanisms of CLIP model and ChatGPT model.

**Weaknesses:**

1. The paper does not present strong enough evidence for its claims. The human evaluation part is not clear enough. Therefore, I am not convinced that current neural models fall far behind the human performance on the STEM dataset.
2. When benchmarking on SOTA models, the paper does not use the newest language model, i.e., GPT-4. If GPT-4 is tested, we might see quite significantly different results.
3. The paper touched the bad case analysis, but not deep enough. As part of its biggest contribution, it will be better to make a more thorough analysis on the examples and patterns where the models fail.

**Questions:**

1. The claims that the paper concluded seem inconsistent with the results it presented. In the abstract, the paper claims that "these models are still well below (averaging 54.7%) the performance of elementary students, not to mention near expert-level performance." My questions and concerns are:
  - How do we define "the performance of elementary students"? I see that the paper compares the model performance with a score of 90, which "according to IXL, is considered excellent for a mastered skill", but it does not make sense to take a score of excellence as "the performance of elementary students", because it is an "expert-level" human performance.
  - Where is the number 54.7% from?
  - Besides, it doesn't seem fair to use Ph.D. students' test scores for the human evaluation of these elementary school questions.
2. What is the reason of not experimenting on GPT-4? It would be beneficial to test it since its performance has largely increased compared to GPT-3.5

---

> ### Author Response · Authors · 2023-11-19
>
> We appreciate your identification of the importance of the work. Our response to your review is below.
>
> **Human evaluation clarification**: We have included descriptions of the human evaluation e.g., setup, procedure, and metrics in Sec. 2.4 and Sec. 3.3. Overall, not only the average exam score of the tested models is below 60, but also their average accuracy is below 60 (Figure 1b). The results support our findings that model performance falls behind human performance on our dataset. Regarding the sources of the human scores, first, the exam scores of elementary students are sourced based on the IXL SmartScore [1] from grades 1 to 6 and we chose 90 as the reference. So if a skill belongs to grade 1, then 90 is for students at grade 1. According to the official guideline of SmartScore [1], a score higher than 90 is considered excellence for a mastered skill, and this standard also takes into account the grade level of the test takers, which are elementary students if we focus on grades 1 to 6. It means that a score of 90 is achievable and a reasonable goal for elementary students. Note that this score is still at the elementary level, and is considered excellent only for elementary students. Second, for expert-level human performance, we do not consider the grade level information so we test graduate students to get the upper bound performance of humans, which we refer to the expert-level performance. Since the SmartScore is only designed for the corresponding grades, we use graduate students’ test scores for the expert-level evaluation. We will add these clarifications in the new version.
>
> **The number of 54.7%**: Thank you for pointing this out. We calculate the average exam scores of zero-shot CLIP from grades 1 to 6 which equals 40.8. This is 54.7% lower than the exam score of 90.
>
> **GPT-4**: The main factor for excluding the GPT-4 results was cost. The approximate cost of evaluating GPT-4 on our benchmark is $6500. Although we don’t include GPT-4, we believe the main contribution of our paper is the multimodal dataset. Given the current results including GPT-3.5 Turbo, we expect the latest models like GPT-4 could improve the performance on our benchmark. This is indeed aligned with the value of contributing our dataset to the research community.
>
> **Error analysis**: We did include the overall error analysis in Sec. 3.4 and more detailed analysis in the appendix (Appendix B.3) due to space limit. We also investigate the performance variance along many other aspects including questions with different skills (Table 8, Figure 23-28), different numbers of answers (Figure 17), questions with different lengths (Figure 18), different question types (Figure 19), different grades (Figure 21). In addition, we will add more error analysis including the top-5 error rate skills for science and math subjects in the Appendix and give an example for each skill.
>
> References:
>
> [1] Learning, I. X. L. "The impact of IXL Math and IXL ELA on student achievement in grades pre-K to 12 (pp. 1–27)." (2019).

---

> > ### Comment · Reviewer_psfv · 2023-11-23
> >
> > Thanks for your response and clarification. Most of my questions have been answered by your comments, and your changes to the paper look good. I still have one more concern: I scanned the guideline of SmartScore you referred to, but didn't find the identification of 90 as a score of excellence. I did find this: "A student is considered proficient in a skill when they reach a SmartScore of 80. A student is considered mastery in a skill when they reach a SmartScore of 100." However, this is inconsistent with the paper's claim that "A score of 90 means a student is proficient in the subject." Could you share more thoughts on this?
> >
> > Besides, I feel it's helpful to add one or two sentences in section 2.4 to briefly introduce the mechanism of SmartScore. Something like "the SmartScore starts at 0, increases as students answer questions correctly, and decreases if questions are answered incorrectly." will be good.

---

> > > ### Author Response · Authors · 2023-11-23
> > > **Response to Reviewer psfv (cont.)**
> > >
> > > Thanks for your continued engagement and the opportunity to clarify things! We included the references [1,2] containing further descriptions of the score of 90 in the updated version. We also added the sentence you suggested in Sec. 2.4. We really appreciate these insightful suggestions.
> > >
> > > References:
> > >
> > > [1] https://www.ixl.com/help-center/article/1272663/how_does_the_smartscore_work
> > >
> > > [2] https://blog.ixl.com/wp-content/uploads/2014/11/SmartScore-guide.pdf

---

> > > > ### Comment · Reviewer_psfv · 2023-11-23
> > > >
> > > > Thank you for the clarification and your continued work to refined the paper. After reviewing the comments and your modified work, I decided to change my score to “6: marginally above the acceptance threshold”.

---

> > > > > ### Author Response · Authors · 2023-11-23
> > > > > **Response to Reviewer psfv (cont.)**
> > > > >
> > > > > Thank you! We really appreciate your valuable suggestions for improving the quality of the work. If you have additional questions, we’d be happy to address them before the end of the discussion period.

---

### Author Response · Authors · 2023-11-21
**A New Version has been Uploaded**

We express our sincere appreciation to the reviewers for your insightful feedback on our paper. The opportunity to receive such valuable input has allowed us to thoroughly enhance our work. We have given serious consideration to all the reviews received and have made appropriate revisions to our paper as a result. A summary of changes can be found at the beginning of the Appendix. We are committed to further improving the quality and impact of our research.

---

### Meta-Review · Area_Chair_bj8n · 2023-12-15

**Metareview:**

This work proposes a new large-scale, multi-modal dataset called STEM to test models on science, technology, engineering and math subjects. An extensive benchmarking of a wide range of language and vision-language models (including baseline and state-of-the-art models) is conducted and showcases the current gap to human-level performance. As such, this dataset provides a valuable resource for the community to benchmark progress on STEM understanding in foundation models. Addressing reviewers concerns about benchmarking the newest models and a better human study would improve this paper.

**Justification For Why Not Higher Score:**

While the experimental evaluation is extensive, reviewers pointed out that some of the strongest state-of-the-art models were not included. Furthermore, there were some concerns about the human study were raised (e.g. small sample size limiting strength of conclusions).

**Justification For Why Not Lower Score:**

Paper presents an novel dataset that can be used for benchmarking STEM skills of foundation models which I believe is an interesting area of future research

---

### Decision · Program_Chairs · 2024-01-16

Accept (poster)